# Modular gateway-ness connectivity and structural core organization in maritime network science

Mengqiao Xu [1,4✉], Qian Pan [1,4], Alessandro Muscoloni [2], Haoxiang Xia [1✉] &
Carlo Vittorio Cannistraci [2,3✉]

Around 80% of global trade by volume is transported by sea, and thus the maritime transportation system is fundamental to the world economy. To better exploit new international shipping routes, we need to understand the current ones and their complex systems association with international trade. We investigate the structure of the global liner shipping network (GLSN), finding it is an economic small-world network with a trade-off between high transportation efficiency and low wiring cost. To enhance understanding of this trade-off, we examine the modular segregation of the GLSN; we study provincial-, connector-hub ports and propose the definition of gateway-hub ports, using three respective structural measures. The gateway-hub structural-core organization seems a salient property of the GLSN, which proves importantly associated to network integration and function in realizing the cargo transportation of international trade. This finding offers new insights into the GLSN's structural organization complexity and its relevance to international trade.

[1] School of Economics and Management, Dalian University of Technology, No. 2 Linggong Road, Ganjingzi District, Dalian City, Liaoning Province 116024, China. [2] Biomedical Cybernetics Group, Biotechnology Center (BIOTEC), Center for Molecular and Cellular Bioengineering (CMCB), Center for Systems Biology Dresden (CSBD), Cluster of Excellence Physics of Life (PoL), Department of Physics, Technische Universität Dresden. Tatzberg 47/49, 01307 Dresden, Germany. [3] Center for Complex Network Intelligence (CCNI), Tsinghua Laboratory of Brain and Intelligence (THBI), Tsinghua University. 160 Chengfu Rd., SanCaiTang Building, Haidian District, Beijing 100084, China. [4] These two authors contributed equally: Mengqiao Xu, Qian Pan. ✉email: stephanie1996@sina.com; hxxia@dlut.edu.cn; kalokagathos.agon@gmail.com

Maritime transport, by far the most cost-effective way (in terms of freight cost) to the mass movement of goods and raw materials across the globe, is the backbone of international trade. Around 80% of global trade by volume and over 70% of global trade by value are carried by sea and are handled by ports worldwide[1]. The importance of maritime transport in supporting international trade makes it indispensable to the sustainable economic development of our world[2–4]. For individual economies, access to world markets depends largely on their transport connectivity, especially as regards liner shipping (i.e., the service of transporting goods primarily by ocean-going container ships that transit regular routes on fixed schedule).

The global liner shipping network (GLSN) is a self-organized complex transportation network, as a result of world's individual liner shipping companies' service networks. The function of the GLSN in supporting international trade is to transport containerized cargoes between countries, which it does by shipping cargoes from port to port across the GLSN until they reach their intended destinations. Certainly, the structure of the GLSN will affect how it accomplishes this function. A fundamental theoretical hypothesis in network science is the concept that structure matters, positing that the functional outcomes of a complex networked system, at both the system level and the individual node level, depend at least in part on the network structure[5–9]. The GLSN, though having been investigated by previous studies from a classical network perspective[10–15], introduction of innovative network science methodologies that characterize the structural organization complexity of GLSN and its complex system association with international trade remain a scientific ambition to pursue. Scientific advancements in these directions could provide novel methodological approaches for quantifying the structural dynamics (the connectivity changes that occur along time due to modifications in liner shipping service routes) of the GLSN and their relevance to international trade.

These facts motivate us to address a central research question of the present study: what specific topological properties and organization principles characterize the GLSN structure, and how are they associated with the GLSN's functional outcomes? Among topological properties, the structural-core organization is crucial, as it refers to the emergence of certain hub ports that (as a cohesive core) play an important role in the structural integration of the entire network. This corresponds to a specific question in liner shipping industry: which ports are the most important hubs in the GLSN structure, in the sense that they form a core that facilitates the efficiency of cargo transportation in the network? Such question is practically relevant because one of the most important issues in liner shipping service network design is to strategically pre-choose hub ports[16–18]. Methodologically, we pursue this particular research interest through investigating the modular community structure of the GLSN, since modular community structure is one of the most prominent properties of complex networks[19] and can influence their function and structural core organization. Indeed, we bring forward an analysis to elucidate how the structural integration of a modularly segregated network is achieved via network hubs[20,21] that resembles a core. Then, we explore the association between a new network structural measure (that we propose and term gateway-ness) and the GLSN's two functional outcomes of practical importance: individual ports' economic performance (i.e., ports' traffic capacities in liner shipping), which is at the node level; countries' international trade statuses (i.e., the international trade value (ITV) of countries and the bilateral trade value (BTV) between countries), which is at the system level.

Here we unveil the structural organization complexity of a recent GLSN, finding it a remarkably economic small-world network[22,23]. We study the modular community structure of the GLSN and the related three types of network hubs (i.e., provincial, gateway, and connector hubs). We discover that the GLSN presents a gateway-hub structural core, which proves to be topologically central and important in supporting long-distance maritime transportation. The gateway-ness strongly associates with ports' economic performance, and the gateway-hub structural core (detected by virtue of the gateway-ness) strongly associates with countries' international trade statuses. Our results highlight that the gateway-hub structural core is a salient topological property of the GLSN, which facilitates the structural integration of this network and is highly relevant to the network's functional outcomes with respect to international trade. The gateway-ness adds a valuable new tool in complex modular network analysis.

## Results

**Data for the GLSN construction**. We collected the data on world's liner shipping service routes from a leading database in liner shipping industry (see "Methods"). Figure 1 shows how we constructed a GLSN using such data (see "Methods"): each service route forms a complete graph where each port in the service route is connected to all the others; by merging all the complete graphs derived from individual service routes, we obtained a GLSN consisting of 977 nodes (i.e., ports) and 16,680 edges (i.e., inter-port links).

**Basic topological properties and economic small-world-ness**. Figure 2 summarizes the basic topological properties of the GLSN. The cumulative probability distribution of port degree (i.e., number of links a port has) is well fitted by an exponential function (Fig. 2a), consistent with the finding of previous work[11,14]. A port's betweenness is defined as the fraction of shortest paths between any two ports that pass through the given port[24]; the betweenness distribution of the GLSN presents a power-law tail (Fig. 2b), similar to that of air transport networks[25,26]. Closeness centrality of a port is defined as the inverse of the average shortest path length between this port and all other ports[27]; more than 80% ports are of closeness centrality larger than 0.333 (Fig. 2c), indicating that cargo transportation between these ports and others can be realized via transshipment within twice, on average. Assortativity[28] measures the tendency that ports with high degrees may connect randomly or preferentially to one another (see "Methods"); the GLSN is close to a neutral assortativity ($= -0.024$). Local-community-paradigm correlation (LCP-corr)[29] examines the tendency that the common neighbor ports between the two end ports of a link may connect with each other (see "Methods"); the GLSN displays a local community paradigm organization (LCP-corr $= 0.97$), similar to air transport networks[29]. The GLSN is a small-world network, with an average shortest path length of 2.671 and an average clustering coefficient of 0.713; small-world-ness is confirmed by two tests[30,31] (see "Methods").

We further investigated the economic small-world-ness properties of the GLSN (see "Methods"). Many real-life small-world networks (e.g., brain networks, communication networks and transportation networks) are found to be economic, in that their network configurations support high global and local efficiency with low wiring cost[22,23]. For spatially embedded networks, efficiency in the flow transfer between a pair of ports is defined as the reciprocal of the shortest distance between them (i.e., the smallest sum of the physical distance throughout all the possible paths between them). Global efficiency of the GLSN is calculated as the average efficiency of all pairs of nodes in the network. Local efficiency of a port is calculated as the global efficiency of the subnetwork consisting of all the neighbors of this

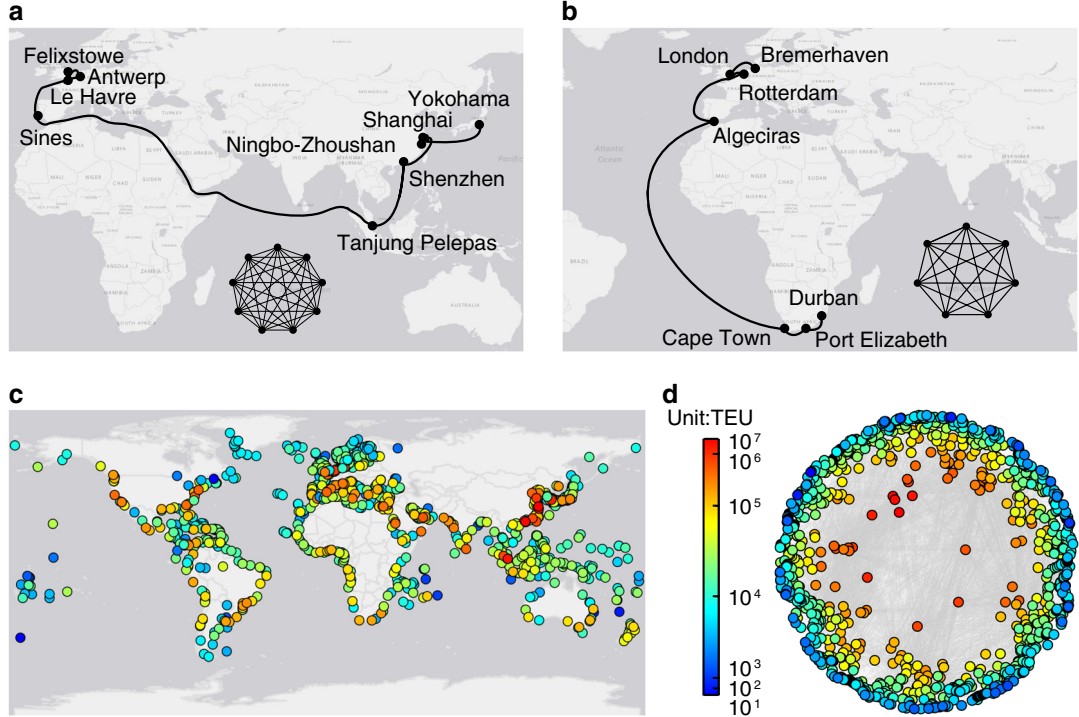

**Fig. 1 Construction of the global liner shipping network.** With the information on ports of call of world's individual liner shipping service routes, we made each service route a complete graph where any two ports in the service route were connected via an edge. By merging the complete graphs derived from all individual liner shipping service routes, we obtained the GLSN. See "Methods", for details about the adopted data on world's liner shipping service routes and details about the adopted network topology representation method for GLSN construction. In (**a** and **b**), we show how complete graphs are derived from individual liner shipping service routes, with two examples: an Asia-Europe service route consisting of nine ports (**a**) and an Africa-Europe service route consisting of seven ports (**b**). In (**c** and **d**), we show ports of the GLSN using a geographical map and show inter-port connections using a hyperbolic layout obtained by coalescent embedding[57], respectively. The color of nodes corresponds to the traffic capacity of ports measured in Twenty-Foot Equivalent Unit (TEU). The coalescent embedding layout clearly points out that TEU gradient is related with the radial coordinate of the hyperbolic model, therefore ports with larger TEU values are more central in the hyperbolic geometry underlying the GLSN. The coalescent embedding hyperbolic layout locates at centre the nodes that are fundamental for the efficient navigability of a complex network[58]. As such, the observed phenomenon that ports with larger TEU are more central in the hyperbolic layout means that ports with larger TEU are fundamental for the efficient navigability of the GLSN in transporting cargoes traded worldwide. This suggests that ports' traffic capacity measured in TEU is indeed a meaningful indicator to be associated with international trade (as we will show in the rest of the study). Source data are provided as a Source Data file.

port, and the local efficiency of the GLSN is the mean over all ports' local efficiencies. Cost of building up the GLSN is the sum of the cost for individual connections, assumed to be proportional to the physical length (here measured by real nautical distance, sourced from https://www.searates.com/services/distances-time/). We found the GLSN configuration remarkably economic: its global efficiency and local efficiency respectively reach 82.7% (*p*-value < 0.001) and 93.2% (*p*-value < 0.001) of the ideal case of network configuration (i.e., all ports are connected with each other), but its wiring cost only accounts for 1.5% (*p*-value < 0.001) of the ideal case; statistical significance test against a configuration null model is detailed in Supplementary Note 1.

More discussions about the associated meaning of the above properties of the GLSN can be found as Supplementary Note 2.

**Multiscale modularity and hubs diversity**. Modular community structure is one of the most prominent properties of complex networks[32]. There are several general advantages to modular and multiscale modular (or hierarchically modular) network organization, including greater robustness, adaptivity and evolvability of network function[33]. We report the GLSN is self-organized into a multiscale modular structure: modules can be further divided into respective sets of submodules (Fig. 3a–c), except for one small module which covers the geographical area mainly consisting of

Greenland and Iceland. The division of port communities is based on a criterion of modularity maximization[19]. And we adopted an algorithm of fast unfolding communities in large networks[34], to search for an optimal partition that maximizes the modularity (*Q*). This algorithm might miss some small structures, but at large scale can be trusted[35]. The seven upper-module port communities in the GLSN are observed to be spatially compact and to correspond to geographically neighboring regions (Fig. 3d), demonstrating the relevance of the method in this case.

The modular structure of the GLSN seems to reflect the contemporarily parallel trends of regionalization and globalization of international trade and economy[10,36]; the OD matrix in Fig. 3e clearly shows the intra-regional concentration of global seaborne trade flows, meanwhile trade between different regional markets can be seen from the existence of a few inter-module links. In the GLSN long-range links are relatively few and mainly appear as inter-module ones (Supplementary Fig. 2), as are the case of many spatial networks (see Supplementary Note 3 for a brief discussion).

With the division of port communities, we then assigned network roles to individual ports based on the pattern of intra-community and inter-community links. We hypothesized that the role of a node can be determined, to a great extent, by its connectivity pattern[37], and thus defined three different types of hub roles (see "Methods"): provincial hubs, gateway hubs, and

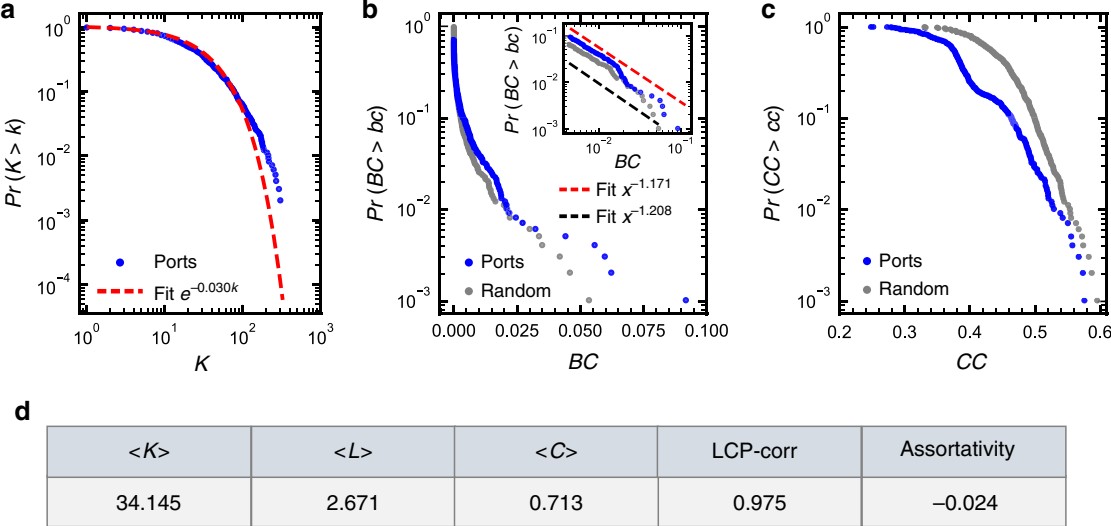

| <K> | <L> | <C> | LCP-corr | Assortativity |
|---|---|---|---|---|
| 34.145 | 2.671 | 0.713 | 0.975 | −0.024 |

**Fig. 2 Basic topological properties of the GLSN.** In (**a**), complementary cumulative distribution function of degree is reported in log-log scale (dots), fitted by an exponential function (dash line) instead of power-law; tests on the power-law distribution of the data failed based on the method of Clauset et al.[59] and the method of Voitalov et al[60] as well. Complementary cumulative probability distributions of betweenness centrality (*BC*) and closeness centrality (*CC*) for ports are plotted in semi-log scale in (**b** and **c**), respectively, in comparison to an equivalent random network which exactly keeps the same degree distribution as the original GLSN. The inset in (**b**) reports in log-log scale a power-law tail (red dash line) in the betweenness distribution with an exponent 1.171, corresponding to ports with $BC \geq 0.0043$ (dots); power-law-ness is tested based on the method of Clauset et al.[59] For an equivalent random network, the betweenness distribution for nodes with $BC \geq 0.0043$ decays with an exponent 1.208 (mean across 80.4% (804 out of 1000) iterations passing the test) (black dash line). The average closeness centrality of the GLSN (i.e., 0.382) is close to that of an equivalent random graph (i.e., 0.440, mean across 1000 iterations). The bottom panel **d** presents the average port degree <K>, average shortest path length <L>, average clustering coefficient <C>, degree assortativity coefficient and local-community-paradigm correlation (LCP-corr). To confirm these basic topological properties of the GLSN, we repeated the analysis by using the liner shipping service routes data of 2017. The results for the GLSN of 2017 are consistent with the present results for the GLSN of 2015 (Supplementary Fig. 1). Source data are provided as a Source Data file.

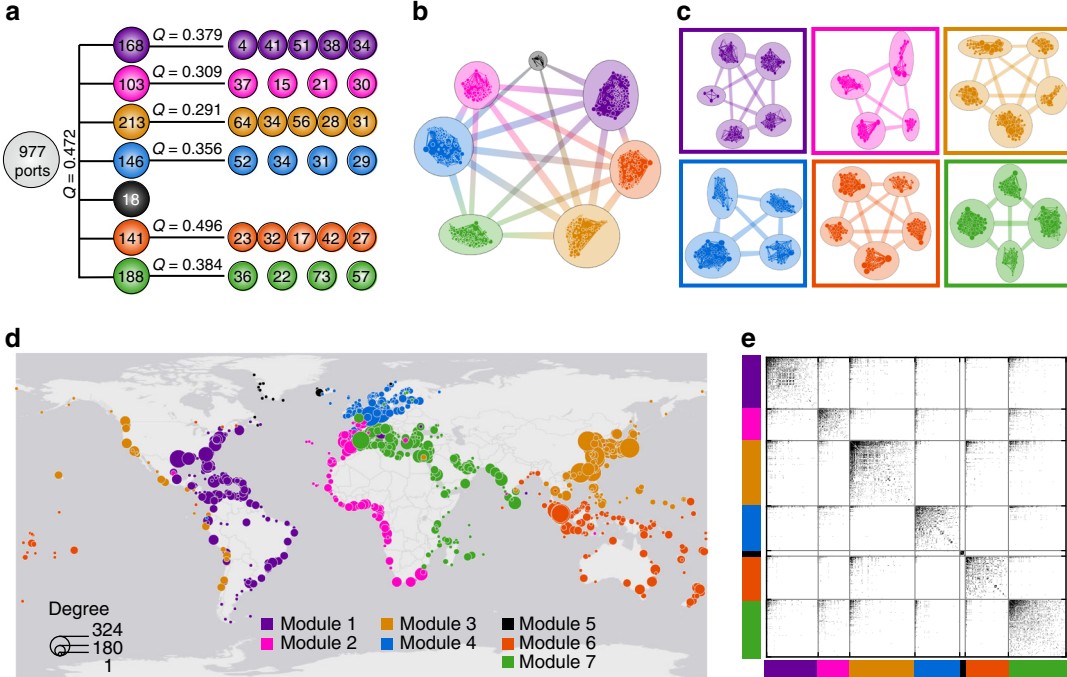

**Fig. 3 Multiscale modular communities in the GLSN.** In (**a–c**), we give results for the division of both modular and submodular port communities. **a** Values of the modularity index (*Q*), together with the size of each module (i.e., number of ports); see "Methods" for a formal definition of *Q*. The network plots show the extracted modules (**b**) and the submodules (**c**); larger separation between models (submodules) is adopted to visualize the weaker connections between them, and inter-module (inter-submodule) connections are simplified. In (**d** and **e**), we show the results for the division of modular port communities. **d** A geographical plot presenting the seven modular communities by color. **e** A matrix plot presenting intra-module and inter-module links, with color black indicating a pair of ports are linked and color white unlinked; in each module ports are sorted in a descending order of degree.

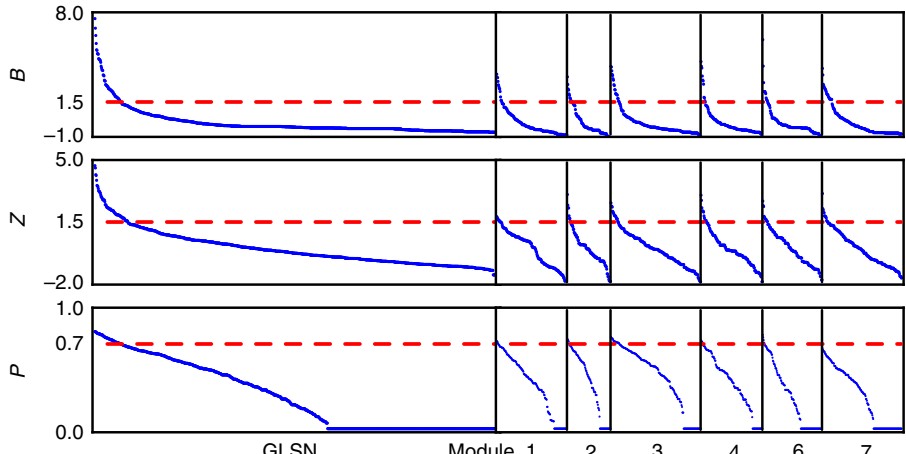

**Fig. 4 Ports' outside-module degree ($B$), inside-module degree ($Z$), and participation coefficient ($P$).** The rank-size distributions of $B$, $Z$, and $P$ of ports are presented in linear plots. The large plots on the left correspond to the results for modular communities in the GLSN, and the small plots on the right the results for submodular communities in individual modules (except for module 5, the smallest one that cannot be further divided into submodules); plots are scaled according to the corresponding number of ports. Dash lines in the plots indicate corresponding threshold values of 1.5, 1.5, and 0.7 used to define gateway hubs, provincial hubs, and connector hubs, respectively.

connector hubs. Provincial hubs refer to ports with inside-module degree ($Z$)[37] at least 1.5 standard deviations above the community means; $Z$ measures how well-connected a port is with others in the module. Gateway hubs refer to ports with outside-module degree ($B$) at least 1.5 standard deviations above the community means; $B$ measures how well-connected a port is with others outside the module. Connector hubs refer to ports with participation coefficient ($P$)[37] at least 0.7; $P$ measures how equally distributed the connections of a port are over the modules. These three indicators help explore whether ports present some connectivity patterns similarities in the structure of the GLSN.

Figure 4 shows the $B$, $Z$, and $P$ values for each port, calculated in contexts of both modular communities and submodular communities. For clarity, in the following analysis we focus on interpreting the results for the modular communities (hereinafter shortened to communities); in a similar way, it should be easy to understand the results for the submodular communities. The fractions of provincial hubs, gateway hubs and connector hubs are 8.3%, 6.6% and 6.6%, respectively. As indicated in Fig. 5, 95.3% of the gateway hubs are also with at least another type of hub role: 29.7% of them are provincial hubs, 32.8% connector hubs, and the rest 32.8% both provincial hubs and connector hubs. Particularly, those ports that simultaneously serve as provincial hubs, gateway hubs and connector hubs concentrate greatly in the world's major trading regions of East Asia, Northwest Europe, North America and Europe Mediterranean. Indeed, port development is mainly related with the development of regional economy and international trade[38]. By contrast, 49.4% of the provincial hubs and 32.8% of the connector hubs turned out to be without any other type of hub roles (Fig. 5).

**Gateway-hub structural core**. To quantify the possibly existent phenomenon of structural-core organization, a structural core of the GLSN is defined as a set of hub ports that meet the following two criteria: first, this set should consist of the largest number of the most important hub ports that form a subgraph of high density (i.e., the proportion of actual links in the maximum possible number of links); second, this set should contain at least one hub port from each modular community in the network. Here, we considered a density threshold of 0.8 (which is heuristically a sufficiently high density) and the three types of hub

ports defined above (i.e., provincial, gateway, and connector hubs). See Supplementary Note 4 for a discussion of the rationale behind the structural core definition.

We detected such a structural core consisting of 37 gateway hubs (Fig. 6a–b), but not detected any structural core based on either provincial hubs (Fig. 6c–d) or connector hubs (Fig. 6e–f). The detected gateway-hub structural core of the real GLSN is statistically significant ($p$-value < 0.001): the probability to detect the same structural core in a null configuration of the GLSN is lower than 0.001 (Supplementary Fig. 3). The detection of the gateway-hub structural core (along with the related findings) is insensitive to the variance of Louvain algorithm performance across 1000 runs (Supplementary Note 5).

The GLSN and its detected structural core are also shown in the hyperbolic space (Fig. 7); the detected structural core is indeed at the center not only of the network topology but also of the hidden network geometry. Most of the structural-core ports are global hubs from major economies, e.g., European ports of Antwerp, Hamburg and Rotterdam, Asian ports of Shanghai, Busan, Singapore, Ningbo-Zhoushan, Hong Kong and Shenzhen, and North American ports of Houston and Savannah. It is worth mentioning that, the structural core also includes a few gateway-hub ports which are relatively small at the global level but are of fundamental importance in integrating their own regions into the global markets. One extreme example is the port of Reykjavik, which serves as the main gateway for transporting cargo to and from the small remote port community that mainly corresponds to the geographical area of Iceland and Greenland. Interestingly, within the contexts of respective modules, there also exist such structural cores consisting of a few submodular gateway hubs (Supplementary Note 6).

Intuitively, a valid structural core of a complex network should be topologically central in the entire network. Therefore, we first investigated the topological centrality of individual structural-core ports based on three basic node centrality measures (i.e., degree, betweenness and closeness), finding that all the 37 structural-core ports rank higher than the top 10th percentile by each measure (Supplementary Table 1). Then, topological centrality of the identified structural core, as a whole, is measured by the percentage of shortest paths between any two non-core ports in the GLSN that pass through the core by node and by link, respectively. This measure was inspired by geodesic node

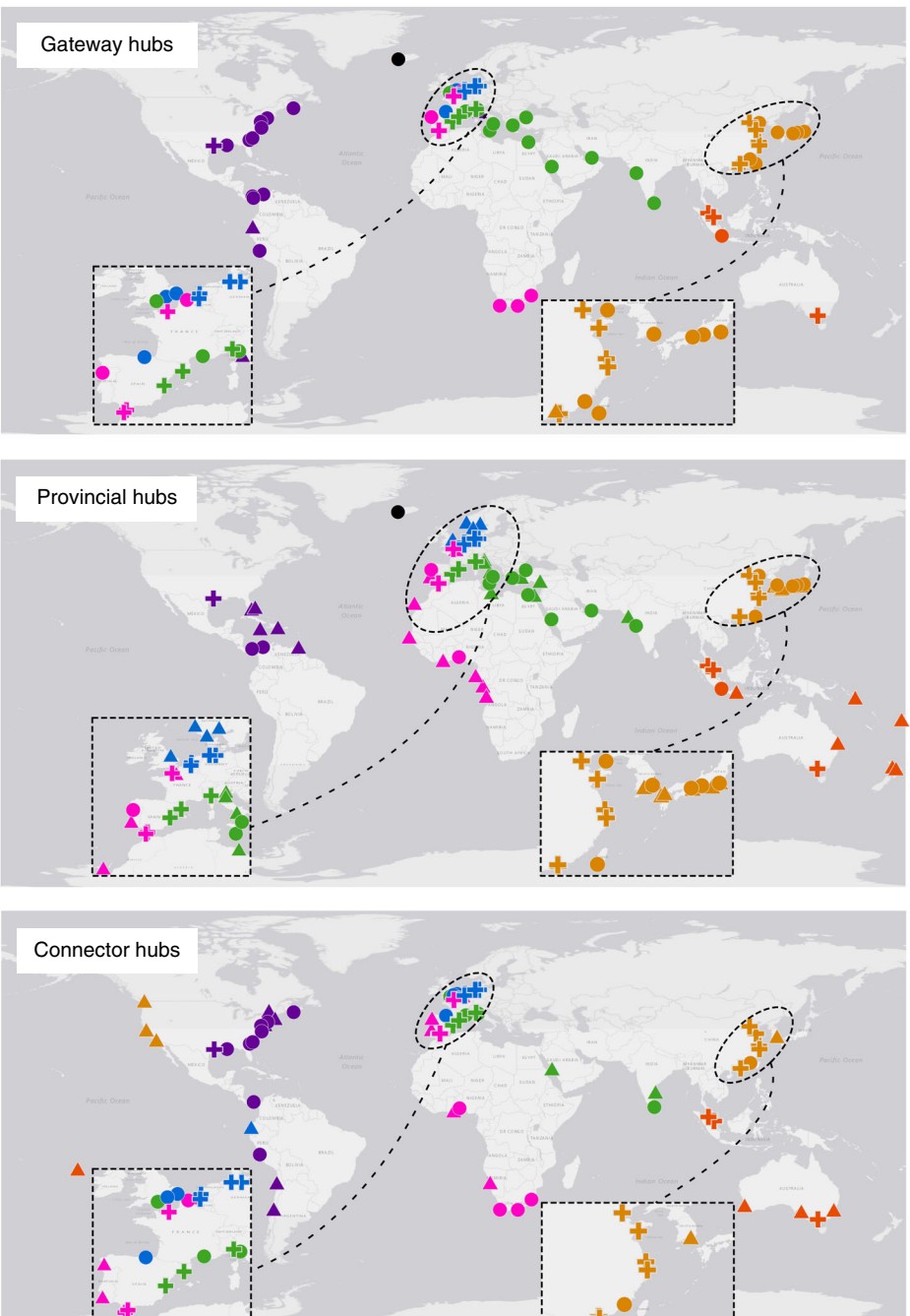

**Fig. 5 Geographical distribution of hub ports in the GLSN.** Here the three plots show the gateway-hub ports (i.e., ports with $B \geq 1.5$), provincial-hub ports (i.e., ports with $Z \geq 1.5$), and connector-hub ports (i.e., ports with $P \geq 0.7$), respectively. Ports are colored according to modular communities. In each plot, triangles denote ports which have only that particular type of hub role under investigation; circles, ports which have one additional type of hub role; crosses, ports which have all the three types of hub roles. Outside-module degrees ($B$), inside-module degrees ($Z$), and participation coefficients ($P$) of the hub ports are reported in the source data. Source data are provided as a Source Data file.

betweenness centrality: for transportation networks, core network components (i.e., nodes or edges) are used relatively frequently, which can be quantified by a betweenness centrality or a similar diagnostic, as compared with other components in the network[39]. We report that, the percentage of the number of shortest paths that pass through at least one of the core ports, normalized to the number of all shortest paths among the non-core ports, reaches as high as 84.22% ($p$-value < 0.001); and the percentage of the number of shortest paths that pass through at least one core connection is 24.65% ($p$-value < 0.001). The statistical significance

tests are as follows. We randomly selected 1000 sets of ports corresponding in size to the number of ports contained in the structural core and traced all shortest paths between ports outside this set, getting the respective percentage values for each set by node and by link; then, we computed a one-side $p$-value as the percentage of random situations greater than or equal to the empirical case. In random situations, the maximum, minimum, and mean percentage values by node are 34.95%, 1.33%, and 9.35% (SD = 4.57%), respectively; and that by link are 2.32%, 0.00%, and 0.21% (SD = 0.27%), respectively.

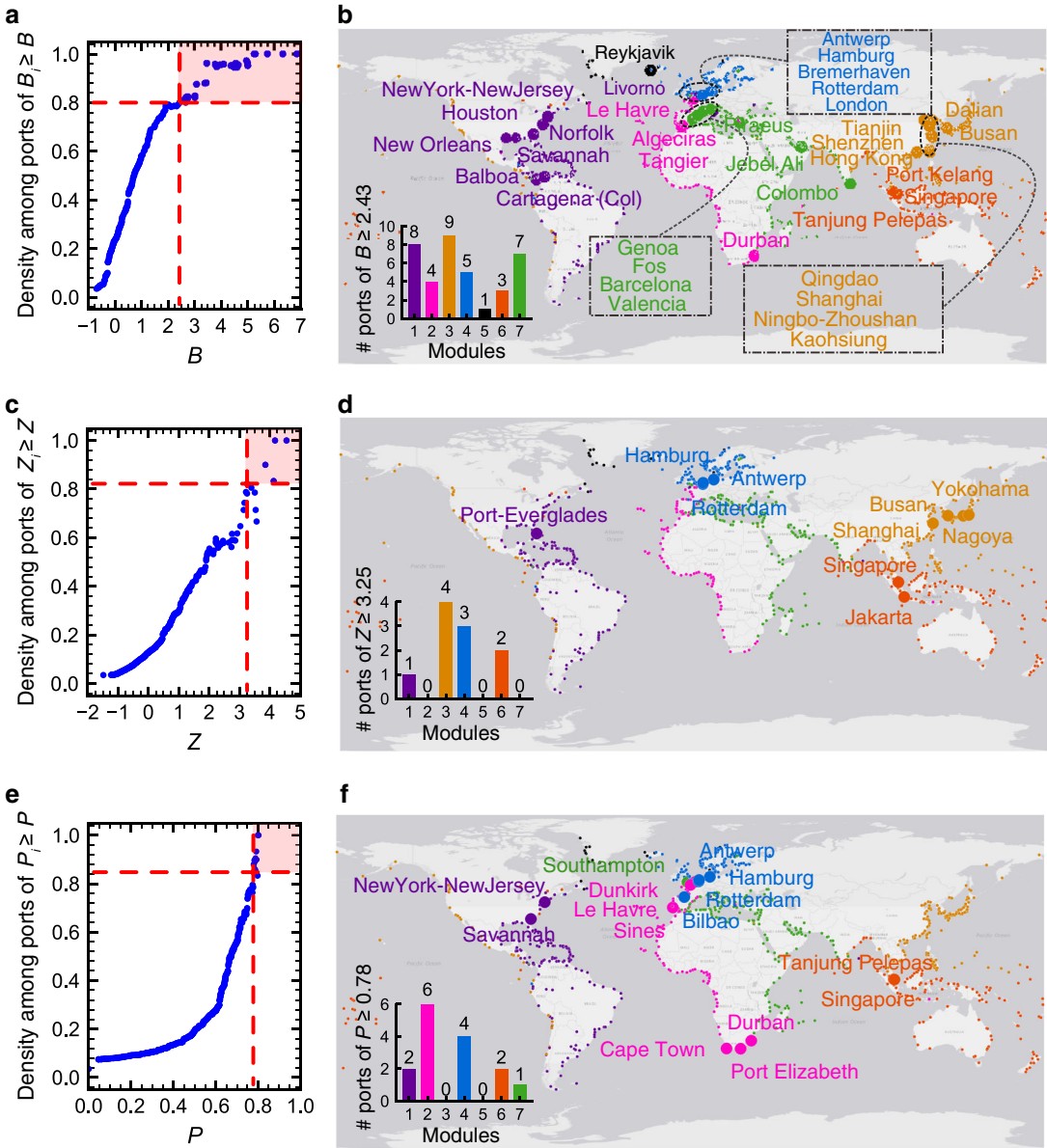

**Fig. 6 Structural core detection of the GLSN.** A structural core of the GLSN is defined as a set of hub ports that meet two criteria: first, this set should consist of the largest number of the most important hub ports that form a subgraph of high density (i.e., here at least 0.8); second, this set should contain at least one hub port from each module in the network. In implementation, we first calculated the connection densities among ports with largest values of $B$, $Z$, and $P$, as presented in plots **a**, **c**, and **e**, respectively. Red regions in the parameter spaces of these three plots correspond to the three respective sets of hub ports that meet the first criterion: ports of $B \geq 2.43$, forming a subgraph of density 0.80; ports of $Z \geq 3.25$, forming a subgraph of density 0.82; and ports of $P \geq 0.78$, forming a subgraph of density 0.85. These three sets of hub ports are further displayed by big dots in geographical plots **b**, **d** and **f**, as well as their respective distributions over different modules (insets). Then, we could evaluate whether any of the three sets meet the second criterion. It turned out that only the set of ports of $B \geq 2.43$ met this criterion and thus constituted a structural core of the GLSN, while the other two sets did not. Modules are indicated by color. Pseudocode of the algorithm for structural core detection is available in Supplementary Note 7.

As ports of the GLSN were divided into structural-core and non-structural-core ports, edges were naturally categorized into three topological types (Fig. 8a): core connections linking structural-core ports, feeder connections linking structural-core ports and non-structural-core ports, and local connections linking non-structural-core ports. Statistical analysis revealed the importance of core connections in supporting long-distance maritime transportation. First, core connections tend to be longer than feeder and local connections (Fig. 8b). Measured by real nautical distance, the average length of core connections (average = 10233 km, SD = 6567 km) is 2.0 times of the average over all inter-port connections (average = 5116 km, SD = 5482 km); feeder connections (average = 7612 km,

SD = 6059 km), 1.5 times; local connections (average = 3585 km, SD = 4379 km), 0.7 times. When looking at the shipping distance for cargo transportation among all the non-core ports in the GLSN (Fig. 8c), we found 16.7% of the total shipping distance—measured by the distance along the edges of their shortest paths—is taken up by core connections. As core connections only account for 3.2% of the total number of inter-port connections in the GLSN, it makes a ratio of distance fraction to the connection fraction to be 5.2, relative to 1.7 and 0.4 for feeder connections and local connections, respectively. When considering those shortest paths that pass through the structural core (i.e., traveling across at least one core connection), the proportion of shipping distance taken up by core

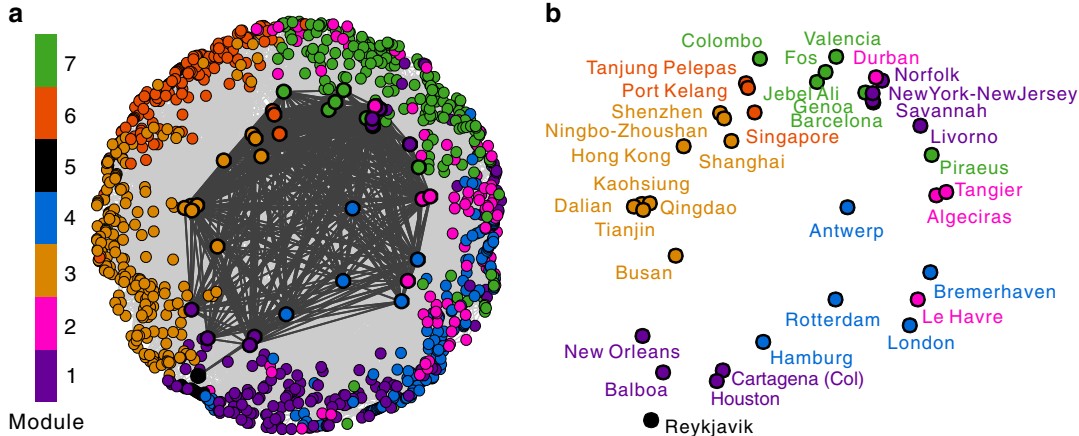

**Fig. 7 Representation of the GLSN and its structural core in the hyperbolic space.** The color of the nodes corresponds to the modular communities. In (**a**), the nodes belonging to the structural core are highlighted with a thicker black border and the intra-core connections are marked in dark gray, whereas the other connections are in light gray. Names of the structural-core ports are indicated in (**b**). The detected gateway-hub structural-core is at the center of the hyperbolic layout. Nodes at the center of this layout are crucial to support the efficient navigability of a complex network[58]. From a previous analysis we know that these nodes at the center have also larger TEU (as presented in Fig. 1d). This indicates that the gateway-hub structural core is indeed mainly composed by ports with larger TEU that are fundamental for efficient navigability of the network in transporting cargoes traded worldwide. Therefore, structural-core ports are important candidates to be associated with international trade (as we will analyse in the next section). In order to be sure that the hyperbolic representation is meaningful and the community separation is significantly and properly represented in the hyperbolic layout, we computed an index of angular separation of the communities (ASI)[61] in respect to the worst scenario in which the nodes of each community are equidistantly distributed over the circumference. This index is in the range [0,1]: 0 indicates the worst case, and 1 indicates a case of perfect angular separation. For the provided embedding the ASI is 0.7, which represents a good angular separation of the communities in the embedding space and is statistically significant with a *p*-value < 0.001 (see Supplementary Fig. 4 for details on the statistical test).

connections reaches 62.0%, and that by feeder and local connections 33.4% and 4.6%, respectively.

**Structural embeddedness and economic performance of ports.** A central goal of network theorizing is to connect network properties with outcome of integrated systems[7,8,40]. Particularly, one fundamental hypothesis of structure-function relations holds that, the functional outcome of a node is at least partly determined by how it is embedded or positioned in the network structure[9,41]. The structural hole theory of Burt[42] posits that, a network player (i.e., a node) embedding within a sparse network of disconnected contacts is in an advantageous network position and gains better economic outcomes.

As such, we are interested to test the relatedness between ports' economic performance and various patterns of structural embeddedness. Here we measured the economic performance of individual ports by the traffic capacity in Twenty-foot Equivalent Unit (hereinafter shortened to capacity) deployed by world liner shipping carriers, which are ground-truth data derived from the database. The various patterns of structural embeddedness considered here are quantified by the following network indicators. Degree (i.e., no. of connections) describes a node's centrality in the simplest way based on local information. Hence, the three topological indicators of outside-module degree, inside-module degree and participation coefficient quantify three respective patterns of local embeddedness, which we term gateway-ness, provincial-ness, and connector-ness. Betweenness, which measures a node's access to the structural holes in the entire network, quantifies the extent to which a node is embedded into the global structure of the network.

We found all indicators showed significantly positive correlations with port capacity and their correlations held well when we controlled for port community (Table 1). The finding that degree performs better than (or at least as good as) betweenness is interesting, indicating that local embeddedness may be an important factor for port economic performance. Different from

nodes in many other real-life complex networks where more connections do not necessarily mean better outcome[40], in the case of container port development within the context of the GLSN, it seems that the more connections a port has the more traffic capacity it will get.

Then we compared the three specific patterns of local embeddedness, finding the port capacity most strongly correlated with the gateway-ness, moderately correlated with the provincial-ness, and weakly correlated with the connector-ness. Such results suggest that, for the development of an individual port, having connections with ports outside its own modular community would help it attract more traffic capacity from liner shipping companies than having connections within its own modular community, regardless of how the outside-module connections are distributed among different modular communities.

We further characterized the gateway-ness by comparing it to the rich-club coefficient, which is a degree-based topological indicator of the rich-club effect in a network (i.e., a tendency for high-degree nodes to be more densely connected among themselves than nodes of a lower degree). The rich-club coefficient quantifies a special pattern of local embeddedness. Specifically, we adopted the unnormalized version[43] and two normalized versions[44,45] to calculate ports' rich-club coefficients in the GLSN (Supplementary Fig. 6). It turned out that the Pearson correlation coefficient between the rich-club coefficient— should it be normalized or not—and port capacity was significantly lower than that between the gateway-ness and port capacity (Table 1). And the structural core is not the same as any possible rich club of the GLSN (Supplementary Fig. 7).

**Structural core and international trade.** Finally, we examined the extent to which the detected structural core of the GLSN is associated with this network's functional outcomes at the system level: the international trade statuses of countries. Specifically, we consider two GLSN topological indicators for countries: the liner shipping connectivity of a country, defined as the number of all

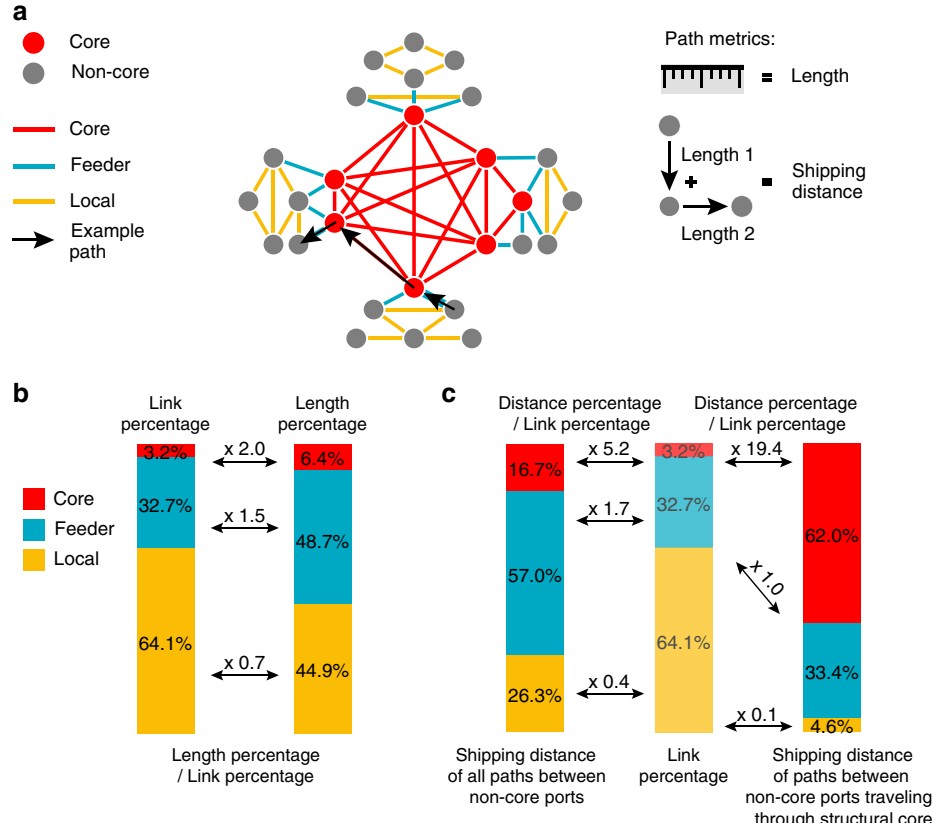

**Fig. 8 Statistics of the core, feeder and local connections. a** Schematic illustration of path metrics; the geographical length of an inter-port connection is measured as the real nautical distance (https://www.searates.com/services/distances-time/) between the two ports, and shipping distance of any port pair is the sum of geographical length of edges along the shortest path. **b** Ratios of length percentage to link percentage for core connections, feeder connections and local connections, respectively. **c** Percentages of core connections, feeder connections and local connections in the total shipping distance of all shortest paths between non-core ports (left bar), and of shortest paths between non-core ports which travel through the structural core (right bar). To better understand those three types of connections related with the structural-core organization of the GLSN, one is encouraged to refer to the hub-and-spoke service network configuration (Hu and Zhu)[13], which is widely adopted by world liner shipping carriers in practice. The hub-and-spoke configuration is illustrated in the Supplementary Fig. 5. In addition, we estimated the physical length of an inter-port connection as the great-circle distance based on ports' geographical locations of latitude and longitude, and then repeated the analysis. We found all the results reported here remain almost invariant (Supplementary Note 8).

**Table 1 Pearson correlation coefficients between network indicators and port capacity.**

| Network indicators | GLSN | Port communities | | | | | | |
|---|---|---|---|---|---|---|---|---|
| | | C1 | C2 | C3 | C4 | C5 | C6 | C7 |
| $B$ | 0.76** | **0.84** | **0.91** | **0.91** | **0.92** | 0.80** | **0.92** | **0.87** |
| $Z$ | 0.50** | 0.49** | 0.52** | 0.59** | 0.64** | 0.60* | 0.55** | 0.70** |
| $P$ | 0.38** | 0.58** | 0.46** | 0.52** | 0.54** | 0.16 | 0.34** | 0.56** |
| $K$ | **0.77** | 0.78** | 0.84** | 0.81** | 0.90** | 0.99** | 0.86** | 0.84** |
| $BC$ | 0.68** | 0.56** | 0.84** | 0.83** | 0.91** | 0.89** | 0.89** | 0.77** |
| $\varphi$ | 0.62** | 0.79** | 0.80** | 0.66** | 0.69** | **1.00** | 0.71** | 0.83** |
| $\rho_C$ | 0.26** | 0.65** | 0.58** | 0.14 | 0.28* | 0.99** | 0.25* | 0.62** |
| $\rho_{CM}$ | 0.58** | 0.79** | 0.81** | 0.59** | 0.66** | 0.99** | 0.67** | 0.84** |

Note: Network indicators $B$, $Z$, $P$, $K$, and $BC$ denote gateway-ness, provincial-ness, connector-ness, degree, and betweenness centrality, respectively; $\varphi$, the unnormalized rich-club coefficient originally proposed in reference[43]; $\rho_C$ and $\rho_{CM}$, two normalized versions of the rich-club coefficient proposed in reference[44] and in reference[45], respectively. Pearson correlation coefficients between network indicators and port capacity are calculated for all ports in the GLSN, as well as separately for ports in individual communities (i.e., C1, C2, C3, C4, C5, C6, C7). C5 is the smallest module that consists of a few ports with similar degree and capacity (and cannot be further divided into submodules).
**$p$-value < 0.001, *$p$-value < 0.01.

connections between ports of this country and ports of all other countries in the world; and the bilateral liner shipping connectivity of a country pair, defined as the number of all inter-port connections between the two countries. And we consider two international trade indicators for countries: the ITV of a country and the BTV of a country pair, sourced from the UN Comtrade database (https://comtrade.un.org/data/). Note that maritime countries altogether account for about 93% of international trade in terms of value. Results show that the two trade indicators are significantly and highly correlated with the respective

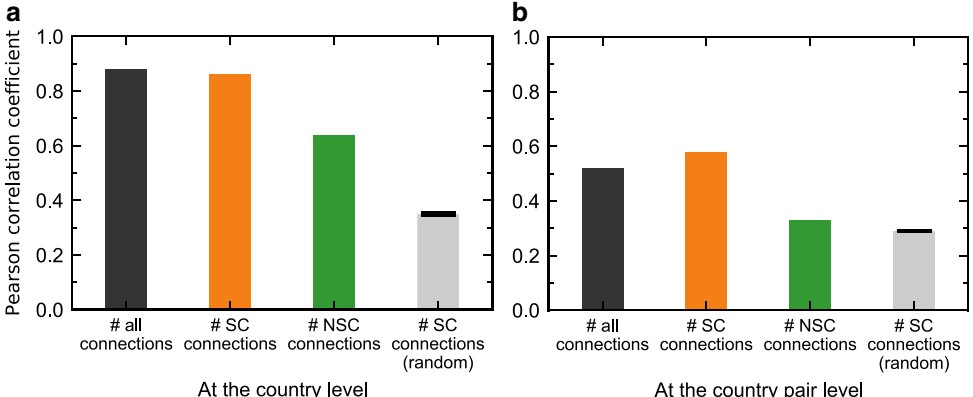

**Fig. 9 Correlation between the GLSN topological indicators and international trade indicators of countries.** In (**a**), we show for countries the Pearson correlation coefficients between international trade value (ITV) and # all (inter-port) connections with all other countries in the world (black bar); between ITV and # SC connections (structural-core connections, those between a country's structural-core ports and ports of other countries), yellow bar; and between ITV and # NSC connections (non-structural-core connections, those between a country's non-structural-core ports and ports of other countries), green bar. In (**b**), we show for country pairs the Pearson correlation coefficients between the bilateral trade value (BTV) and # all (inter-port) connections between the two countries (black bar); between BTV and # SC connections (structural-core connections, those with at least one end-node of the connection being a structural-core port), yellow bar; and between BTV and # NSC connections (non-structural-core connections, those with two end-nodes of the connection being both non-structural-core ports), green bar. We test the statistical significance of the results reported for SC connections (yellow bars), by randomly selecting 1000 sets of ports corresponding in size to the number of ports contained in the structural core (which is 37) and repeating for each random set of ports the same analysis as we did for the detected structural core. Gray bars show the averages of 1000 random cases, and error bars report the standard errors. Source data are provided as a Source Data file.

topological indicators, with the Pearson correlation coefficient between the ITV and the liner shipping connectivity being 0.88 (Fig. 9a, black bar) and that between the BTV and the bilateral liner shipping connectivity being 0.52 (Fig. 9b, black bar).

To further investigate the association between the structural-core ports and countries' international trade statuses, we then recomputed: the liner shipping connectivity of a country, considering for each country only the interactions that its structural-core ports have with ports of all other countries; the bilateral liner shipping connectivity of a country pair, considering only those connections involving structural-core ports (i.e., at least one end-node of a connection is a structural-core port). For comparison, we also recomputed: the liner shipping connectivity of a country, considering for each country only the interactions that its non-structural-core ports have with ports of all other countries; the bilateral liner shipping connectivity of a country pair, considering only those connections involving non-structural-core ports (i.e., two end-nodes of a connection are both non-structural-core ports).

We report that, for individual countries, the Pearson correlation coefficient between the number of structural-core connections and the ITV arrives at 0.86 ($p$-value < 0.001) (Fig. 9a, yellow bar), almost equivalent to that between the number of all connections (which defines the liner shipping connectivity of a country) and the ITV (Fig. 9a, black bar). By contrast, the Pearson correlation coefficient between the number of non-structural-core connections and the ITV is only 0.64 ($p$-value < 0.001) (Fig. 9a, green bar). For country pairs, the Pearson correlation coefficient between the number of structural-core connections and the BTV reaches 0.58 ($p$-value < 0.001) (Fig. 9b, yellow bar), even slightly higher than that between the number of all connections (which defines the bilateral liner shipping connectivity of a country pair) and the BTV (Fig. 9b, black bar). By contrast, the Pearson correlation coefficient between the number of non-structural-core connections and the BTV is only 0.33 ($p$-value < 0.001) (Fig. 9b, green bar). These results suggest that the detected structural core of the GLSN is indeed of great relevance to international trade, because its topological indicators

of countries' connectivity offer a performance of correlation with countries' international trade indicators that is comparable with the performance the entire network can offer and is significantly better than the performance offered by either the non-structural-core ports (Fig. 9a–b, green bar) or the randomly selected structural cores (Fig. 9a–b, gray bar).

## Discussion

The analysis of the gateway-hub structural core provides quantitative findings on the structural organization complexity of the GLSN and its association with international trade, which remain robust when using another dataset on world's liner shipping service routes in the year 2017 (Supplementary Note 9).

From the perspective of maritime transportation sector, the finding that a few ports with largest gateway-ness form a cohesive structural core and support long-distance maritime transportation worldwide has the following practical implications. First, a good knowledge of the gateway-hub structure-core of the GLSN may help liner shipping companies better select the hub ports' locations[17,18], in the sense that it helps understand how the selected hub nodes can serve as transshipment and switching points to improve the overall efficiency and economy of flow transportation in the entire network[16]. Indeed, the hub-and-spoke network configuration has been widely adopted in the design of liner shipping service networks and also many other transportation systems (e.g., air and railway transportations), and one of the most crucial issues in designing such network configuration is to strategically select hub nodes. The gateway-hub structural-core organization discovered in the GLSN therefore provides new insights into understanding and designing hub-and-spoke networks for various transportation systems.

Second, well-positioned ports enabled by frequent and regular liner shipping services are key to countries' access to global markets. A realistic challenge that faces port authorities and national governments is to effectively monitor how well their ports are positioned in the evolving GLSN, relative to any other ports of interest. We suggest that ports with largest gateway-ness values

form a structural core that is positioned at the center of the GLSN (Fig. 7), and thus the proposed measures of gateway-ness and gateway-hub structural core can have policy relevance to quantitative monitoring ports' GLSN positions. We emphasize how the insight brought by the proposed measures distinguishes from that offered by the traditional measure of container throughput, in monitoring the relative competitive positions of two competing ports. Take, for example, the case of ports of Hong Kong and Guangzhou (which are two geographically adjacent world-class ports in fierce competition): Guangzhou port has surpassed Hong Kong port in terms of container throughput; Hong Kong, however, remains more competitively positioned within the GLSN, in the sense that it is still a part of the structural core of the GLSN whereas Guangzhou has not yet become (Fig. 7).

From the point of view of network analysis, our study offers new tools to investigate the interface between network structure and functional outcomes[6,37,41,46,47] in real-world networked systems. We provide novel evidences supporting a pivotal theoretical notion of network cartography: networks are formed by nodes with network-specific roles. The pioneer scheme of network node-role assignment[37,46] classifies nodes for modular networks into topological roles based on two statistics (i.e., inside-module degree and participation coefficient), and has witnessed a great success in its application to investigating various networks (e.g., protein-protein interaction networks[48] and brain networks[49]). Unlike these networks, however, here we discover that the GLSN functionality significantly depends on ports with high outside-module degrees, which we term gateway hubs. Our work suggests that network-specific node-roles could vary across different types of networks. As we have illustrated, the gateway-hub role with its formal definition provides one example that is worth further attention.

The topological property termed gateway-hub structural core helps better understand how the segregated modules in complex modular networks are integrated as a whole via a few hub nodes. Segregation and integration in networks with modular structure has been considered fundamentally important for understanding how such complex networked systems fulfill their functions[20,21]. Integration processes in networks can be viewed from at least two different perspectives, one based on the efficiency of global communication and another on the ability of the network to integrate distributed information. The integration of a network largely relies on network hubs, i.e., a few nodes that are highly mutually connected and highly central in the network. Then one crucial starting step to address such integration processes is to identify special classes of network hubs, which can be defined on a number of criteria. As shown here, in the GLSN such hub nodes form a gateway-hub structural core that proves to be greatly important in the integration of the GLSN and highly associated to the network's functional outcomes.

The underlying mechanisms that govern the emergence of the gateway-hub structural-core organization in the GLSN cannot be answered in the present study. As a preliminary understanding, this organization seems to be affected by the unique constraints regarding the number of liner shipping routes, the physical geography of the earth, and the economy of liner shipping service network configuration (see Supplementary Note 10); and it is not the same as small-world scaling (Supplementary Note 11). Nevertheless, we emphasize that the topological indicator termed gateway-ness, which is for the first time proposed in the present study, adds a new tool in investigating complex modular networks. For the GLSN, the analysis based on gateway-ness leads to empirical findings on the structural-core organization of the network.

One limitation of existing studies of GLSN that are based on liner shipping service routes data at the global level, including the

present study, is the lack of longitudinal analyses of the GLSN structure due to the current limited open access availability of such longitudinal data. It would be worthwhile investigating how the structural organization of the GLSN had evolved over time and how such evolution interplays with the development of international trade. For future studies, analyses that include actual seaborne trade volume could help assess the extent to which gateway-hub ports show unique properties in freight transportation volume between local connections and core connections. This would enhance our understanding of how liner shipping patterns in practice correlate with the connectivity structure of the GLSN. Furthermore, additional research will be required to understand the impact of the GLSN structure on international trade; in particular, establishing the causal influence mechanisms underlying the observed correspondence between the GLSN's structural-core organization and international trade may require additional longitudinal network and economic data and methods on causal inference.

## Methods

**Data on liner shipping service routes.** In this study, the GLSN was derived from service routes data of world shipping companies in the year 2015. We collected the required data from a leading database in the liner shipping industry, Alphaliner (https://www.alphaliner.com/). The data in total includes 1622 liner shipping service routes with detailed information about port rotation of each service route. Port rotation of a liner shipping service route refers to the list of ports that a container ship consecutively calls at during the voyage from the port of origin to the port of destination. The data avoids taking account of any other port-calling activities unrelated with cargo loading and unloading, and thus it guarantees a high-level relevance to world seaborne trade. By aggregating the service route data of at least world's top 100 liner shipping companies (altogether accounting for more than 92% of the world's total liner shipping capacity), the Alphaliner database also provides the information on the traffic capacity (measured in Twenty-Foot Equivalent Unit, TEU) deployed on each existing liner shipping service route in the world, and thus it offers us a good opportunity to define an inter-port network at the global level along with the valuable information on individual ports' traffic capacities. Note that the traffic capacity of a port (abbreviated as port capacity), defined as the total capacity of all the individual liner shipping service routes that the port is involved in, is an important indicator evaluating port economic performance in global liner shipping markets.

Previous studies had used two types of shipping data to construct liner shipping networks: container vessels' movement trajectory data[10,11] and liner shipping service routes data[12,13]. The two types of data are by nature very different: the former are container vessels' voyage data logs, which can be known only after voyages are completed; the latter are service schedule data, which are regular and are known much earlier than making voyages. Both types of data have merits and are useful for analyzing different research topics. However, regarding the particular topic that is (for the first time) investigated in the present study (i.e., the association between the structure of the GLSN and international trade), liner shipping service routes data are more suitable than container vessels' movement trajectory data, due to the following facts of liner shipping. There in fact exists a unique property of liner shipping, which does not exist in any other maritime shipping mode: shipping companies always pre-fix liner shipping service routes, including the end-point ports and multiple other ports between them and the vessels deployed; vessels go back and forth on such pre-fixed service routes. The information of this unique property is precisely contained in liner shipping service routes data but is lost in container vessels' movement trajectory data; regarding the trajectory data, one cannot know the two end-point ports of a service route and thus loses the information on the unique property of liner shipping. Indeed, the unique property of pre-fixed routes makes liner shipping service essentially different from other modes of maritime shipping services, in the sense that it does not just simply serve the demand of international trade but also could potentially influence the demand of international trade between countries. As the economic theory of induced traffic demand[50] posits, increased transportation capacity can stimulate extra traffic demand. Note that, in industry practice, shipping companies always pre-release their liner shipping service routes, which in many cases could be even one year prior to making voyages.

**Network topology representation method for GLSN construction.** To further illustrate the adopted method of GLSN construction, it is worth mentioning the so-called concept of space L and P[51,52] that has been used in many transport network studies to properly represent the topology of transport networks. In space L, a link between two nodes exists if they are consecutive stops on a same route; in space P, a link between two nodes means that there is a single route connecting them. Consequently, the node degree in space-L topology is just the number of directions one can take from a given node, and the node degree in space-P topology indicates

the total number of nodes reachable to a given node by using a single route[51]. Both of the two methods of network topology representation have been adopted in maritime transportation network research[10,13,53], and the methodological choices depend on specific research focuses. The present study emphasizes the fact that cargo between any two ports on a same service route can be transported via a single ship, which is indeed important information contained in liner shipping service routes data. Such information will be precisely kept in a network representation of space P but will be lost in that of space L. Therefore, a GLSN topology was constructed here in space P, consistent with that in reference[53].

**Network assortativity.** Ports with high degree may connect randomly or they may connect preferentially to one another. We examined the degree correlation of inter-port connections in the GLSN by computing the assortativity, $r$, which is defined as the Pearson correlation coefficient of the degrees at endpoints of an edge[28]. It is calculated as follows:

$$r = \frac{\langle k_i k_j \rangle - \langle k_i \rangle \langle k_j \rangle}{\sqrt{\left(\langle k_i^2 \rangle - \langle k_i \rangle^2\right) \times \left(\langle k_j^2 \rangle - \langle k_j \rangle^2\right)}} \quad (1)$$

where $k_i$ and $k_j$ are degrees of the nodes at either end of a link and $<>$ represents the average over all links. It varies between $-1$ and $1$: For $r > 0$ the network is assortative, for $r < 0$ the network is disassortative, and for $r = 0$ the network is neutral. If a network's assortativity is positive, hub nodes tend to be connected with other hubs, and vice versa.

**Small-world-ness evaluation.** The small-world-ness[30,31] was proposed for the characterization of a given network as small-world, meaning that it exhibits a low average shortest path length and a high average clustering coefficient[7]. Shortest path length is the minimum number of edges that must be traversed to go from one port to another, and clustering coefficient quantifies the number of connections that exist between the nearest neighbors of a port as a proportion of the maximum number of possible connections[7]. It relies on comparing a given network with an equivalent random network and lattice network on the basis of the average clustering coefficient (which is a local measure) and the average shortest path length (which is a global measure). A coefficient called $\sigma$ for characterizing small-world networks was introduced by Humphries et al.[30]. To calculate this measure, the average clustering coefficient $C$ and the average shortest path length $L$ of the network are compared with $C_{rand}$ and $L_{rand}$ of an equivalent random network with the same node degree distribution, obtaining the small-world coefficient:

$$\sigma = \frac{C / C_{rand}}{L / L_{rand}} \quad (2)$$

A condition for a network to exhibit small-world-ness is that the average shortest path length should be close to that of an equivalent random network, $L \approx L_{rand}$. Meanwhile, the average clustering coefficient should be close to that of an equivalent lattice network, which also implies that $C$ should be much higher than that of an equivalent random network, $C \gg C_{rand}$. These boundary conditions, if met, restrict the value of $\sigma > 1$ for small-world networks. The problem with this coefficient is that even small variations in the already low value of the average clustering coefficient for random networks, $C_{rand}$, significantly influence the value of the ratio $C / C_{rand}$. To overcome this problem, a new robust measure was introduced by Telesford et al.[31], which is called $\omega$. The average shortest path length $L$ is compared with $L_{rand}$ of an equivalent random network and the average clustering coefficient $C$ is compared with $C_{latt}$ of an equivalent lattice network, obtaining the small-world coefficient:

$$\omega = \frac{L_{rand}}{L} - \frac{C}{C_{latt}} \quad (3)$$

Note that $C_{rand}$ is not considered, therefore this measure neglects its fluctuations. Since the boundary conditions for small-world-ness are $L \approx L_{rand}$ and $C \approx C_{latt}$, the values of $\omega$ are expected to be close to 0 in small-world networks. The equation suggests that the typical range for the coefficient is $\omega \in [-1,1]$, with positive values representing a network closer to a random one ($L \approx L_{rand}$ and $C \ll C_{latt}$), and negative values representing a network closer to a lattice ($L \gg L_{rand}$ and $C \approx C_{latt}$).

According to the test suggested by Humphries et al.[30], the GLSN is small-world with $\sigma = 2.892$. When applying the test proposed by Telesford et al.[31], the GLSN is also found to be small-world with $\omega = 0.009$.

**Economic small-world-ness evaluation.** For real-world systems one would expect the efficiency of the underlying network structure to be higher as the number of edges increases, but the cost for the network construction also rises since there is a price to pay for number and physical length of edges. A network is economic small-world if it has high global and local efficiency and low cost[23]: efficient in information propagation at both global and local levels but nevertheless cheap to build. Based on indicators of global efficiency ($E_{global}$), local efficiency ($E_{local}$) and cost ($C$)—all defined in the range from 0 to 1—the economic small-world property of a network structure can be quantitatively analyzed. For spatially embedded networks, $\in_{ij}$, the efficiency in propagating information between two nodes $i$ and $j$, is assumed to be inversely proportional to the shortest path distance between them (i.e. the

smallest sum of the physical distance throughout all the possible paths, $d_{ij}$); $\in_{ij} = 1/d_{ij}$ $\forall i, j$. The global efficiency of a network structure ($E_{global}$) is defined as the average efficiency of all the node pairs in the network, normalized by that in an ideal case of network configuration where all nodes are connected with each other. It is calculated as follows:

$$E_{global} = \frac{E(G)}{E(G^{ideal})} \quad (4)$$

$$E(G) = \frac{\sum_{i \neq j \in G} \epsilon_{ij}}{N(N-1)} = \frac{1}{N(N-1)} \sum_{i \neq j \in G} \frac{1}{d_{ij}} \quad (5)$$

$$E(G^{ideal}) = \frac{\sum_{i \neq j \in G} \epsilon_{ij}}{N(N-1)} = \frac{1}{N(N-1)} \sum_{i \neq j \in G} \frac{1}{l_{ij}} \quad (6)$$

where $E(G)$ and $E(G^{ideal})$ represent the efficiency of the real network structure and the ideal case, respectively; the latter has all the possible edges among the nodes and supports a highest efficiency in information propagation, as $d_{ij} = l_{ij}$ $\forall i, j$. $l_{ij}$ is the geographical distance between the two nodes.

The local efficiency of a network structure ($E_{local}$) is characterized by evaluating for each individual node $i$ the efficiency of $G_i$, the subgraph consisting of the neighbors of $i$. It is defined as follows:

$$E_{local} = \frac{1}{N} \sum_{i \in G} \frac{E(G_i)}{E(G_i^{ideal})} \quad (7)$$

where $E(G_i)$ is the efficiency of the subgraph of the neighbors of $i$, and $E(G_i^{ideal})$ that of the ideal case for the subgraph where all neighbors of $i$ are mutually connected.

In parallel, cost of building up a network structure ($C$) is the sum of the cost of all individual connections, normalized by the cost of an ideal case of network configuration (i.e., all nodes are connected with each other). The cost for a connection between nodes $i$ and $j$ is assumed to be proportional to the geographical length, $l_{ij}$, which, in the case of the inter-port connection, is the real nautical distance (https://www.searates.com/services/distances-time/). It is calculated as follows:

$$C = \frac{C(G)}{C(G^{ideal})} = \sum_{i \neq j \in G} a_{ij} l_{ij} / \sum_{i \neq j \in G} l_{ij} \quad (8)$$

where $C(G)$ and $C(G^{ideal})$ represent the cost for the real network structure and the ideal case, respectively. $a_{ij}$ equals 1 if nodes $i$ and $j$ are connected, and 0 otherwise.

**Local community paradigm organization.** The local-community-paradigm (LCP) is a general brain-network-inspired theory proposed to justify the process of topological-based link-growth and link-formation both in monopartite complex networks[29] and bipartite complex networks[54]. It was also employed in the domain of neuroplasticity[55] and creation of memory associated with pain by means of network rewiring in the rat's brain. The LCP theory finds also a practical application in one of the most intriguing topics of applied network science: the link prediction problem, which refers to modeling the intrinsic laws that govern network organization and growth[29,54].

According to LCP, several real-world complex networks have a structural organization consisting of many local communities that, similarly to the brain, favor local signaling activity, which in turn promotes the development of new connections within those local communities. This idea was inspired by the famous assumption behind Hebbian learning: neurons that fire together wire together. Cannistraci et al. noticed that the network topology plays a crucial role in isolating cohorts of neurons in functional modules that can naturally and preferentially perform local processing. The reason is that the local-community organization of the network topology creates a physical and structural energy barrier (a topological gap between distinct communities) that confines these neurons to preferentially fire together within a certain community and consequently to create new links within that community. This process implements a type of local topological learning that they have termed epitopological learning, which stems as a general complex network topological interpretation of Hebbian learning, whose definition was only given for neuronal networks. Hence, epitopological learning and the associated LCP have been proposed as local rules of topological learning and organization, which are generally valid for modeling link growth and for topological link prediction in any complex network with LCP architecture. To determine whether a network has a LCP architecture, a procedure was proposed based on an indicator called local-community-paradigm correlation (LCP-corr). The procedure suggests that for a link in a given network topology, the number of the common neighbors of the two end nodes of the link is positively correlated with the number of links among those common neighbors. It is calculated as follows:

$$\text{LCP-corr} = \frac{\text{cov}(\text{CN}, \text{LCL})}{\sigma_{CN} \sigma_{LCL}}, \text{ when CN} > 0 \quad (9)$$

where cov indicates the covariance operator and $\sigma$ the standard deviation. This formula is a Pearson correlation between the CN and LCL variables. CN indicates a one-dimensional array. Its length is equal to the number of links in the network

that have at least one common neighbor, and it reports the number of common neighbors for each of these links. LCL indicates a one-dimensional array of the same size as CN, and it reports the number of local community links between the common neighbors. Mathematically, the value of LCP-corr would be in the interval [−1,1]. However, extensive tests on many artificial and real-world complex networks demonstrate that an inverse correlation between CN and LCL is unlikely, therefore the interval is in general between [0,1]. In particular, it was revealed that LCP networks are the ones with high LCP-corr (>0.7) and are very frequent to occur: they are related to dynamic and heterogeneous systems that are characterised by weak interactions (relatively expensive or relatively strong) that in turn facilitate network evolution and remodeling. These are typical features of social and biological systems, where the LCP architecture facilitates not only the rapid delivery of information across the various network modules, but also the local processing. In contrast, non-LCP networks (with low LCP-corr, i.e., <0.4) are less frequent to occur and characterize steady and homogeneous systems that are assembled through strong (often quite expensive) interactions. An emblematic example of non-LCP networks is offered by the road networks[29], for which the costs of creating additional roads are very high, and in which a community of strongly connected and crowded links resembles an impractical labyrinth.

**Modularity and community structure**. To discover port groups within the GLSN, we expect to determine sets of ports that are strongly connected with each other while less connected to the rest of the network. Under the method of modularity maximization, each partition of a network into several disjoint modules is given a score Q, called the modularity. It is defined as[19]:

$$Q = \frac{1}{2L} \sum_{i,j} \left( A_{i,j} - \frac{k_i k_j}{2L} \right) \delta(i,j) \qquad (10)$$

where $A_{i,j}$ equals 1 if a link exists between node $i$ and node $j$, and 0 otherwise; $L$ is the total number of network links, $k_i$ and $k_j$ are the degrees of $i$ and $j$ respectively; $\delta(i,j)$ equals 1 if $i$ and $j$ belong to a same group, and 0 otherwise. The modularity $Q$ measures the difference between the number of links between node groups in the actual network and the expected number of links between these same groups in an equivalent random network that preserves the same link density as the actual network. Therefore, $Q = 0$ corresponds to a random network where two nodes are linked with probability that is proportional to their respective degrees. This formulation recasts the problem of identifying modules as a problem of finding the so-called optimal partition, i.e., the partition that maximizes the modularity function $Q$. Generally, a modularity value of $Q$ larger than 0.3 is indicative of true community structure in a network[37,56].

In the present work the modularity is implemented by adopting an algorithm of fast unfolding communities in large networks[34]. This algorithm might miss some small structures, but at large scale can be trusted[35].

**Defining various hub ports in the GLSN**. For modular network structure nodes with similar roles in individual modules are expected to present similar connectivity patterns[37]. Based on ports' connectivity patterns, we define ports as: provincial hubs, gateway hubs, connector hubs and non-hubs. First of all, port modules may be structurally organized in different ways: they could be centralized that in each module there exist a few ports widely connected with others, and they could also be decentralized with all ports being equally connected. The inside-module degree of a port $i$ ($Z_i$) measures how well-connected port $i$ is with others in the module, indicating the extent of the provincial-ness of port $i$ in its modular community. The higher inside-module degree, the more significant status of being a provincial hub. It is defined as follows[37]:

$$Z_i = \frac{k_{iT} - k_T}{\sigma_{k_T}} \qquad (11)$$

where $k_{iT}$ is the number of connections of port $i$ to other ports in its own module $T$ (i.e., intra-module connections), $k_T$ and $\sigma_{k_T}$ are respectively the average number and the standard deviation, of intra-module connections over all the ports in module $T$. Here, ports with inside-module degrees at least 1.5 standard deviations above the community mean are regarded as provincial hubs.

Secondly, as ports with similar inside-module connectivity could be different in connecting with the outside world, we introduce the indicator of outside-module degree, for the first time in the present study. The outside-module degree of port $i$ ($B_i$) measures how well-connected of port $i$ is with ports outside its own community, reflecting the extent of gateway-ness of port $i$ in realizing the interaction between its host community and the outside world. It is defined as:

$$B_i = \frac{m_{iT} - m_T}{\sigma_{m_T}} \qquad (12)$$

where $m_{iT}$ is the number of connections of port $i$ to other ports out of its own module $T$ (i.e., out-module connections), $m_T$ and $\sigma_{m_T}$ are respectively the average number and the standard deviation, of out-module connections over all the ports in module $T$. We consider ports with outside-module degrees at least 1.5 standard deviations above the community mean as gateway hubs. This indicator is particularly useful in uncovering those ports that are of great importance in facilitating the integration of their own regions into the whole network structure,

whose roles would otherwise be underestimated if merely evaluated by their degrees which are not large enough to be global hubs.

In addition, ports with similar outside-module connectivity may differ in the distribution of outside-module connections over different communities, if some specialize in one or two communities while the other ports are evenly connected with many communities. The participation coefficient of port $i$ ($P_i$) measures how equally distributed the connections of port $i$ are over different modules in the GLSN, implying the level of seaside market diversification. The more equally a port's connections distribute over the more modules, the more significant role of connector hub of the port in the structure of the GLSN. It is mathematically defined as[37]:

$$P_i = 1 - \sum_{s=1}^{N_M} \left( \frac{k_{is}}{k_i} \right)^2 \qquad (13)$$

where $k_i$ is the total connections (degree) of port $i$, $k_{is}$ is the number of connections of port $i$ to ports in module $S$, and $N_M$ is the number of modules in the GLSN. Thus, the participation coefficient of a port is close to 1 if its connections are evenly distributed over different modules and 0 if all its connections are inside its own module. In the present study, ports with participation coefficients at least 0.7 are considered connector hubs. Note that, by definition, connector hubs here are just different from the so-called transshipment hubs that are frequently discussed in literature of maritime transportation research. Specifically, connector hubs refer to ports whose connections are much equally distributed among different modular communities in the GLSN, emphasizing a high-level of equal participation in different parts of the overall network structure. Transshipment hubs are ports that facilitate cargo transportation between different shipping routes, regardless of whether the transported cargoes concentrate on a few specific destination regions or distribute evenly among many destination regions.

**Reporting summary**. Further information on research design is available in the Nature Research Reporting Summary linked to this article.

## Data availability

The GLSN data and all other data supporting the findings of this study are available in: https://github.com/Network-Maritime-Complexity/Structural-core. This GitHub link contains the source data for Figs. 1d, 2b–c, 5, and 9, Supplementary Figs. 1b–c, 7a, 9, 10, 14, 18, 20, 24–25, 26b, Supplementary Tables 2, 4–10. Note that the source data for Fig. 1d is a dataset on the GLSN topology of the year 2015. Raw data on world liner shipping services were provided by a third-party commercial database (Alphaliner, https://www.alphaliner.com/, one of the world's leading databases in the liner shipping industry) and were used under the license for the current study, and so are not publicly available. The dataset on the GLSN topology of the year 2017 is too recent and is not publicly available for the sake of the business of the database. All data generated during this study are however available from the corresponding authors on reasonable request. Data on the nautical distance between ports are publicly available in: https://www.searates.com/services/distances-time. Data on countries' international trade value and country pairs' bilateral trade value are publicly available in: https://comtrade.un.org/data. Source data are provided with this paper.

## Code availability

All the code used in our study is available in: https://github.com/Network-Maritime-Complexity/Structural-core. The code allows the reproducibility of the quantitative results reported in both the main article and its supplementary information. Source data are provided with this paper.

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

## Acknowledgements
We thank Naoki Masuda for his precious comments on the draft. We also thank the lab members Jia Song and Wen Li for carefully testing all the code and lab engineer Yongsheng Qian for the computation facility support. H.X. acknowledges the support provided through National Natural Science Foundation of China, respectively under Grant Numbers 71871042, 71533001, and 71421001. M.X. acknowledges the Alphaliner database for providing high-quality world liner shipping service routes data, and the support provided through China Postdoctoral Science Foundation under Grant Number 2017M621141. C.V.C. research group was mainly supported by the independent group leader starting grant of the BIOTEC at the Technische Universität Dresden (TUD). H.X. research group is also supported by the Scientific and Technological Innovation Foundation of Dalian, China under Grant Number 2018J11CY009.

## Author contributions
M.X., H.X., and C.V.C. collaboratively conceived and designed the study. M.X. invented the topological measure termed gateway-ness. M.X. wrote the manuscript with C.V.C. important participation, together with inputs, comments and suggestions from the other authors. Q.P. generated the code and performed the computation. A.M. and C.V.C. performed the network analysis in the hyperbolic space and realized the hyperbolic layout figures. All authors collaboratively performed the result analysis. H.X. initialized the overall conception of local segregation and global integration in the GLSN. C.V.C. supervised the network science analysis. H.X. and C.V.C co-directed the study.

## Competing interests
The authors declare no competing interests.
