## [Peer Review File · Nature Communications]

Reviewers' comments:

Reviewer #1 (Remarks to the Author):

General comment

This is a tentative of innovating in the fields of transport geography AND network analysis. Unfortunately and to my opinion, the article mainly failed this attempt but there a number of good aspects that encourage a resubmission with stronger academic and thematic dimensions. Mainly, it lacks a central, proper and clear research question to guide the analysis. This makes the article too much data-driven and lacks of focus. The time evolution would be a plus and a stronger knowledge / literature reading and understanding of the transport geography in top of pure "network analysis" literature would help. See detailed comments below for more explanation of this decision.

Detailed comments

Abstract

- global commodity trade: no, global trade is sufficient here
- two times "trade" is repeated in the first two sentences; consider revising the English by a native speaker
- it has self-organized: over what period of time? when?
- global liner shipping network: not sure that non-transport specialists can follow this term; consider container shipping instead?
- segregation, integration: not clear
- reveal the presence: outcome of the article; better introduce as "this paper proposes..."
- provincial, gateway ...: only one is explained over three; keep it shorter or complement or remove from the abstract or say it differently?
- economic performance: very broad; should be better linked with the rest of the main idea, which is ... still unclear to me; same for "seems to support the world trade"; are we heading towards a tes of how much sea transport influenced world trade and vice-versa?
- structural core: common expression? is that your main question, or one the issues your article wants to check? if yes then we miss a proper and central question, it is too diffused and fuzzy

Introduction

- reached 0.99% instead of reach; are transshipment flows included in UNCTAD data?
- most cost-effective way: why and how?
- 70% or 80% of global trade: sources?
- eco. dev. see Bernhofen et al. for an earlier study without the network dimension but econometrics based on a time-series
- high-capacity: less than bulks
- UNCTAD index: be more critical; what is the value of this index compared to the reality of shipping and to other ones like the World Bank's LPI?
- at THE national level (check English)
- given the port-interface: this is a geographic area, not a research question, so why is the point here? Do you mean port-city relationships? which ones? spatial, economical, else?
- global port cities: why only the global ones??
- witted: witnessed?
- several realistic questions: academic questions?
- self-organizing capacity: too long sentence; consider developing each issue separately
- similarities: of which type?
- socio-economic factors: which ones?
- BY unveiling

- contagion, disease: not so much related with the present network
- "remain barely explored" not sure given the last 2 edited books on maritime networks published by Routledge in 2015 and 2017 and a plethora of related analyses in multiples journal papers using complex networks among other methods; see also Wang et al. 2016 in Nature Communications on the same network studied here ...
- data quality / shipping industry: what is the link?
- a recent ...: why not adopting time series data like in many other researches of the kind to demonstrated a structural change?
- 2015: better justify the use of this date for this particular study
- GLSN: we need a proper mathematical definition of the graph
- structural embeddedness: define
- how to test last paragraph ideas? see DEA studies of port performance indicators in MPM, JTG, TRA, TRE etc.

Results

- see the chapter Liner Service Networks in Container Shipping in the edited Kogan Page book for strengthening your explanation of service types
- seaside transport capacities: awkward expression, consider changing
- complete graph: see the Space-P introduced by Hu and Zhu 2009 (Physica A) and also Deng et al. 2009 for a reference to your same methodology to define the network... This needs you to further discussing the choice of space-P instead of space-L!!
- page 5 line 150: wrong, see Bernhofen et al. 2013 (working paper) as well as Guerrero et al. 2015 (Routledge book)
- subsection "degree distribution...": too much common in network analysis, why not comparing with previous studies instead of restarting from scratch yet with another (but already studied) network?
- not scale-free? see Gastner and Ducrest 2015 (Routledge book) on the fit between statistical models and real-world degree distributions for the global shipping network since 1890 up to 2000...
- concentration in big ports: this is not happening by "nature"; should develop a bit more here or earlier the causes of economies of scale since the mid-1990s in this particular sector and the emergence of hubs etc. Given your study year you should also take into account the advent of slow steaming ...
- figure 2: average BC: useless
- trunk line: explain further for non-specialists
- anomalous centrality in shipping: not as critical as for airlines, to be discussed ...
- unevenly developed: in what sense?
- self-organized: explain
- 0.723: it is 0.713 in the figure; should discuss this high value - because of choosing space-P instead of space-L for instance, see 2012 global networks paper on the GLSN 1996-2006
- node pairs: do you mean nodes?
- cost of building: much composite and hard to define, especially in the context of shipping; the "imaginary routes" depicted by Ross Robinson (1968)
- LAT and LONG bird's fly: why not nautical distances to get closer to reality?

Multiscale modularity

- not sure whether this analysis is well connected to the previous
- length of physical links: yes but those longer links over 10,000km do not exist; they are artificially "made" by the choice of space-P over space-L and so include intermediary stops ... be more critical and comparative
- try to name your 3 port categories in a more generic manner ... "provincial" often refers to subnational ...
- figure 5: absolutely NO geographic logic in this result; even certain regions such as the MEd, the

Caribbean and SE Asia which are hub regions do not even appear in the right types such as "connector" for instance!
- figure 8 should refer to hu and zhu 2009 also drawing a figure on hub-and-spokes structure

Reviewer #2 (Remarks to the Author):

This paper analyzes the network of ports connected by freight ships. It is a relatively rarely studied type of spatial network. The authors adapt some existing methods and ideas to this data to find uncover the organization of the system. I don't think the findings are interesting enough to publish this in Nature Communications. The methods are not advancing the field of network science enough, and the new knowledge about the shipping network does not change much from the pioneering study of Kaluza et al.

There are some unique constraints on this network: 1. The geography of the earth and the fact there are not so many shipping routes in many locations. 2. Unlike the (more well-studied) cargo flight network longer routes are more common, and it is more restricted by geography. 3. The economy is different than the flight network since ships don't primarily go back and forth between the same points. I would recommend this paper for Nature Communications if the author could take unique point like these as the starting point, construct (or choose) analysis methods from this, and discover some unknown fact about the system (and also reproduce known features as proof-of-concepts of their methods.

Now the paper tries to find different types of hubs . . . and finds different types of hubs. Wouldn't they do just the same with a different type of data? They find a structural core, but would it be possible that they wouldn't find a structural core? As I understand it a network without it—so the hubs would not be interconnected—would be quite extreme at least not small world. So then the existence of a structural core is the same as the existence of small-world distance scaling (that was already reported in Kaluza et al.)?

[About modularity] First, the Q modularity is the objective measure that the Louvain algorithm maximizes. You write "modularity" for what is really "maximum modularity". That's OK (always saying "maximum modularity" is awkward too), but at least you should point that out

[About modularity] The maximum modularity is strongly affected by the number of nodes and links in the network, and also the degree sequence. AFAIK, there is no non-pathologically measure to say which of two network that is most modular. Maybe it is even an ill-defined problem.

[About the analysis of betweenness, closeness] This would be much better if you compared to other networks—null models and similar transportation networks.

Reviewer #3 (Remarks to the Author):

This manuscript introduces a new network measure, termed "gateway-ness," which characterizes a nodes outside module/community degree of a node. This is a unique measure in that most measures of network topology focus on within module connectivity to characterize properties like the provincial and non-provincial nodes (i.e., network cartography). The authors provide a systematic analysis of the network properties and details how their measure of gateway-ness is unique compared to other related measures like within-module degree and participation coefficient.

This is a well-written manuscript and extremely thorough in its analysis. However, there are some things that the authors should include to strengthen their analysis. Looking at the various measures of centrality, it would be beneficial to also include some analysis of the rich-club coefficients of nodes in the network to further characterize gateway-ness. It would seem this would be a prudent analysis considering Fig. 8a identifies feeder and local connections similar to rich-club analyses. One thing that distinguishes this analysis is the association of port capacity with gateway-ness, but it would seem that it would be helpful to determine the degree that the rich-club coefficient captures or does not capture this property.

Another thing that may not be available, but is there any other data related to the volume of trade between ports? Much of the analysis focuses on the structural network and relates findings to port capacity, but what seems to be missing is identifying if these gateway nodes show unique properties in transportation between the core and local connections. For instance, the authors discuss how Reykjavik provides connections for Iceland and Greenland, but it would be helpful to see if shipping patterns correlate well with the apparent connectivity structure.

Overall, this is a thorough paper, that provides additional tools for examining the topological structure of a network. Nonetheless, I think it will be improved by further differentiating this measure from other hub measure (e.g., rich-club coefficient).

Reviewer #1 (Remarks to the Author):

General comment:

This is a tentative of innovating in the fields of transport geography AND network analysis. Unfortunately and to my opinion, the article mainly failed this attempt but there a number of good aspects that encourage a resubmission with stronger academic and thematic dimensions. Mainly, it lacks a central, proper and clear research question to guide the analysis. This makes the article too much data-driven and lacks of focus.

REPLY #1: We thank the reviewer for the encouraging comments and for the useful suggestions to clarify in the introduction the *central research question* of our study which (although not explicitly disclosed) was indeed behind the several methodological investigations already introduced in the previous manuscript and further improved in the revised one (lines 56-70, page 2, main text). Accordingly, we now have explicitly commented in the introduction of the revised manuscript (lines 50-55, page 2, main text) as follows:

<< These facts give rise to the necessity of addressing a central research question of the present study: What unique topological properties does the structural organization of the global liner shipping system present, and to which extent can the topological structure of this system explain the traffic capacity of world ports and trading statuses of world countries? We decided to address this pivotal question by exploiting advanced network science tools, and proposing also new ones, to investigate the many facets of global liner shipping network and their complex organization. >>

Therefore, the first four sub-sections of the “Results” provide a careful and extensive analysis of topological properties of the global liner shipping network (GLSN), from basic topological properties (e.g., ports’ centralities and economic small-world-ness) to more advanced topological properties (e.g., multiscale modular structure, ports’ connectivity patterns, structural-core organization). Then, the fifth and sixth sub-sections of the “Results” investigate the extent to which the topological structure of the GLSN—especially its structural-core organization, which is the primary finding of the present study—can explain the traffic capacity of world ports and trading statuses of world countries. Additionally, as illuminated by the second reviewer, we now have added in the end of the “Result” of the revised manuscript a seventh sub-section titled “Structural core and constraints of the GLSN”. In this sub-section, we not only provide further analyses to demonstrate the robustness of the key findings on the structural-core organization of the GLSN, using multiple datasets, but also provide further analyses and discussions for the unique constraints of the GLSN and how the formation of the structural-core organization of the GLSN is influenced by such constraints.

Based on the analyses and results stated above, we have carefully discussed in the Discussion section of the manuscript (both the previous one and the revised one) the innovation of this article and how the innovation contributes to the research fields of transport geography and network analysis. To assist the reviewer to check out the related main points in the Discussion section of the revised manuscript, we now also summarize and explain these main points as follows. First of all, we would like to emphasize that the “gateway-ness”, a topological indicator introduced for the first time by the present study and formalized in a mathematical formula, is a valuable innovation that contributes to the topological analysis of complex networks in general. This point is also acknowledged by the third reviewer. Second, the introduction of the “gateway-ness” and the related concept “gateway-hub-based structural core” is an innovation relevant to the field of transport geography, in the sense that it provides novel approaches to quantitatively characterize the structural organization of a transportation system. And, as shown in this study, such approaches provide us with a valuable new perspective to understand how the configuration of a transportation system is affected by the constraint regarding the physical geography of the earth. Let us further explain this point by taking for example the present GLSN, a maritime transportation system dedicated to delivering cargo across the globe. The introduction of the gateway-ness allows us to uncover a unique topological property of the GLSN (i.e., the gateway-hub-based structural-core organization), which proves highly relevant to world trade and has never been unfolded in previous studies on maritime shipping networks. And we further evaluated the extent to which the structural core organization of the GLSN was indeed influenced by the geographical constraint, based on computational experiments that were carefully designed to modify the geographical constraints of the real GLSN. Last but not least, we will release all the code for the analysis of “gateway-ness” and the related concept of “structural core”, once the work gets published. It will contribute to the scientific community new analytic tools that are readily applicable to not only maritime transportation systems but also various networked systems including all modes of transportation systems (e.g., airlines, railways and buses).

The time evolution would be a plus and a stronger knowledge / literature reading and understanding of the transport geography in top of pure "network analysis" literature would help. See detailed comments below for more explanation of this decision.

REPLY #2: We believe the time evolution is a very interesting point. But we want to stress that previous studies had used two types of shipping data to construct global liner shipping networks: container vessels’ movement trajectory data (e.g., Ducruet and Notteboom (2012), Kaluza et al (2010)) and liner shipping service routes data (e.g. Hu and Zhu (2009), Wang et al (2016)). To conduct time evolution analyses, while the former type of data are available at the longitudinal level that corresponds to more than one decade, the latter type of data (which are the one we

use here) are not. Therefore, so far we are unable to conduct such analyses due to the unavailability of high-quality longitudinal data on global liner shipping service routes, the reason of which is more extensively explained in the reply #30 to reviewer #1. This is a limitation not only of our study but of any study that would be conducted at the moment and would be based on liner shipping service routes data at the global level. However, we want to emphasize the following improvement of our manuscript: in previous submission we analyzed only a network associate to the year 2015, but in this revised submission we analyze also a network associated to the year 2017 and we confirm the results obtained for 2015. We are aware the reviewer #1 would prefer a longitudinal analysis involving more years, but we hope the reviewer appreciate that we have tried our best to confirm the robustness of our results and conclusions by adopting two different time points of the network structure. In the future, we will do our best to collect such data and actualize a longitudinal analysis of the GLSN. And we now recommend this direction to future studies on maritime shipping networks. For further explanations of the issue regarding the availability of high-quality longitudinal data on liner shipping service routes at the global level and all those points stated here, please see below our reply #30 to reviewer #1.

Thank you for recommending that we read more literature of transport geography, to better interpret the results obtained by network analysis. We have followed your advice; in the text of the revised manuscript, we carefully consulted and cited some related literature of transport geography. Please see below our replies to the related detailed comments.

Detailed comments:

Abstract

- global commodity trade: no, global trade is sufficient here

REPLY #3: It has been revised accordingly.

- two times "trade" is repeated in the first two sentences; consider revising the English by a native speaker

REPLY #4: This sentence has been revised as follows: "Around 80% of global trade by volume is transported by sea and thus the maritime transportation system is fundamental to the world economy". And we have carefully revised the English of the entire manuscript.

- it has self-organized: over what period of time? when?

REPLY #5: Thanks for the comment. We admit this original expression is improper and have changed it into "it is self-organized". Because the present study investigates the global liner shipping system using cross-sectional data instead of panel data, it cannot answer when the system started the self-organization processes. But this is not the research focus of the present study.

- global liner shipping network: not sure that non-transport specialists can follow this term; consider container shipping instead?

REPLY #6: This is a useful comment, which reminds us to better explain the notion of liner shipping to general readers. We now have added a further explanation of liner shipping and discussions for two different types of shipping data that have been used to construct liner shipping networks. (Please see lines 133-159, page 4 in the text of the revised manuscript, which is also pasted in the end of this reply). But we choose to keep the terminology “global liner shipping network”, because the shipping networks under study here were exactly constructed based on world liner shipping service routes data. We are afraid that, if we changed the title into the “global container shipping network”, readers probably would be confused whether our networks were based on liner shipping service routes data or based on container vessels’ movement trajectory data. The two types of data are by nature very different and are suitable for different research topics, which we have carefully addressed in the text of the revised manuscript. And we believe that readers will be able to well understand the meaning of “liner shipping” and “global liner shipping network”, since the two terminologies had been quite widely used and carefully explained by previous studies on maritime transportation in general; Notteboom and Rodrigue (2008), Agarwal and Ergun (2008), Ducruet et al (2010), and Wang and Meng (2012), to name a few. Please, verify the modification in the text of the revised manuscript (lines 133-159, page 4, main text):

<< Previous studies had used two types of shipping data to construct liner shipping networks: container vessels’ movement trajectory data^{17,19} and liner shipping service routes data^{18,24}. The two types of data are by nature very different: the former are container vessels’ voyage data logs, which can be known only after voyages are completed; the latter are service schedule data, which are regular and are typically released at least one year prior to making voyages. Both types of data have merits and are useful for analyzing different research topics. However, regarding the particular topic that is (for the first time) investigated in the present study—the association between the structure of the global liner shipping network and world trade—liner shipping service routes data are more suitable than container vessels’ movement trajectory data. There are two main reasons. First, there in fact exists a unique property of liner shipping practices: world shipping companies always pre-fix liner shipping service routes, including the end-point ports and multiple other ports between them and the vessels deployed; vessels go back and forth on such pre-fixed service routes. The information of this unique property is precisely contained in liner shipping service routes data but is lost in container vessels’ movement trajectory data; for the latter type of data, one cannot know the end-point ports of a service route. Indeed, the unique property of pre-fixed routes makes liner shipping service essentially different from other maritime transportation services, in the sense that it does not just simply serve the demand of the international trade but also could potentially influence the demand of international trade between countries. As the economic theory of induced traffic demand²⁵ posits, increased transportation capacity can stimulate extra traffic demand. Second, liner

shipping services, though carried out primarily by container vessels, do involve some non-full-container vessels in a few routes. The percent of non-full-container vessels is estimated to be about 13% in the world's total liner ship fleet, according to the latest statistics²⁶. It is true that non-full-container routes are often of less traffic capacity (measured in TEUs) than most full-container routes. However, they cannot be neglected in a global liner shipping network because their presence indeed facilitates the integration of many remote areas (e.g., small island economies) into the global maritime transportation system and thus into global markets. >>

- segregation, integration: not clear

REPLY #7: We investigated the segregation and integration of the GLSN based on analysis of modular community structure, as represented in Fig. 3 (page 13, main text). And all the related results regarding the segregation and integration of the GLSN are carefully presented in the two sub-sections titled "Multiscale modularity and hubs diversity" (pages 12-17) and "Gateway-hub-based structural core" (pages 17-23). However, since the reviewer asked for further clarification, we now have added the following explanation into the Discussion section of the revised manuscript (lines 1153-1170, page 34, main text).

<< We now better understand the segregation and integration of the GLSN by investigating the modular community structure of this network. Indeed, modular community structure is one of the most ubiquitous properties of complex networks⁷⁷; many networks of interest in the sciences, including social networks, computer networks, and metabolic and regulatory networks, are found to divide naturally into communities or modules. And segregation and integration in the topological structure of such a network with modular community structure has been explicitly explained by previous studies like reference^{78,79}. Specifically, segregation can be seen from the existence of individual modular communities, which are defined by high density of connectivity among members of the same community and low density of connections between members of different communities. Integrative processes in networks can be viewed from at least two different perspectives, one based on the efficiency of global communication and another on the ability of the network to integrate distributed information. And the integration of a network is realized by network hubs, i.e., a few nodes that are highly mutually connected and highly central in the topological structure of the network. Hub nodes can be defined on a number of criteria, depending on the specific network under study. In the present GLSN such hub nodes are found to be a few gateway-hub ports, constituting a gateway-hub-based structural core that facilitates the integration of the GLSN.>>

- reveal the presence: outcome of the article; better introduce as "this paper proposes..."

REPLY #8: It has been revised accordingly.

- provincial, gateway ...: only one is explained over three; keep it shorter or complement or remove from the abstract or say it differently?

REPLY #9: We have shortened the sentence by removing the brief explanation for the indicator 'gateway-ness'.

- economic performance: very broad; should be better linked with the rest of the main idea, which is ... still unclear to me; same for "seems to support the world trade"; are we heading towards a test of how much sea transport influenced world trade and vice-versa?

REPLY #10: Thanks for pointing them out. We have modified the expression "ports' economic performance" into "ports' traffic capacities", a straightforward expression of what we focus on. We do understand that the economic performance of ports can be analyzed from a broad range of aspects. In this study, we choose to focus on ports' performance in obtaining the traffic capacity—measured in TEU (Twenty-foot Equivalent Unit)—that are deployed by world liner shipping carriers. Note that the traffic capacity is an important factor that can determine the fate of container ports, as is also recognized in the UNCTAD's Review of Maritime Transportation 2018 (https://unctad.org/en/PublicationsLibrary/rmt2018_en.pdf).

The detected structural core of the GLSN seems to support the world trade, in the sense that there exist strong positive associations of structural-core ports to international trade statistics of individual countries and bilateral trade statistics of country pairs. Please refer to the results presented in Fig. 11 (page 28, main text) and the sub-section entitled 'Structural core and the global trade' (pages 27-28, main text).

- structural core: common expression? is that your main question, or one the issues your article wants to check? if yes then we miss a proper and central question, it is too diffused and fuzzy

REPLY #11: Yes, the expression "structural core" is commonly accepted in the literature, e.g., Daianu et al (2013) and Hagmann et al (2018).

The existence of a structural core of the GLSN topology and its strong association to world trade is one of the major findings of the present study, in answering the central research question here. Thanks for reminding us to further clarify our central research question. Now it is summarized in the "Introduction" section of the revised manuscript (lines 50-53, page 2, main text) as follows:

"These facts give rise to the necessity of addressing a central research question of the present study: What unique topological properties does the structural organization of the global liner shipping system present, and to which extent can the topological structure of this system explain the traffic capacity of world ports and trading statuses of world countries?"

Note that this study unfolds for the first time the existence of a structural core of the GLSN, thanks to a topological measure we invent and term as “gateway-ness”. Indeed, the structural core proved to be topologically central in the GLSN and significantly important in supporting long-distance maritime transportation, and moreover, strongly associated to world trade.

Introduction

- reached 0.99% instead of reach; are transshipment flows included in UNCTAD data?

REPLY #12: It has been revised accordingly.

Transshipment flows are excluded in the UNCTAD data on global merchandise trade. As the United Nations Statistics Division (UNSD) claims, “goods entering and leaving a country with the exclusive purpose of reaching a third country are excluded, since they do not add to or subtract from the stock of material resources of the country through which they pass.” (https://unstats.un.org/unsd/publication/SeriesM/SeriesM_52rev2E.pdf)

- most cost-effective way: why and how?

REPLY #13: We thank the reviewer for the comment. It reminds us to further clarify in the text of the revised manuscript that the cost-effectiveness of maritime shipping is emphasized in terms of the freight cost (but not time cost) of the mass movement of goods and raw materials around the globe. Note that such an advantage of maritime transport is also acknowledged by the United Nations Conference on Trade and Development (UNCTAD; <https://unctad.org/en/pages/newsdetails.aspx?OriginalVersionID=1850>) and also by the International Maritime Organization (IMO; <https://business.un.org/en/entities/13>)—the United Nations system’s regulatory agency for the maritime sector. Now we report this in the revised text; lines 21-22, page 1, main text.

<< Maritime transport, by far the most cost-effective way (in terms of freight cost) to the mass movement of goods and raw materials across the globe, is the backbone of international trade. >>

- 70% or 80% of global trade: sources?

REPLY #14: Around 80% of global trade by volume and over 70% of global trade by value are carried by sea and are handled by ports worldwide. Statistics are sourced from Review of Maritime Transport (Series), which is published annually by the United Nations Conference on Trade and Development. We now have added into the revised manuscript the data source, in line 24, page 1, main text:

<< [https://unctad.org/en/Pages/Publications/Review-of-Maritime-Transport-\(Series\).aspx](https://unctad.org/en/Pages/Publications/Review-of-Maritime-Transport-(Series).aspx). >>

- eco. dev. see Bernhofen et al. for an earlier study without the network dimension but econometrics based on a time-series

REPLY #15: Thanks for the recommendation. We have consulted the recommended paper and also have cited in the text of the revised manuscript. Indeed, it demonstrated the importance of maritime transport in supporting international trade. See the following text in lines 30-31, page 1, main text.

<< The importance of maritime transport in supporting international trade makes it indispensable to the sustainable socio-economic development of our world³. >>

- high-capacity: less than bulks

REPLY #16: We agree, and we never claimed the opposite. To avoid any ambiguity, we now have removed this word “high-capacity” from the sentence. See, lines 37-38, page 1, main text.

<< [...] the service of transporting goods primarily by ocean-going container ships that transit regular routes on fixed schedules. >>

- UNCTAD index: be more critical; what is the value of this index compared to the reality of shipping and to other ones like the World Bank's LPI?

REPLY #17: The UNCTAD's liner shipping connectivity index (LSCI) is the one of most relevance to the present study, compared to the reality of shipping and to other ones like the World Bank's LPI.

Specifically, in recognition of the vital role of liner shipping connectivity in determining countries' accesses to world markets, UNCTAD developed the LSCI that aims to investigate individual countries' positions within the global liner shipping network based on the information of world liner ship fleet's service schedules. This practical and interesting point, together with other factors, motivated us to carry out the present study, to explore the unique structural properties of the global liner shipping network and the extent to which the traffic capacity of world ports and trading statuses of world countries can be explained by the structural properties of the global liner shipping network. For calculation details about the LSCI, please refer to <https://unctadstat.unctad.org/wds/TableView/tableView.aspx?ReportId=92>.

The World Bank's logistics performance index (LPI) is defined very differently from the UNCTAD's LSCI. LPI is generated based on a worldwide survey of operators on the ground (i.e., global freight forwarders and express carriers). This indicator aims to help countries identify the

challenges and opportunities they face in their performance on trade logistics, according to operators' feedback on the trade logistics "friendliness" of the countries in which they operate and those with which they trade. It is important to note that maritime shipping is only one specific segment of the trade logistics that the LPI focuses. Therefore, the LPI is not that relevant to the present study. For calculation details about the LPI, please refer to <https://lpi.worldbank.org/>.

- at THE national level (check English)

REPLY #18: It has been revised accordingly.

- given the port-interface: this is a geographic area, not a research question, so why is the point here? Do you mean port-city relationships? which ones? spatial, economical, else?

REPLY #19: We thank the reviewer for pointing this out, which helps us better deliver the message of the corresponding sentence. Yes, what we want to address is the port-city relationships in economic development and spatial planning. It has been revised accordingly. See revised manuscript; lines 45-46, page 2, main text.

<< And given the port-city relationships in economic development and spatial planning⁴, port development is also essential to [...] >>

- global port cities: why only the global ones??

REPLY #20: We actually intended to emphasize "port cities in the world", not just global ones. We have modified the expression to clarify this point. See text of the revised manuscript, lines 46-47, page 2, main text.

<< [...] port development is also essential to the sustainable urbanization process for port cities in the world [...] >>

- witted: witnessed?

REPLY #21: Thanks for pointing this out. Yes, it should be "witnessed". We have corrected this typo.

- several realistic questions: academic questions?

REPLY #22: We now have summarized the central academic question of the present study as follows (lines 50-53, page 2, main text):

<< These facts give rise to the necessity of addressing a central research question of the present study: What unique topological properties does the structural organization of the global liner shipping system present, and to which extent can the topological structure of this system explain the traffic capacity of world ports and trading statuses of world countries? >>

- self-organizing capacity: too long sentence; consider developing each issue separately

REPLY #23: We have broken it into two sentences. See text of the revised manuscript (lines 56-59, page 2, main text):

<< With respect to the self-organizing complexity of transportation systems, what specific complex structure has the global liner shipping network evolved into? The global liner shipping system does not result from any “global governance”, but instead it is a result of the multiple decisions on service network design of world’s individual shipping companies that seek in primary for profits. >>

- similarities: of which type?

REPLY #24: It is similarities of connectivity patterns. We have clarified it in the text of the revised manuscript, lines 64-66, page 2, main text:

<< Do ports present some similarities of connectivity patterns, with respect to their positions in the underling structure of the global liner shipping network? >>

- socio-economic factors: which ones?

REPLY #25: The socio-economic factors under consideration here are socio-economic statuses of world countries in a global economy, i.e., the international trade value of countries and the bilateral trade value between countries. We have further clarified this point in the text of the revised manuscript, lines 67-70, page 2, main text:

<< [...] it is essential to characterize the structure of the global liner shipping network and its association to socio-economic statuses of world countries in a global economy (i.e., the international trade value of countries and the bilateral trade value between countries). >>

- BY unveiling

REPLY #26: It has been revised accordingly.

- contagion, disease: not so much related with the present network

REPLY #27: We have removed them from the text of the revised manuscript.

- "remain barely explored" not sure given the last 2 edited books on maritime networks published by Routledge in 2015 and 2017 and a plethora of related analyses in multiples journal papers using complex networks among other methods; see also Wang et al. 2016 in Nature Communications on the same network studied here ...

REPLY #28: We understand that there exist some illuminating studies on maritime networks in general, including those you've mentioned here and those we already cited in the previous manuscript. Scientific studies specifically dedicated to global maritime shipping networks, however, remain relatively rare as compared with other types of spatial networks. This point is also acknowledged by the second reviewer. As the second reviewer writes in the general remarks to the authors, "this paper analyzes the network of ports connected by freight ships, and it is a relatively rarely studied type of spatial network."

Thanks for reminding us the study of Wang et al. 2016. Now it is cited in our revised manuscript. But please note that this study was published in Scientific Reports (<https://www.nature.com/articles/srep34217>), rather than in Nature Communications. Indeed, Wang et al. also used the liner shipping service routes data, the same type of data as our study adopted, but their data are of much lower quality. We are confident of two main merits of our study, as compared with the one of Wang et al; to further clarify this, we now have added the following text into the Discussion section of the revised manuscript (lines 1203-1212, page 35, main text). The revised text is also pasted below. Besides, we are aware of the limitation of our study (which is also a limitation of the one of Wang et al), that we could not design longitudinal analyses due to the obstacles regarding the data availability; for a detailed discussion about this limitation, please see below our reply # 30 to the reviewer #1.

<< The present study is not the first one that adopts the liner shipping service routes data to investigate a GLSN, but as compared with previous literature²⁴ it has two unique merits. First and most important, we focus on uncovering not only the unique structural properties of the GLSN but also its strong and positive association to world trade, while Wang et al.²⁴ only focuses on two specific structural properties: robustness and vulnerability. Second, the data used in our study are of a better quality than the data used in reference²⁴: Ours are the liner shipping service routes data of world's top 100 liner shipping companies, which altogether account for more than 92% of the world's total capacity (in TEU); the data of reference²⁴ are the liner shipping service routes data of world's top 25 liner shipping companies, constituting only about 80% of the world's total capacity (in TEU). >>

- data quality / shipping industry: what is the link?

REPLY #29: Those marked advances in data quality and remarkable changes in the maritime shipping industry that had occurred over the last decade provide greater opportunities for the

scientific community to deepen our understanding of the global maritime shipping system. Hence, they are linked with the interest of the present study and also future studies dedicated to maritime shipping networks. Detailed illustrations are as follows. And, to better explain this issue to general readers, we now also modify the text of the revised manuscript (lines 79-87, pages 2-3, main text) in accordance with the following illustrations.

First, the quality of the required data for constructing a global maritime shipping network—where nodes are world ports and links are inter-port connections realized via cargo ships—has been greatly improved in the recent decade, thanks to the progresses of technologies involved in collecting such shipping data. To the best of the authors' knowledge, so far such data can be categorized mainly into two types: cargo vessels' movement trajectories data, which are now recorded via automatic identification system (AIS); liner shipping service routes data, which are released by individual shipping companies in the world.

For more information on the recent development of AIS technology used in the shipping industry, please consult the following website of a leading database: <https://maritimeintelligence.informa.com/resources/product-content/understanding-the-automatic-identification-system>

For more information on the recent development of the systematical collection of world shipping companies' liner shipping service routes, please consult the following website of another leading database specialized in liner shipping: <https://www.alphaliner.com/>

Second, since the 2008 global financial crisis, the liner shipping industry has seen significant adjustments in the service networks of both large and small shipping companies. Such changes provide precious opportunities to the scientific community to expand previous findings on the structural properties of the global liner shipping system and as well as its various related issues. New studies on the global liner shipping network that are based on recent liner shipping data are therefore badly needed, as our present work provides one example.

Please find below the text of the revised manuscript (lines 79-87, pages 2-3, main text):

<< The improved availability of high-quality shipping data is especially thanks to the progress of AIS (automatic identification system) technologies involved in collecting the cargo vessels' movement trajectories data and to the progress of systematical collection of world shipping companies' liner shipping service routes data. Meanwhile, the liner shipping industry has seen significant adjustments in the service networks of both large and small shipping companies, since the 2008 global financial crisis. Those marked advances in data quality and remarkable changes in the maritime shipping industry provide greater opportunities for the scientific community to deepen our understanding of the structural properties of the global liner shipping system and various issues related with this system. >>

- a recent ...: why not adopting time series data like in many other researches of the kind to demonstrated a structural change?

REPLY #30: Indeed, one limitation of our study is the lack of longitudinal analyses of the structural change of the GLSN, and this is also a limitation of other existing studies of GLSN that are based on liner shipping service routes data at the global level. We have already clarified this limitation of our revised study in the reply #2 and reply #28 to reviewer #1. To the best of our knowledge, at the moment, high-quality longitudinal data on liner shipping service routes at the global level are not openly and freely available to the public, for the reasons that we now have explained in the modified text of the revised manuscript (which is also pasted at the end of this reply).

[Redacted]

Therefore, at this moment it is beyond our ability to obtain any high-quality longitudinal data on liner shipping service routes at the global level. However, we want to stress that in previous submission we analyzed only a network associate to the year 2015. In this revised submission we analyze also a network associated to the year 2017 and we confirm the results obtained for 2015. We are aware the reviewer would encourage an analysis based on more years datasets, but we hope the reviewer can appreciate that we have tried our best to confirm the robustness of our results and conclusions by adopting two different time points of the network structure. In conclusion, in order to address the reviewer's comments, we now have reported in the revised version of the manuscript all those points stated above. We hope in this way, that also other readers of our manuscript will understand the obstacles that should be surpassed to be able to design longitudinal studies. Please refer to the modified text below, which is added in the revised study (lines 1213-1232, pages 35-36, main text):

<< One limitation of existing studies of GLSN that are based on liner shipping service routes data at the global level, including the present study, is the lack of longitudinal analyses of the GLSN

structure. This limit is mainly due to two aspects regarding liner shipping industry. One is that historical data on world's individual shipping companies' liner shipping service routes—which are abstracted from their liner service schedules data—are very difficult to recover, if not impossible at all; unlike the historical data of ships' movement trajectories that are by convention recorded as ships' voyage data logs, the historical data on world shipping company's liner shipping service routes are just not recorded by convention. Another aspect: although there now have emerged a few commercial databases committed to systematically collecting the data on world's liner shipping service routes from the recent decade onwards, such data—if on a large scale like the scale in the present study (i.e. world's top 100 liner shipping companies in terms of capacity measured in TEU)—are not openly and freely available to the public; one should understand that such data, if on a large scale, are indeed sensitive information that can be used in the service network design of a liner shipping company and thus can influence the core business of the company. Given this limitation, in the present study we initially analyzed a GLSN associated to the year 2015, and later we also analyzed a GLSN associated to the year 2017 and we confirmed the robustness of our results. Still, we hope that in the near future the obstacles regarding the data availability will be surpassed, and it will be worthwhile for future studies to investigate how the structural organization of the GLSN had evolved over time and how the evolution of the GLSN structure interplays with the development of world trade. >>

- 2015: better justify the use of this date for this particular study

REPLY #31: The same reply as the reply #30.

- GLSN: we need a proper mathematical definition of the graph

REPLY #32: Done accordingly. See, revised manuscript (lines 90-91, page 3, main text):

<< The GLSN is an undirected graph consisting of a set of nodes (i.e., ports) and a set of edges (i.e., connections between the ports). >>

- structural embeddedness: define

REPLY #33: It has been clarified in the text of the revised study, lines 110-111, page 3, main text:

<< Since ports are economic entities, we are also interested to test the hypothesis of structural embeddedness, which refers to nodes' patterns of connections with other nodes in the network. >>

- how to test last paragraph ideas? see DEA studies of port performance indicators in MPM, JTG, TRA, TRE etc.

REPLY #34: We thank the reviewer for this valuable comment. The core idea of the last paragraph in the introduction section of the original manuscript—which now is the

second-to-last paragraph in the revised manuscript—is that a few structural indicators (i.e., degree, betweenness, gateway-ness, provincial-ness, and connector-ness) are found to be highly correlated with ports' economic performance. In the present study, the economic performance of a port is measured by the traffic capacity (in Twenty-foot Equivalent Unit) deployed by world liner shipping carriers, ground-truth data derived from the database. We have carefully consulted some DEA studies of port performance indicators that the reviewer had recommended, e.g. Wang et al (2003), Cullinane et al (2004), Rios and Maçada (2006), and Nguyen et al (2016). We do understand that the port performance indicators adopted by those studies are valuable in reflecting the production performance of ports; among them, two commonly accepted indicators are container throughput (measured in TEUs) and average number of containers handled per hour/ship. Unfortunately, we do not have any access to such data. It is therefore infeasible for the present study to further test the correlation between the structural indicators and the port production performance indicators adopted in those DEA studies. To the best of the author's knowledge, data on such indicators of port production performance for world ports are not freely available.

Results

- see the chapter Liner Service Networks in Container Shipping in the edited Kogan Page book for strengthening your explanation of service types

REPLY #35: In the text of the revised manuscript, we have recommended this study to readers who might be interested in getting more information on different types of liner shipping service. See revised manuscript (lines 163-165, page 4, main text):

<< For more information on how these services can be categorized into different types in the liner shipping industry operation, please refer to the reference²⁷. >>

- seaside transport capacities: awkward expression, consider changing

REPLY #36: In the text of the revised manuscript, we have modified the expression “seaside transport capacity of a port” into the expression “traffic capacity of a port” (abbreviated as port capacity).

- complete graph: see the Space-P introduced by Hu and Zhu 2009 (Physica A) and also Deng et al. 2009 for a reference to your same methodology to define the network... This needs you to further discussing the choice of space-P instead of space-L!!

REPLY #37: In the text of the revised manuscript, we have added the following paragraph to further discuss the choice of using space P to build a GLSN topology out of the liner shipping service data. We also referred to the recommended studies and also two earlier studies (i.e., Sen

et al (2003) and Sienkiewicz Hołyst (2005)) that initially introduced the concept of space L and P for representing the topology of transport networks. See revised manuscript (lines 181-193, page 5, main text):

<< To further illustrate such a method of GLSN construction, it is worth mentioning the so-called concept of space L and P ^{13,28} that has been used in many transport network studies to properly represent the topology of transport networks. In space L , a link between two nodes exists if they are consecutive stops on a same route; in space P , a link between two nodes means that there is a single route connecting them. Consequently, the node degree in space- L topology is just the number of directions one can take from a given node, and the node degree in space- P topology indicates the total number of nodes reachable to a given node by using a single route²⁸. Both of the two methods of network topology representation have been adopted in maritime transportation network research^{17,18}, and the methodological choices depend on specific research focuses. The present study emphasizes the fact that cargo between any two ports on a same service route can be transported via a single ship, which is indeed important information contained in liner shipping service routes data. Therefore, a GLSN topology was created here in space P . >>

- page 5 line 150: wrong, see Bernhofen et al. 2013 (working paper) as well as Guerrero et al. 2015 (Routledge book)

REPLY #38: Thanks for recommending the two studies to us. We admit the initial expression was improper and now have cited the two recommended studies in the text of our revised manuscript. Please find our revision pasted in the end of this reply. Reasons for the current modification are explained as follows.

We agree that the two recommended studies are relevant to our paper. These two studies and our study all contribute valuable insights to understanding the broader topic of socio-economic linkages between world trade and maritime transport. Since we do acknowledge the two studies' meaningful attempts to quantitatively explore the influence of containerized shipping on international trade flows, we now have cited them in the text of the revised manuscript to better frame our study in a wider literature context.

However, we would like to emphasize the unique contribution of our study, as compared to the two recommended studies. Our work for the first time investigates the extent to which world trade is associated to the topological structure of a modern GLSN (which is defined by the links among world ports). But in the two recommended studies, not any shipping network's topological structure was even analyzed. See revised manuscript (lines 211-216, page 6, main text):

<< Following the meaningful attempts of previous studies^{3,30} to quantitatively explore the influence of containerized shipping on international trade flows, this study for the first time investigates the association between world trade and the topological structure of a modern GLSN. To this end, we perform a careful and extensive analysis of its basic topological properties and the associated meaning. This analysis can enlighten maritime scientists on understanding the main structural properties which might influence the GLSN functionality. >>

- subsection "degree distribution...": too much common in network analysis, why not comparing with previous studies instead of restarting from scratch yet with another (but already studied) network?

REPLY #39: Degree is the most fundamental network measure to which most other connectivity measures are ultimately linked, and thus investigating degree distribution is generally a first step to detect whether a network is structurally heterogeneous. This reason for analyzing the degree distribution of ports is written in the text of both the present and previous manuscripts. Please see lines 218-222, pages 6-7 of the present manuscript.

Actually, we did compare our finding of ports' degree distribution to the findings of some earlier studies on maritime shipping networks. Indeed, our finding—the non-scale-free behavior of ports' degree distribution in a global liner shipping network—is consistent with the finding of previous work of Deng et al (2009) and Kaluza et al (2010); please see line 226 in page 7 of the revised manuscript. Now we have added another comparison between our finding and the finding of Gastner and Ducruet 2015 (Routledge book), according to the reviewer's very next comment. Please see this modification in the revised manuscript (lines 227-229, page 7, main text), which is also pasted below.

<< Another study that was based on cargo ships' trajectories data reported that world ports' degree distribution was best fitted by a lognormal function in early years (around 1890-1960) and by a Weibull distribution in later years (around 1960-2008)³⁴. >>

- not scale-free? see Gastner and Ducruet 2015 (Routledge book) on the fit between statistical models and real-world degree distributions for the global shipping network since 1890 up to 2000...

REPLY #40: No, it is not strictly scale-free. According to our study and the previous ones of Deng et al (2009) and Kaluza et al (2010), port degree probability distribution follows an exponential decay, instead of power-law. Specifically, regarding the statistical test on the power-law distribution, our study adopted the method of Clauset et al (2009), and Kaluza et al (2010) used another method introduced by Burnham and Anderson (1998) and Edwards et al (2007). Both methods are widely accepted in the field. Note: the shipping data used in Kaluza et al (2010) is

cargo vessels' movement trajectory data, which is essentially the same type of shipping data as used in Gastner and Ducruet (2015). In addition, we now have adopted another recent method proposed by Voitalov et al (2018; <https://arxiv.org/abs/1811.02071>) to conduct the statistical test on power-law distribution, and we report in the revised manuscript that the port degree probability distribution also failed in this new test (lines 222-224, page 7, main text).

<< We report the cumulative probability distribution of port degree is well fitted by an exponential function (Fig. 2a), instead of power-law (Tests on the power-law distribution of the data failed based on the method of Clauset et al³¹ and the method of Voitalov et al³² as well). >>

We also have carefully consulted the work of Gastner and Ducruet (2015). In fact, they did not report any scale-free behavior of ports' degree distribution, but instead they reported that world ports' degree distribution was best fitted by a lognormal function in early years (around 1890-1960) and by a Weibull distribution in later years (around 1960-2008). Please see in the end of the first paragraph of the conclusion section of their work, "when the empirical call distribution is replaced by the degree distribution, the lognormal model is plausible in early years, but in later years Weibull distributions fit the data better". However, we acknowledge the work of Gastner and Ducruet (2015), and thus we have added in the revised manuscript another comparison between our finding and Gastner and Ducruet (2015)'s. This point is already written in our reply #39 to the previous comment just before this one.

- concentration in big ports: this is not happening by "nature"; should develop a bit more here or earlier the causes of economies of scale since the mid-1990s in this particular sector and the emergence of hubs etc. Given your study year you should also take into account the advent of slow steaming ...

REPLY #41: Thanks for the kind advice. We have added the following two sentences to further explain the traffic concentration in big ports, by carefully consulting the related work of Rodrigue and Notteboom (2010) and Zohil and Prijon (1999). See revised manuscript (lines 238-242, page 7, main text).

<< Most of those big ports are intermediate hubs, whose emergence was mainly fostered by world liner shipping companies in the pursuit of economies of scale (Rodrigue and Notteboom, 2010)³⁷. Since the mid-1990s, the world has seen a fast increase of vessel size, frequency of liner shipping companies' services, and the concentration of their services at a few large intermediate hubs (Zohil and Prijon, 1999)³⁸. >>

- figure 2: average BC: useless

REPLY #42: In the revised manuscript, we have removed the average BC and the average CC from figure 2.

- trunk line: explain further for non-specialists

REPLY #43: We have added the following sentence to further explain trunk lines for non-specialists. See revised manuscript (lines 300-302, page 9, main text):

<< In the context of the hub-and-spoke service network design of world liner shipping companies, trunk line services are those linking hub ports of different geographical regions. >>

- anomalous centrality in shipping: not as critical as for airlines, to be discussed ...

REPLY #44: In consideration of this comment, we have modified the related text in the revised manuscript to make our expression more appropriate. Briefly, we are unable to precisely evaluate if the degree-betweenness anomalies in the GLSN is less or more critical than in the global air transportation network, due to lacking the full information on world airports' degree and betweenness centralities. But this does not harm the present study, since we are not focusing on evaluating the extent of the degree-betweenness anomalies observed in the two networks. Instead, we are simply interested in understanding the degree-betweenness anomalies of the GLSN by further investigating this network's community structure, illuminated by a previous study on the global air transportation network (Guimerà et al (2005)). See revised manuscript (lines 313-316, page 9, main text):

<< This is also the case of the GLSN, even though the degree-betweenness anomalies in the GLSN may not be as critical as in the global air transportation network. Therefore, the GLSN's community structure will be unveiled and discussed below in the present paper. >>

- unevenly developed: in what sense?

REPLY #45: It is in the sense of port technology. See revised manuscript (line 317, page 9, main text).

<< Most of the world ports, though unevenly developed in port technology, [...] >>

- self-organized: explain

REPLY #46: The GLSN is "self-organized", in the sense that the GLSN does not result from any global blueprint or central governance, but instead it is a result of the multiple decisions on service network design of world's individual shipping companies that seek in primary for profits. This explanation was already written in the text of the previous manuscript, and it is also written in the revised manuscript (lines 57-59, page 2, main text).

<< The global liner shipping network does not result from any global blueprint, but instead it is a result of the multiple decisions on service network design of world's individual shipping companies that seek in primary for profits. >>

- 0.723: it is 0.713 in the figure; should discussion this high value - because of choosing space-P instead of space-L for instance, see 2012 global networks paper on the GLSN 1996-2006

REPLY #47: Thank you for carefully reading the manuscript and pointing out this mistake. We have corrected the typo, and the correct one is 0.713. We also have added the following brief discussion about the influence of different methods of network topology representation on the average clustering coefficient and the average shortest path length of the GLSN. See revised manuscript (lines 342-349, page 10, main text).

<< We note that for transportation networks based on a series of transport routes (e.g., subway networks and liner shipping networks), the space-*P* representation of network topology (where any two nodes in a route is assigned with an edge) will naturally give an average shortest path length shorter than and an average clustering coefficient larger than their respective counterparts given by the space-*L* representation of network topology (where only consecutive nodes in a route is assigned with an edge). Previous studies^{17,18} had compared some basic topological properties of global maritime shipping networks observed under the two representations of network topology. >>

- node pairs: do you mean nodes?

REPLY #48: "Node pairs" means "pairs of nodes". Now we have revised the expression accordingly.

- cost of building: much composite and hard to define, especially in the context of shipping; the "imaginary routes" depicted by Ross Robinson (1968)

REPLY #49: We do understand that the navigation routes between ports are not absolutely fixed on the surface of the ocean, unlike the routes in ground transportation networks such as buses and railways. And we agree that in reality the financial cost of building up a liner shipping service network is composite and hard to calculate precisely. In the present study the wiring cost of a network configuration is estimated as the sum of the wiring cost for individual connections, based on Latora and Marchiori (2001, 2003)'s method. Specifically, for spatial networks where nodes are embedded into physical space, the wiring cost for a connection is assumed to be proportional to the physical length between the two end nodes. Granted, this definition considers only the influence of physical distance on the cost of network configuration, and thus may not be precise in estimating the real financial cost of building up a network configuration.

However, the analysis of such wiring cost of a network configuration here serves the aim to understand the economic-small-world property of the network configuration, rather than to precisely calculate the financial cost of building up a network configuration. Indeed, it had been reported that many real-life small-world spatial networks are economical in the sense of supporting high global and local efficiency with low cost (Latora and Marchiori (2001, 2003)), ranging from brain networks (e.g., Bassett and Bullmore (2006), Bullmore and Sporns (2009)) to transportation networks (e.g., Porta et al (2006), Crucitti et al (2006)). Therefore, we are interested in exploring whether the GLSN is self-organized into an economical structure, in an effort to shed new light on understanding the unique structural properties of the existing configuration of the GLSN.

- LAT and LONG bird's fly: why not nautical distances to get closer to reality?

REPLY #50: We now have obtained the nautical distance data from an online shipping platform (<https://www.searates.com/services/distances-time/>), and we have revised all the distance related results reported in the manuscript. At the time of initial submission, we did not have such empirical data on nautical distance. Interestingly, the revised results based on the precise nautical distance data remain almost the same as the previous results based on the approximate great-circle distance data. We list below the related modifications in the main text of the revised manuscript, which are based on the real nautical distance data. In addition, we now also put in supplementary information of the revised manuscript (i.e., supplementary notes 2, 3, and 14) the previous results based on the great-circle distance data. See the following modifications in the revised manuscript, main text:

(1) lines 363-368, page 10:

<< Indeed, it has global and local efficiency arriving to 82.7% ($p < 0.001$, testing against a configuration null model) and 93.2% ($p < 0.001$, testing against a configuration null model) of the corresponding ideal case of network configuration (i.e. all ports are connected with each other), respectively, but holds an extremely small network cost that only accounts for 1.5% ($p < 0.001$, testing against a configuration null model) of the ideal case (for details of testing the statistical significance of the economic small-world-ness of the GLSN, *see supplementary note 2*). >>

(2) lines 458-459, page 13:

<< [...] for instance, 87.4 percent of inter-port connections longer than 10,000 km are inter-community links; [...] >>

(3) lines 693-723, pages 21-23:

<< First, core connections themselves tend to be longer than feeder and local connections (Fig. 9b). Measured by real nautical distance⁴⁶ (hereafter referred to as distance), the average length

of core connections (average = 10,233 km, SD = 6,567 km) is 2.0 times of the average over all inter-port connections (average = 5,116 km, SD = 5,482 km); feeder connections (average= 7,612 km, SD = 6,059 km), 1.3 times; local connections (average = 3,585 km, SD= 4,379 km), 2.9 times.

When looking at the shipping distance for cargo transportation among all the non-core ports in the GLSN (Fig. 9c), we found 16.7% of the total shipping distance—measured by the distance along the edges of their shortest paths—is taken up by core connections. As core connections only account for 3.2% of the total number of inter-port connections in the GLSN, it makes a ratio of distance fraction to the connection fraction to be 5.2, relative to 1.7 and 0.4 for feeder connections and local connections, respectively. When considering those shortest paths that pass through the structural core (i.e. travelling across at least one core connection), the proportion of shipping distance taken up by core connections reaches 62.0%, and that by feeder and local connections 33.4% and 4.6%, respectively.

In addition, we estimated the physical length of an inter-port connection as the great-circle distance based on ports' geographical locations of latitude and longitude, and then repeated the analysis. We found all the results reported here remain almost invariant (*supplementary note 14*).

Fig. 9 Statistics of the core, feeder and local connections. **(a)** Schematic illustration of path metrics; the geographical length of an inter-port connection is measured as the real nautical distance⁴⁶ between the two ports, and shipping distance of any port pair is the sum of geographical length of edges along the shortest path. **(b)** Ratios of length percentage to link percentage for core connections, feeder connections and local connections, respectively. **(c)** Percentages of core connections, feeder connections and local connections in the total shipping distance of all shortest paths between non-core ports (**Left**), and of shortest paths between non-core ports which travel through the structural core (**Right**). To better understand those three types of connections related with the structural-core organization of the GLSN, one is encouraged to refer to the hub-and-spoke service network configuration (Hu and Zhu, 2009)¹⁸ that is widely adopted by world liner shipping carriers in practice. The hub-and-spoke configuration is illustrated in the *supplementary Fig. 12 in supplementary note 9*. >>

Multiscale modularity

- not sure whether this analysis is well connected to the previous

REPLY #51: We thank the reviewer for this comment, which encourages us to improve the logical consequence of the topics discussed in the draft. The two sections discuss different topics: the first is on the basic topological measures; the second one is on a more advanced topic of meso-scale network structure, which is the community structure of the network evaluated by multi-scale modularity. Now the section "Multiscale modular structure" begins with this clarification. See revised manuscript (lines 432-434, page 12, main text):

<< In the previous section we discussed basic topological properties of the GLSN. In this section we now move to investigating the community structure of the GLSN based on multi-scale modularity, which is a more advanced topic of meso-scale network structure. >>

- length of physical links: yes but those longer links over 10,000km do not exist; they are artificially "made" by the choice of space-P over space-L and so include intermediary stops ... be more critical and comparative

REPLY #52: In the context of the present study, the existence of a link between two ports means that cargo between the two ports can be directly transported by a single vessel that travels on a single service route, regardless of how distant the two ports are and how many intermediary stops the vessel makes. Note: such services of multiple ports of call are uniquely adopted in liner shipping, which is an important factor that makes liner shipping essentially different from other shipping modes (e.g. dry bulk shipping, oil shipping). It is therefore that, methodologically, the space-P representation of network topology is more suitable for the present study on global liner shipping network. And the selection of space-P representation method has already been carefully explained in our previous reply #37 to reviewer #1.

- try to name your 3 port categories in a more generic manner ... "provincial" often refers to subnational ...

REPLY #53: Thank you for helping us scrutinize the terminology usage. We do understand the word "provincial" is commonly used to refer to something pertaining to a province, though it also has a meaning pertaining to local. However, we would like to keep the current names of the three port categories, in order to make our study consistent with the large body of related existing studies in terminology usage. Please note: the "provincial hubs" and "connector hubs" are two key terminologies in network cartographical analysis, which were initially introduced by Guimerà and Amaral (2005) and later had been widely adopted by other studies in the field. The "gateway hubs" is introduced by the present study, adding a new analytics tool to network cartographical analysis.

- figure 5: absolutely NO geographic logic in this result; even certain regions such as the MEd, the Caribbean and SE Asia which are hub regions do not even appear in the right types such as "connector" for instance!

REPLY #54: Please note that in figure 5, there do exist a few connector hubs in the Mediterranean region and in Southeast Asia. The identified Mediterranean connector hubs are ports of Valencia, Genoa, Barcelona, Algeciras and Tangier, and the Southeast Asian connector hubs are ports of Singapore, Tanjung Pelepas, and Port Kelang. The names of all three types of hub ports are available in a supplementary file named "supplementary file to Fig. 5". In the Caribbean Sea there was no port that could be reported as a connect hub, according to the definition of connect hubs adopted in the present study. Further explanations are as follows. We now also have modified the relevant text of the revised manuscript in accordance with the explanations below; the modified text is in lines 1420-1427, pages 40-41, main text, and it is also pasted at the end of this reply.

By definition, "connector hubs" here are just different from "hub regions" mentioned by the reviewer. Specifically, "connector hubs" refer to ports whose connections are much equally distributed among different communities in the GLSN, emphasizing a high-level of equal participation in different parts of the overall network structure. The extent to which a port's connections are equally distributed is measured by a topological indicator termed participation coefficient (Guimerà and Amaral (2005)); please see a detailed illustration of this indicator in the sub-section titled "Defining various hub ports in the GLSN" in the "Methods" section.

"Hub regions", in the context of maritime shipping, refer to certain geographical regions—such as the Mediterranean Sea, the Caribbean Sea and Southeast Asia—where a few transshipment hub ports are located. Such transshipment hubs transport cargoes between different trunk

shipping routes and also between trunk routes and feeder routes, regardless of whether the transported cargos concentrate on a few specific destination regions or distribute evenly among many destination regions.

To sum up, if a transshipment hub distributes its connections relatively evenly among ports in many communities, then it will be identified as a connector hub in the present study. If a transshipment hub concentrates its connections with ports from only a few particular communities (like one or two), then it will not be identified as a connector hub. According to the world liner shipping service routes data and the definition of “connector hubs” of the present study, Caribbean ports’ shipping connections are not sufficiently evenly distributed among different communities in the GLSN, and thus they are not identified as connector hubs here. Such results do not contradict the reality that the Caribbean Sea accommodates a few transshipment ports for liner shipping and thus serves as a hub region in maritime transportation. See revised manuscript, lines 1420-1427, pages 40-41, main text.

<< Note that, by definition, “connector hubs” here are just different from the so-called “transshipment hubs” that are frequently discussed in literature of maritime transportation research. Specifically, “connector hubs” refer to ports whose connections are much equally distributed among different communities in the GLSN, emphasizing a high-level of equal participation in different parts of the overall network structure. Transshipment hubs are ports that facilitate cargo transportation between different shipping routes, regardless of whether the transported cargos concentrate on a few specific destination regions or distribute evenly among many destination regions. >>

- figure 8 should refer to hu and zhu 2009 also drawing a figure on hub-and-spokes structure

REPLY #55: We have made the revision accordingly. Now the reference of Hu and Zhu (2009) is referred to in the caption of this figure (which is numbered as figure 9 in the revised manuscript), and a figure on hub-and-spoke structure is also provided (i.e., supplementary Fig. 12 in supplementary note 9). See revised manuscript pasted below.

Caption of figure 9; lines 719-723, page 23, main text:

<< Fig. 9 [...] To better understand those three types of connections related with the structural-core organization of the GLSN, one is encouraged to refer to the hub-and-spoke service network configuration (Hu and Zhu, 2009)¹⁸ that is widely adopted by world liner shipping carriers in practice. The hub-and-spoke configuration is illustrated in the *supplementary Fig. 12 in supplementary note 9*. >>

Reviewer #2 (Remarks to the Author):

This paper analyzes the network of ports connected by freight ships. It is a relatively rarely studied type of spatial network. The authors adapt some existing methods and ideas to this data to find uncover the organization of the system. I don't think the findings are interesting enough to publish this in Nature Communications.

REPLY #1: First of all, we would like to emphasize that the gateway-ness, a topological indicator introduced for the first time by the present study, is a valuable innovation for the topological analysis of complex networks in general. This point is also acknowledged by reviewer 3. The introduction of the gateway-ness allows us not only to uncover the gateway-hub-based structural-core organization of the GLSN, but also to reveal the important and strong association between the GLSN topological organization and world trade (i.e., the international trade of individual countries and bilateral trade between countries), as well as the strong association between the gateway-ness and port capacity. To the best of our knowledge, gateway-ness is a relevant discovery that has never been reported in previous studies.

Second, the liner shipping service routes data at the global level, which we used in the present study, had rarely been tapped into by previous studies on maritime shipping networks. Most of the previous studies were based on the cargo vessels' movement trajectory data (i.e., vessels' voyage data logs), for instance, the ones of Kazula et al (2010) and Ducruet and Notteboom (2012). Both types of data have merits and are useful for analyzing different research topics. However, regarding the specific topic that is (for the first time) investigated in the present study—the association between the structure of the global liner shipping network and world trade, liner shipping service routes data are more suitable than container vessels' movement trajectory data. And, it is our hope to release the global liner shipping network data to the whole scientific community, once the paper is published. To clarify the points here, now we have added into the text of the revised manuscript a discussion regarding the difference between the two types of data for maritime shipping network studies; please see lines 133-159, page 4, main text. The modifications are also pasted below.

Putting together, we believe that the new topological indicator "gateway-ness" and the related concept "gateway-hub-based structural core" and our study as a whole are of interest to the research fields of transportation engineering and planning, network science, and macroeconomics and international trade. See revised manuscript (lines 133-159, page 4, main text).

<< Previous studies had used two types of shipping data to construct liner shipping networks: container vessels' movement trajectory data^{17,19} and liner shipping service routes data^{18,24}. The two types of data are by nature very different: the former are container vessels' voyage data logs,

which can be known only after voyages are completed; the latter are service schedule data, which are regular and are typically released at least one year prior to making voyages. Both types of data have merits and are useful for analyzing different research topics. However, regarding the particular topic that is (for the first time) investigated in the present study—the association between the structure of the global liner shipping network and world trade, liner shipping service routes data are more suitable than container vessels' movement trajectory data. There are two main reasons. First, there in fact exists a unique property of liner shipping practices: world shipping companies always pre-fix liner shipping service routes, including the end-point ports and multiple other ports between them and the vessels deployed; vessels go back and forth on such pre-fixed service routes. The information of this unique property is precisely contained in liner shipping service routes data but is lost in container vessels' movement trajectory data; for the latter type of data, one cannot know the end-point ports of a service route. Indeed, the unique property of pre-fixed routes makes liner shipping service essentially different from other maritime transportation services, in the sense that it does not just simply serve the demand of the international trade but also could potentially influence the demand of international trade between countries. As the economic theory of induced traffic demand²⁵ posits, increased transportation capacity can stimulate extra traffic demand. Second, liner shipping services, though carried out primarily by container vessels, do involve some non-full-container vessels in a few routes. The percent of non-full-container vessels is estimated to be about 13% in the world's total liner ship fleet, according to the latest statistics²⁶. It is true that non-full-container routes are often of less traffic capacity (measured in TEUs) than most full-container routes. However, they cannot be neglected in a global liner shipping network because their presence indeed facilitates the integration of many remote areas (e.g., small island economies) into the global maritime transportation system and thus into global markets. >>

The methods are not advancing the field of network science enough, and the new knowledge about the shipping network does not change much from the pioneering study of Kaluza et al.

REPLY #2: We would like to agree with the second reviewer, but this would make us disagree with the third reviewer who indeed emphasized that gateway-ness analysis and its association to port capacity is a significant contribution, in reference to previous network science tools applied to transportation networks. Gateway-ness, a topological indicator for the first time introduced by the present study, is a valuable innovation that enriches the analytical toolbox in network science, particularly, in cartographical analysis of complex networks.

We considered the article of Kaluza et al (2010) was one pioneering study in the field of maritime shipping network research and was relevant to our study, and thus we already cited it in the previous draft. We had carefully read the article of Kaluza et al (2010), and we are confident that not any of our innovation was presented there. To clarify this point, we now have

added the following paragraph into the “Discussion” section of the revised draft. See revised manuscript, lines 1197-1202, page 35, main text.

<< We emphasize that the topological indicator termed gateway-ness adds an analytical tool in complex network analysis and is applicable to various transportation and spatial networks. For the GLSN, the analysis based on gateway-ness leads to empirical findings on the structural-core organization of the network and its strong association to world trade, as well as the strong association between the gateway-ness and port capacity. None of those findings had ever been available in existing studies of global maritime shipping networks¹⁷⁻¹⁹. >>

There are some unique constraints on this network: 1. The geography of the earth and the fact there are not so many shipping routes in many locations. 2. Unlike the (more well-studied) cargo flight network longer routes are more common, and it is more restricted by geography. 3. The economy is different than the flight network since ships don't primarily go back and forth between the same points. I would recommend this paper for Nature Communications if the author could take unique point like these as the starting point, construct (or choose) analysis methods from this, and discover some unknown fact about the system (and also reproduce known features as proof-of-concepts of their methods).

REPLY #3: Indeed, we agree with the reviewer that the GLSN confronts these unique constraints, and we never wrote the opposite. In consideration of the unique constraints of the GLSN, we constructed analytical methods based on the “gateway-ness” and the relate concept “structural core”. Neither of these two important concepts had ever been mentioned in previous studies of maritime shipping networks, including the one of Kaluza et al (2010). We had offered proof-of-concepts of our methods, as we successfully reproduced a known feature of the global liner shipping system: port capacity is strongly associated to the gateway-ness. Please note that port capacity is in fact a known feature of the global liner shipping network data. Meanwhile, we had discovered an unknown fact about the global liner shipping system: the structure core of the GLSN (whose detection is based on gateway-ness) is strongly associated with world trade (i.e., international trade value of world countries and bilateral trade value between countries). Please note that such trade value is an unknown feature of the global liner shipping system, since it is not included in the shipping network data. However, the reviewer's remarks are properly posed, emphasizing that we need to further clarify in our revised manuscript those points stated above. Therefore, now we also have provided more analysis and discussion for the constraints regarding the number of shipping routes, geographical restriction and the economy of shipping network. Specifically, we first provide explanations for the unique constraints of the GLSN and how the present study's analysis methods are selected and constructed in consideration of those unique constraints. Then, we investigate the influence of the unique constraints of the GLSN on the structural-core organization of the GLSN, using both the previous dataset of the

year 2015 (which we used in the previous manuscript) and a new dataset of the year 2017 (which we recently obtained and added into the revised manuscript). Please see the modification in the newly added sub-section titled “Influence of the constraints on the structural core of the GLSN” (lines 961-1093, pages 30-33, main text) and its related supplementary note 12 (pages 25-37, supplementary text) where we provide supplementary figures and tables to better present the results. See the modified text in the revised manuscript (lines 961-1093, pages 30-33, main text), which is also pasted below.

<< Influence of the constraints on the structural core of the GLSN

The GLSN, as an ensemble of world liner shipping companies’ service networks, is essentially a complex spatial network constrained by the number of liner shipping routes, the physical geography of the earth, and the economy of liner shipping service network configuration. To evaluate the extent to which the structural-core organization of the GLSN was affected by these constraints, we repeated the analysis of structural-core detection in various modified networks where the constraints of the real GLSNs of 2015 and 2017 were modified in different ways and at different levels.

(1) Constraints of the number of shipping routes

There are many ports in the world which have just a few liner shipping routes. Most of them are either located at geographically remote areas or at countries poorly participating in world trade, while some are just feeder ports mainly connected to adjacent hub ports. Although these ports and the related shipping routes are negligible at the global level, they nonetheless are indispensable to the trade development of the regions they serve; they simply facilitate the integration of these regions into the wider global markets. To cover as many ports and geographical regions in the world as possible, we therefore initially constructed a GLSN using all liner shipping routes provided by the database.

Now, to evaluate the extent to which the structural-core organization of the GLSN is influenced by the constraint of the number of shipping routes, we have repeated the analysis in two types of modified GLSNs where we take control for the number of liner shipping service routes adopted. For the first type of modified networks, we increasingly removed between 10%, 20%, 30%, 40%, and 50% of shipping routes with smallest traffic capacity from the original datasets. For the second type of modified networks, in 1,000 iterations, we uniformly randomly removed between 10%, 20%, 30%, 40%, and 50% of shipping routes from the original datasets. The experiments were conducted in the GLSN datasets of 2015 and 2017.

Results show that the detection of a gateway-hub-based structural core was not subject to the number of shipping routes with small traffic capacity, while being sensitive to the number of shipping routes with large traffic capacity. As supplementary Fig. 16 shows, when 10%, 20%, 30%, 40%, and 50% of shipping routes with smallest traffic capacity were removed, the probabilities with which one could detect a structural core by using gateway-ness (B) were estimated to be always close to 100% (yellow bars); probability estimation was based on 1000 repetitions of the Lovain algorithm for modular community partition. For instance, for the GLSN of 2015, such probabilities were 98%, 99%, 99%, 93%, and 100%, respectively. However, when the same percent of shipping routes were uniformly randomly removed, such probabilities would obviously decrease (gray bars); probability estimation was based on 1000 iterations of the

uniform random removal of shipping routes. Nevertheless, they were still significantly greater than their counterparts based on provincial-ness (Z), connector-ness (P), degree (K), or betweenness centrality (BC). Note: in cases of uniform random removal, the topology of the GLSN was inevitably damaged due to loss of some shipping routes with large traffic capacity; in practice, such routes are generally trunk routes linking big ports in different geographical regions and thus are indispensable for the integration of the global liner shipping system.

Nevertheless, the strong association between gateway-ness and port capacity and the strong association between the structural-core ports and global trade hold well, across removal methods and percent of removed routes. In the modified GLSNs, the Pearson correlation coefficients between gateway-ness and port capacity remain at the same level as their counterparts in their respective original networks (supplementary tables 6 and 7 in supplementary note 12); it is also the case for the Pearson correlation coefficients between the GLSN topological indicators and socio-economic indicators of world maritime countries (supplementary Fig. 17 in supplementary note 12).

(2) Geographical constraints

The GLSN is naturally constrained by the geography of the earth, that the longer an inter-port connection is, the more transportation cost it will need. Consequently, long-haul inter-port connections are much fewer than short-haul connections and are available mainly for cargo transportation between a few hub ports of different geographical regions rather than between small ports of different regions. That is, due to such geographical constraints, liner shipping carriers tend to adopt the so-called hub-and-spoke service network configuration that allows them to consolidate cargo in a few hub ports of world regions and thus provide inter-regional transportation between regional hubs via larger vessels. The hub-and-spoke service network configuration proves effective in lowering the overall transport cost of a service network, pursuing the economies of scale³⁷.

Now, to evaluate the extent to which the structural-core organization of the GLSN is influenced by the geographical constraint, we now have repeated the analysis in three modified GLSNs where the geographical constraint of the real GLSNs of 2015 and 2017 is altered based on three different methods. For the first modified network, in 1,000 iterations, we uniformly randomly added into the original GLSN between 10%, 20%, 30%, 40%, and 50% of the amount of edges in an original GLSN. For the second and the third modified networks, in 1000 iterations, we randomly rewired between 10%, 20%, 30%, 40%, and 50% of the edges in an original GLSN based on two different edge-rewiring processes. One was developed by Maslov and Sneppen⁷⁰ (abbreviated to MS null model), where two edges to swap are sampled with a uniform probability. The other was developed by Muscoloni and Cannistraci⁶⁹ (abbreviated to CM null model), where the two probabilities of the two sampled swapping edges are defined separately in function of the adjacent nodes' degree and have an 'inverted tendency': one edge is sampled with a probability directly proportional to the product of the degrees of the adjacent nodes, whereas the other edge with a probability inversely proportional. With all these three methods, the geographical restriction of the real global liner shipping system was indeed altered, as can be seen from either or both of the following two phenomena. One is that long-haul connections would be much more common in modified networks than in the real GLSN; the larger amount of the rewired edges, the greater extent of the modification is (supplementary Figs. 18(b) and 19(b) in supplementary note 12). Another is that in modified networks long-haul connections would

involve many small ports (i.e., low degrees), whereas in the real GLSN they are quite restricted to big ports (supplementary Figs. 18 (a, c-e) and 19(a, c-e) in supplementary note 12).

It turned out that the existence of a gateway-hub-based structural core in such modified GLSNs was apparently much rarer—especially in the ones based on permutation methods of edge rewiring—than that in the original GLSN. Supplementary Fig. 20 presents for each modified GLSN the probabilities with which one could detect a structural core based on the gateway-ness and other topological indicators; in each case, such a detection probability for any given topological indicator was estimated based on 1000 repetitions of the modification process. Moreover, in such modified networks the association between port capacity and gateway-ness would become less outstanding, relative to the association between port capacity and other topological indicators (supplementary tables 8 and 9 in supplementary note 12). Similarly, regarding the association between the GLSN topological indicators and socio-economic indicators of world maritime countries, the performance of structural-core ports (if detected) would be less outstanding, relative to the performance of non-structural-core ports (supplementary Figs. 21 and 22 in supplementary note 12). To sum up, if the geographical constraints of the real GLSN were altered, a structural core of this network and its strong association with world trade would be less likely (if not impossible) to exist. These analyses demonstrate that the geographical constraints indeed play a crucial role in the formation of the structural-core organization of the GLSN.

(3) The constraints of the economy of liner shipping network

The hub-and-spoke network configuration has long been adopted by world liner shipping companies as well as by air lines, for maximizing the economy of their service networks^{71,72}. With such a service network configuration, the economy of a liner shipping service network is obtained by the adoption of pre-fixed liner shipping service routes with multiple ports of call. That is, ships go back and forth among a series of ports along pre-fixed shipping routes, so that cargo between any two ports of a same route can be directly transported via a single ship with a relatively low cost. But the economy of an air transportation network is diffident; an air transportation route typically consists of only two end-point airports, and aircrafts simply fly back and forth between them. Such difference between liner shipping and air transportation is rooted in the different service aims of the two industries: the primary goal of an air company is to deliver air freights as fast as possible, whereas the goal of a liner carrier is to achieve a good trade-off between operating cost and time efficiency. Hence, freight transportation by sea is much cheaper than by air. Considering that the multiple-ports-of-call property of liner shipping service routes is curial for the economy of liner shipping network, we constructed the original GLSN by assigning edges to any two ports of a same shipping route.

Now, to evaluate the extent to which the structural-core organization of the GLSN is influenced by the constraint of the economy of liner shipping network, we have repeated the analysis on the modified GLSNs of 2015 and 2017 where edges are restricted to link only consecutive ports along same shipping routes. In such modified GLSNs the economy of the original GLSNs is altered, in the sense that the multiple-ports-of-call property of liner shipping service routes is neglected.

Results show that in such a modified GLSN one completely cannot find a structural core (in 1000 iterations) by using whichever of the five topological indicators, indicating that the economy of liner shipping network is an important factor which facilitates the structural-core

organization of the GLSN. Specifically, we performed 1000 times the Louvain algorithm for modular community division, repeated the structural-core detection experiment in the obtained 1000 community partitions, and counted the frequency with which a structural core was detected. And this frequency turned out to be 0, for any of the five topological indicators. Moreover, the association of port capacity to gateway-ness is also weakened (supplementary table 10 in supplementary note 12). Note that, when applied to such a modified GLSN, gateway-ness's power in capturing the gateway-hub role of a port in cargo transportation between its host region and the rest of the world in the real global liner shipping system would inevitably be damaged, due to the following reason. To a certain extent, the topology of the modified GLSN—which does not have edges linking any non-consecutive ports of a same shipping route—contains only partial information about the real global liner shipping system, since in reality there exist a considerable amount of cargo transportation between non-consecutive ports of a liner shipping service route via a single vessel. >>

Now the paper tries to find different types of hubs . . . and finds different types of hubs. Wouldn't they do just the same with a different type of data?

REPLY #4: Among other types of hub ports, gateway hubs (which are revealed thanks to the introduction of the new topological indicator termed “gateway-ness”) are indeed unique. A structural core of the GLSN can be detected with a probability of about 92% based on the gateway-ness, whereas such probability of structural core detection would be smaller than 4% if it is based on other topological indicators; please refer to the supplementary figure 9 in the supplementary note 5. Furthermore, there exist a strong association between the gateway-ness and port capacity (table 1 in page 25 of the main text) and also a strong association between the gateway-hub-based structural core and world trade (Fig. 11 in page 28 of the main text).

We now add a discussion for two different types of shipping data that have been used for maritime shipping network research in general, and their suitability for the present study. Please see the modification in the text of the revised manuscript (lines 133-159, page 4, main text); it is also pasted in the end of our reply #1 to the reviewer #2. The two data types are container vessels' movement trajectory data and liner shipping service routes data; both types have merits and are useful for analyzing different research topics. But regarding the particular topic that is (for the first time) investigated in the present study—the association between the structure of the global liner shipping network and world trade, liner shipping service routes data (i.e., the data type adopted in the present study) are more suitable than container vessels' movement trajectory data. Please also note that the cargo vessels' movement trajectory data at the global level are only commercially available from a few companies in the world, and we do not have access to this type of shipping data. We therefore concentrated on the liner shipping service routes data only and formalized specific questions to address the stated topic of the present study.

But we now have provided further analyses for different datasets, to demonstrate the robustness of the key findings on the structural-core organization of the GLSN. Please see the modification in the newly added sub-section titled “Robustness of the structural-core organization of the GLSN across multiple datasets” (pages 29-30, main text) and its related supplementary note 11 (pages 22-25, supplementary text) where we provide supplementary figures and tables to better present the results. See the modified text in the revised manuscript (pages 29-30, main text), which is also pasted below.

<< Robustness of the structural-core organization of the GLSN across multiple datasets

Here we used the following new datasets and replicated the key findings on the structural-core organization of the GLSN—the existence of a (gateway-hub-based) structural core, the strong association between gateway-ness and port capacity, and the strong association between the structural core and world trade. Specifically, we now have used the dataset of world liner shipping service routes in the year 2017 as well as the previous one in the year 2015. In addition, we derived from each dataset 3 sub-datasets that consist of three different types of liner shipping service routes: full-container routes (i.e., routes deployed with full-container vessels; denoted as FC routes), international routes (i.e., routes including ports from at least two countries), full-container international routes (i.e., routes that are deployed with full-container vessels and include ports from at least two countries; denoted as FC International routes). Note that in practice liner shipping service routes are indeed categorized into different types by shipping companies, according to the type of vessels deployed and the number of countries involved. The dataset of GLSN 2015 contains 1622 liner shipping service routes, including 1449 full-container routes, 1472 international routes, and 1316 full-container international routes. The dataset of GLSN 2017 contains 1604 liner shipping service routes, including 1518 full-container routes, 1469 international routes, and 1393 full-container international routes. Detailed results are as follows.

First of all, the detection of a gateway-hub-based structural core proves robust across multiple datasets. For all datasets and sub-datasets, one can detect a structural core of the GLSN based on the topological indicator “gateway-ness” with a probability close to 100% (supplementary Fig. 14 in supplementary note 11). The probability of detection was estimated based on 1000 iterations of the Louvain algorithm; for detailed analyses regarding the robustness of empirical findings on the structural-core organization of the GLSN to the non-detrimental property of the Louvain algorithm in community division, see supplementary note 5. Specifically, we performed 1000 times the Louvain algorithm for modular community division, repeated the structural-core detection experiment in the obtained 1000 community partitions, and counted the frequency with which a structural core was detected by using any given topological indicator.

Moreover, the strong association between gateway-ness and port capacity and strong

association between the structural-core ports and world trade hold well, across datasets. For all datasets and sub-datasets, the Pearson correlation coefficients between gateway-ness and port capacity are at the same level as their respective counterparts based on the previous dataset of world liner shipping service routes in 2015 (supplementary table 5 in supplementary note 11). Likewise, the Pearson correlation coefficients between the GLSN topological indicators and socio-economic indicators of world maritime countries remain at the same level as those results based on the previous dataset of world liner shipping service routes in 2015 (supplementary Fig. 15 in supplementary note 11). >>

They find a structural core, but would it be possible that they wouldn't find a structural core?

REPLY #5: The reply to this question was already in the article, but from the reviewer comment we understand that it was not properly emphasized in the text, and we very much thank the reviewer for helping us with this comment to improve our manuscript. We now have revised the text and introduced an important figure—Fig. 7 in page 20 of the revised manuscript—that explains the algorithm we designed to detect a structural core and to assess its significance in comparison to a null model. As the reviewer can notice from the figure pasted below (which displays the steps of the algorithm to detect the structural core and its significance), we not only detect a (gateway-hub-based) structural core but we build a null model against which we assess whether the probability to detect this structural core in the real GLSN is significantly higher than the probability to detect the same structural core in a null configuration of the same GLSN network. The result is that the structural core of the real GLSN is very significant ($p_value < 0.001$), meaning that the probability to generate the same structural core at random is very low. Please check this Fig. 7 and the related text modification in the revised manuscript (lines 577-590, pages 17-18, main text); they are also pasted below.

In addition, please note the following empirical finding on the structural core detection of the GLSN: the structural core of the GLSN was successfully detected by virtue of gateway-ness (i.e., a topological indicator we introduced for the first time in the present study), instead of existing topological indicators. Specifically, this structural core was detected within gateway-hub ports (defined by gateway-ness), but not within provincial hubs or connector hubs (defined by respective topological indicators that already exist); for detailed results please see Fig. 6, which is available both in the previous manuscript and in the revised one (page 19). Furthermore, in our reply # 8 to the reviewer #2, this empirical finding on the structural core detection of the GLSN has been proved robust to the non-detrimental property of the Louvain algorithm in community division (supplementary figure 9 in supplementary note 5); for more details, please see that reply below. All these results mean that one can hardly detect a structural core of the GLSN if using existing topological indicators such as the ones we had tested in the present study (i.e. provincial-ness, connector-ness, degree, and betweenness). And even in very rare cases a structural core was detected by using those existing topological indicators, it can anyway be

detected by using the gateway-ness. See below the revised manuscript; lines 577-590, pages 17-18, main text.

<< We found a structural core consisting of 37 gateway-hub ports (Fig. 6a). But we could not detect any structural core, based on either of the other two types of hub ports (i.e. provincial hubs and connector hubs): with these two types of hub ports, we did filter out two respective sets of hub ports that form subgraphs of density at least 0.8 (Fig. 6b (Left) and 6c (Left)), but neither of the two sets involve all the individual communities of the GLSN (Fig. 6b (Right) and 6c (Right)). Furthermore, the detected gateway-hub-based structural core of the real GLSN is statistically significant ($p < 0.001$), in the sense that the probability to detect the same structural core in a null configuration of the GLSN network—which is generated by randomly rewiring the links in the real GLSN while keeping the nodes’ degrees—is lower than 0.001 (Fig. 7). (We note that the structural-core organization of the GLSN is related with the modular community structure of this network. Therefore, we further tested the robustness of our empirical findings on the structural-core organization of the GLSN to the non-detrimental property of the Louvain algorithm in community division; see supplementary note 5).

Fig. 7 Statistical significance of a structural core in the real GLSN. A structural core that consists of 37 gateway-hub ports with $B \geq 2.43$ and forms a subgraph of density 0.80 is detected in the real GLSN, **a(Upper Right)**. Links among the structural-core ports in the real GLSN are represented in **b(Upper Right)**, with color indicating these ports’ respective modules. We further evaluated the statistical significance of this structural core in the real GLSN, by accessing whether the probability to detect the same structural core in a null configuration of the GLSN is significantly low. Specifically, two different statistical tests and respective p-values were computed: **a(Lower)**, the probability that in the null model the set of nodes with $B \geq 2.43$ form a subgraph of density larger than 0.80; **b(Lower)**, the probability that in the null model the

same structural-core nodes of the real GLSN form a subgraph of density larger than 0.80. The result, obtained over 1000 repetitions, is that the structural core of the real GLSN is very significant ($p < 0.001$), meaning that the probability to detect the same structural core at random is very low. (See *supplementary note 7* for pseudocode of the algorithm for structural core detection and assessment of its significance.) >>

As I understand it a network without it—so the hubs would not be interconnected—would be quite extreme at least not small world. So then the existence of a structural core is the same as the existence of small-world distance scaling (that was already reported in Kaluza et al.)?

REPLY #6: We have conducted a computational analysis to address this issue and we concluded that the existence of a structural core of the GLSN is not the same as the existence of small-world distance scaling; the analysis and its related results are all presented in *supplementary note 13*. And we paste this note in the end of this reply. However, we agree that the small-world-ness is probably an important factor that contributes to the structural-core organization of the GLSN, but it is not sufficient to create the structural core (as clearly pointed out by our results and discussion reported in the *supplementary note 13*). To carefully explain all the points stated here, we now add into the Discussion section the following comment, lines 1140-1151, page 34, main text.

<< Moreover, thanks to the introduction of the gateway-ness, a gateway-hub-based structural core of the GLSN was uncovered. The underlying mechanisms that govern the formation of such a structural core cannot be answered in the present study. But we notice that our network presents an average shortest path length ($L = 2.67$) that is very close to that of the global container shipping network reported by Kaluza et al.¹⁹ ($L = 2.76$), although the two networks were constructed based on different types of shipping data. For this reason, we comment that the small-world-ness property is probably an important factor that contributes to the existence of the structural core. However, it is not sufficient to create the structural core, and our further analysis proves that the existence of a structural core of the GLSN is not the same as small-world distance scaling (*supplementary note 13*). By definition, the gateway-ness and the associated gateway-hub-based structural core are dependent on the community structure of a network; the small-world-ness can be defined regardless of the community structure. >>

In order to simplify the reviewer #2 revision, we report below the supplementary note 13 (pages 38-39, supplementary information file).

<< Supplementary note 13: Existence of a structural core of the GLSN is not the same as small-world distance scaling

The existence of a structural core of the GLSN is not the same as the existence of small-world distance scaling. By definition, the gateway-ness and the associated gateway-hub-based structural core are dependent on the community structure of a network; so

the best of our knowledge, the small-world-ness can be defined regardless of the community structure. To further clarify the difference between the structural-core organization and the small-world distance scaling, we now have conducted the following experiment. And we show that one cannot find a structural-core in networks simply with a small-world property, regardless of the level of small-world-ness.

We used an equivalent WS (Watts-Strogatz) model¹⁵ of small world generation to generate a family of small-world networks, which keep the number of nodes and the average degree of the real GLSN and as well as the small-world-ness property. Specifically, the WS model begins with a ring of n nodes (here $n = 977$, i.e. the number of nodes in the real GLSN), with each node being connected to its nearest neighbors out to some range K (here $K = 34$, i.e. the average degree of the real GLSN). Each edge in turn is re-wired to a new target node with probability p , ranging from 0 to 1 (supplementary Fig. 23 (a)). Values of $p = 0$ and $p = 1$ give regular and random networks, respectively, with intermediate p values resulting in ‘small-world’ networks.

Then, using the measure of small-world-ness (denoted as ω) developed by Telesford et al¹⁶, we quantified the level of small-world-ness of the rewired network obtained in 1000 iterations under each rewiring probability p , as well as the small-world-ness level of the real GLSN. The coefficient ω of a network is calculated based on the comparison of the characteristic path length L to L_{rand} of an equivalent random network (that preserves the degree distribution of the network under study), and the comparison of the average clustering coefficient C to C_{latt} of an equivalent lattice network; for more details about ω , see *methods*. As measured by ω , the WS model indeed generated for the real GLSN a family of small-world networks that include not only ones with the same level of small-world-ness as the GLSN but also many others with different levels of small-world-ness (supplementary Fig. 23 (b)).

Finally, we repeated the procedure of structural-core detection in the networks generated under each re-wiring probability p . We report that one cannot find a structural-core in any of these networks, regardless of the level of small-world-ness. Such analyses demonstrate that the existence of a structural core of the GLSN is not the same as the small-world distance scaling of the network.

Supplementary Fig. 23 Small-world networks generated by adoption of the Watts-Strogatz (WS) model¹⁵, corresponding to the real GLSN of 2015. Small-world-ness (ω) behaves as required on the WS model of small-world networks. **(a)** The WS model begins with a ring of n nodes (here $n = 977$, i.e. the number of nodes in the real GLSN), each node being connected to its nearest neighbors out to some range K (here $K = 34$, i.e. the average degree of the real GLSN). Each edge in turn is rewired to a new target node with probability p . **(b)** The WS model shows that $p = 0$ gives a regular network, with high clustering but high characteristic path length; $p = 1$ gives a

random network, with low clustering and characteristic path length; intermediate p values give small-world networks with high clustering and low character path length. Squares and circles indicate the normalized clustering coefficient and the normalized characteristic path length of rewired networks, respectively. The ω metric¹⁶ (blue dots) traces the small-world behavior of the networks generated at varying rewiring probabilities (p): a value close to 0 represents a small-world network; a positive value, a network closer to a random one; a negative value, a network closer to a lattice. The red circle indicates the ω value of the real GLSN. Note that for each p , the numerical result obtained on each data point is the mean value over 1,000 realizations. Source data are provided as a Source Data file. >>

[About modularity] First, the Q modularity is the objective measure that the Louvain algorithm maximizes. You write “modularity” for what is really “maximum modularity”. That's OK (always saying “maximum modularity” is awkward too), but at least you should point that out

REPLY #7: Thanks for pointing this out. To make this point clear, we have added the following text into the revised manuscript, in the sub-section titled “Multiscale modular structure” in the “Results” section (lines 441-443, page 12, main text):

<< And we adopted an algorithm of fast unfolding communities in large networks⁵², to search for an optimal partition that maximizes the modularity (Q). >>

sub-section titled “Modularity and community structure” in the “Methods” section (lines 1363-1365, page 39, main text):

<< Under the method of modularity maximization, each partition of a network into several disjoint modules is given a score Q , called the modularity. >>

sub-section titled “Modularity and community structure” in the “Methods” section (lines 1373-1375, page 39, main text):

<< This formulation recasts the problem of identifying modules as a problem of finding the so-called optimal partition, i.e., the partition that maximizes the modularity function Q . >>

[About modularity] The maximum modularity is strongly affected by the number of nodes and links in the network, and also the degree sequence. AFAIK, there is no non-pathologically measure to say which of two network that is most modular. Maybe it is even an ill-defined problem.

REPLY #8: We do understand that the modularity maximization is hard, as posed by many studies (Brandes et al (2008); Good et al (2010)). But this does not invalid the modular community division of the GLSN and its related analysis. The Louvain algorithm is a greedy algorithm that attempts to maximize the modularity of a partition of the network (Blondel et al, 2008). Although modularity maximization is hard and the Louvain algorithm might miss some

small structures in optimizing a partition for the modularity maximization, at large scale this algorithm can be trusted (Fortunato and Hric, 2016). Also, the goal of our work, just like those of many others, is not to find the best community division (or the most modular structure) of the network. Instead, we, in agreement with Newman (2006), regard the community structure detection simply as a data analysis technique used to shed light on the structure of large-scale network data sets. And this technique indeed helps one conduct more advanced analyses, e.g., network cartography. To address the reviewer’s proper concern, we now provide further analyses to show that our findings on the structural-core organization of the GLSN are robust to the non-detrimental property of the Louvain algorithm in community division. Specifically, we ran the Louvain algorithm for 1000 times in the GLSN of 2015 and thus obtained 1000 community partitions, then repeated all the analyses for “structural-core” detection over the 1000 runs, and demonstrated that our findings held well. Detailed analyses and results are presented in supplementary note 5 (pages 13-16, supplementary text). Regarding this, we also have added a brief illustration in the main text of the manuscript (lines 586-593, pages 17-18, main text).

<< (We note that the structural-core organization of the GLSN is related with the modular community structure of this network. Therefore, we further tested the robustness of our empirical findings on the structural-core organization of the GLSN to the non-detrimental property of the Louvain algorithm in community division; see *supplementary note 5*. In brief, we ran the Louvain algorithm for 1000 times in the GLSN of 2015 and thus obtained 1000 community partitions, then repeated all the analyses for “structural-core” detection over the 1000 runs, and demonstrated that our findings on the structural-core organization of the GLSN held well.) >>

[About the analysis of betweenness, closeness] This would be much better if you compared to other networks—null models and similar transportation networks.

REPLY #9: Thanks for reminding us of providing a further comparison analysis. We now have compared to an equivalent random network that preserves nodes’ degree in the real GLSN and as well as to air transportation networks. Additionally, to further confirm the present results about basic topological properties including ports’ betweenness and closeness centrality (which were obtained based on the liner shipping data of 2015), we replicated the results using the liner shipping data of 2017. Modifications are pasted below.

(1) Fig. 2; page 8, main text

Fig. 2 Basic topological properties of the GLSN. In panel (a), complementary cumulative distribution function of degree is reported in log-log scale (dots) and fitted by an exponential function (dash line). Complementary cumulative probability distributions of betweenness centrality (BC) and closeness centrality (CC) for world ports are plotted in semi-log scale in panel (b) and (c), respectively, in comparison to an equivalent random network which exactly keeps the same degree distribution as the original GLSN. The inset in panel (b) reports in log-log scale the betweenness distribution for ports with relatively large BC ($BC \geq 0.001$) (dots), fitted by a power-law function (red dash line) with an exponent 0.740; power-law-ness is tested based on the method of Clauset et al³¹. For an equivalent random network, the betweenness distribution for nodes with $BC \geq 0.001$ decays with an exponent 0.873 (mean across 1000 realizations) (black dash line). The average closeness centrality of the GLSN (i.e., 0.382) is close to that of an equivalent random graph (i.e., 0.440, mean value across 1000 realizations). The bottom panel (d) presents the average port degree $\langle K \rangle$, average shortest path length $\langle L \rangle$, average clustering coefficient $\langle C \rangle$, degree assortativity coefficient and local-community-paradigm correlation ($LCP\text{-}corr$). To confirm these basic topological properties of the GLSN, we repeated the analysis by using the liner shipping data of 2017. The results for the GLSN of 2017 are consistent with the present results for the GLSN of 2015 (*supplementary note 1*). Source data are provided as a Source Data file.

(2) lines 282-305, pages 8-9, main text

As can be seen from the complementary cumulative probability distribution of betweenness centrality for world ports (Fig. 2b), in the GLSN only a few ports are with significantly larger betweenness centralities. The complementary cumulative probability distribution for those ports with relatively large betweenness centralities (i.e. $BC \geq 0.001$) follows a power-law decay, with an exponent 0.740; power-law-ness was confirmed according to the statistical test of Clauset et al³¹.

In an equivalent random network (i.e., a random graph with the same degree distribution as the real network), the distribution of betweennesses would still decay as a power law with a slightly larger exponent value, 0.873 (mean value across 1000 realizations), but the top five largest betweenness values would be much smaller than the counterparts in the real GLSN. A comparison of the two cases shows the existence of some excessively large betweenness values in the GLSN. Central ports of the GLSN mainly concentrate in Europe (e.g., ports of Rotterdam, Antwerp, and Hamburg), Asia (e.g., ports of Singapore, Shanghai, and Hong Kong) and North America (e.g., port of New York-New Jersey and port of Houston), and it can be explained in reference to the wide context of international trade and the industry practice of global liner shipping. As Asia-Europe, Asia-North America, and Europe-North America are the three main inter-regional commodity flows in the world, maritime transport linking these three continents—known as the “East-West trunk line”—dominates world’s inter-regional shipping activities. Therefore, ports located along this trunk line are more likely to develop into transshipment hubs and thus to have relatively large betweenness centralities. In the context of the hub-and-spoke service network design of world liner shipping companies, trunk line services are those linking hub ports of different geographical regions. Such hub-and-spoke routing strategies are also commonly adopted in airlines’ service network configuration. And the betweenness centrality distribution of the global air transport network¹⁶ and that of the Chinese domestic air transport network⁴¹ had also been reported to present a power-law decay.

(3) lines 317-329, page 9, main text

Most of the world ports, though unevenly developed in port technology, have quite good reachability in the GLSN measured by closeness centrality. Closeness centrality is defined as the inverse of the average shortest path length from one node to all other nodes in the network⁴². As shown in Fig. 2c, more than 80 percent ports are of closeness centrality larger than 0.333, indicating that cargo transportation between these ports and others can be realized with transshipment no more than twice, on average. The average closeness centrality of world ports in the GLSN is 0.382. In an equivalent random graph, the average closeness centrality is estimated to be 0.440 (mean value across 1000 realizations). A comparison of closeness centrality values between the empirical and random cases shows that ports’ accessibility in the GLSN is close to what would be expected to appear at random: in both cases cargo can be transported to any port in the world from any given port by transshipment within twice, on average. Hence, it would be natural to expect the GLSN as a whole is self-organized into a small-world structure. In the following section, we then analyzed the small-world-ness of the GLSN.

(4) supplementary note 1; page 1, supplementary text

Supplementary note 1: Basic topological properties of the GLSN of 2017

To confirm the basic topological properties of the GLSN, we repeated the analysis by using the liner shipping data of 2017. The results for the GLSN of 2017 are consistent with the present results for the GLSN of 2015.

Supplementary Fig. 1 Basic topological properties of the GLSN of 2017. In panel (a), complementary cumulative distribution function of degree is reported in log-log scale (dots) and fitted by an exponential function (dash line). Complementary cumulative probability distributions of betweenness centrality (BC) and closeness centrality (CC) for world ports are plotted in semi-log scale in panel (b) and (c), respectively, in comparison to an equivalent random network which exactly keeps the same degree distribution as the original GLSN. The inset in panel (b) reports in log-log scale the betweenness distribution for ports with relatively large BC ($BC \geq 0.001$) (dots), fitted by a power-law function (red dash line) with an exponent 0.726; power-law-ness is tested based on the method of Clauset et al¹. For an equivalent random network, the betweenness distribution for nodes with $BC \geq 0.001$ decays with an exponent 0.847 (mean across 1000 realizations) (black dash line). The average closeness centrality of the GLSN (i.e., 0.380) is close to that of an equivalent random graph (i.e., 0.438, mean value across 1000 realizations). The bottom panel (d) presents the average port degree $\langle K \rangle$, average shortest path length $\langle L \rangle$, average clustering coefficient $\langle C \rangle$, degree assortativity coefficient and local-community-paradigm correlation (LCP-corr). Source data are provided as a Source Data file.

Reviewer #3 (Remarks to the Author):

This manuscript introduces a new network measure, termed "gateway-ness," which characterizes a nodes outside module/community degree of a node. This is a unique measure in that most measures of network topology focus on within module connectivity to characterize properties like the provincial and non-provincial nodes (i.e., network cartography). The authors provide a systematic analysis of the network properties and details how their measure of gateway-ness is unique compared to other related measures like within-module degree and participation coefficient.

This is a well-written manuscript and extremely thorough in its analysis. However, there are some things that the authors should include to strengthen their analysis. Looking at the various measures of centrality, it would be beneficial to also include some analysis of the rich-club coefficients of nodes in the network to further characterize gateway-ness. It would seem this would be a prudent analysis considering Fig. 8a identifies feeder and local connections similar to rich-club analyses. One thing that distinguishes this analysis is the association of port capacity with gateway-ness, but it would seem that it would be helpful to determine the degree that the rich-club coefficient captures or does not capture this property.

REPLY #1: We sincerely thank the reviewer for the positive evaluation of our study and the valuable comment. We now have conducted the suggested analysis of the rich-club coefficients of world ports and the association between port capacity to the rich-club coefficients. Specifically, we considered the unnormalized rich-club coefficient and two normalized versions of it. Additionally, we analyzed the association between the gateway-ness and rich-club coefficient, and measured the extent to which the gateway-hub-based structural core overlapped with the rich club. We have added the related new results into the text of the revised manuscript (lines 789-848, pages 24-27, main text), which are also appended below. As the reviewer will notice, the positive association between the gateway-ness and port capacity is significantly stronger than that between the rich-club coefficients and port capacity. And we have further demonstrated the robustness of this finding (among others) by using another recent shipping dataset of year 2017, in addition to the previous dataset of year 2015; please see the results reported in supplementary table 5 of the supplementary note 11 (page 23, supplementary informational file) and the corresponding sub-section titled "Robustness of the structural-core organization of the GLSN across multiple datasets" (pages 29-30, main text).

The section titled "Structural embeddedness and economic performance of world ports" (lines 789-848, pages 24-27, main text)

<< In addition, we were motivated to further investigate the possibility that a gateway-hub-based structural core and a rich club play a similar role in the high-level topological

organization of a network. The rich-club coefficient φ is a degree-based topological indicator that quantifies the rich-club effect in a network—a tendency for high-degree nodes to be more densely connected among themselves than nodes of a lower degree. $\varphi(k)$ is calculated as the ratio of the total actual number of links to the maximum possible number of links between nodes with a degree larger than k . We adopted the unnormalized version (φ) originally proposed in reference⁶⁷ and two normalized versions (ρ_C and ρ_{CM} , proposed respectively by Colizza et al⁶⁸ and by Cannistraci and Muscoloni⁶⁹) to calculate ports’ rich-club coefficients in the GLSN (see methods on rich club for GLSN in *supplementary note 10*). It turned out that the Pearson correlation coefficient between the rich-club coefficient—should it be normalized or not—and port capacity was significantly lower than the Pearson correlation coefficient between gateway-ness and port capacity (Table 1). But interestingly, we found that the gateway-ness and the rich-club coefficient of ports was correlated: the Pearson correlation coefficient between them ranges from 0.61 to 0.85 in the context of the whole GLSN, depending on whether or not the rich-club coefficient is normalized and the specific normalization procedures. And the detected gateway-hub-based structural core overlapped, to some extent, with the rich club of the GLSN (Fig. 10). As the Fig. 10(a) shows, the Jaccard similarity between the structural-core ports and the rich-club ports ranges from 0.036 to 0.603; the peak (i.e. highest similarity) is reached at the degree value of 135. We notice that the ρ_{CM} also has one (and only one) peak at the degree of 172; indeed, it was here preferred to the other rich-clubness measures because it offers a smoother trend and clear peak (at higher degree than other measures) with which the detection of a compact rich club with high degree threshold is patent. While both the φ and the ρ_C do not display a neat peak value (*supplementary fig. 13 in supplementary note 10*). Furthermore, in Fig. 10b, we display the Venn diagram, which includes the overlapped and the mutually exclusive sets of ports between the ρ_{CM} rich club and the structural core of the GLSN. The overlap is due to the crucial gateway-hub roles of some ports with largest degrees, and it is natural to expect that such ports constitute a part of the structural core. But one shall note that the structural core includes also a few gateway-hub ports with relatively small degrees; these ports, though relatively small at the global level, are of fundamental importance to the development of their own regions in the wider context of global economy.

Table 1 Pearson correlation coefficients between network indicators and port capacity

Network Indicators	GLSN	Port Communities						
		C1	C2	C3	C4	C5	C6	C7
B	0.76**	0.84**	0.91**	0.91**	0.92**	0.80**	0.92**	0.87**
Z	0.50**	0.49**	0.52**	0.59**	0.64**	0.60*	0.55**	0.70**
P	0.38**	0.58**	0.46**	0.52**	0.54**	0.16	0.34**	0.56**

K	0.77**	0.78**	0.84**	0.81**	0.90**	0.99**	0.86**	0.84**
BC	0.68**	0.56**	0.84**	0.83**	0.91**	0.89**	0.89**	0.77**
φ	0.62**	0.79**	0.80**	0.66**	0.69**	1.00**	0.71**	0.83**
ρ_C	0.26**	0.65**	0.58**	0.14	0.28*	0.99**	0.25*	0.62**
ρ_{CM}	0.58**	0.79**	0.81**	0.59**	0.66**	0.99**	0.67**	0.84**

Note: Network indicators B , Z , P , K , and BC denote gateway-ness, provincial-ness, connector-ness, degree, and betweenness centrality, respectively; φ , the unnormalized rich-club coefficient originally proposed in reference⁶⁷; ρ_C and ρ_{CM} , two normalized versions of the rich-club coefficient proposed in reference⁶⁸ and in reference⁶⁹, respectively. Pearson correlation coefficients between network indicators and port capacity are calculated for all ports in the GLSN, as well as separately for ports in individual communities (i.e., C1, C2, C3, C4, C5, C6). C5 is the smallest module that consists of a few ports with similar degree and capacity (and cannot be further divided into submodules). ** indicates p_value < 0.001, * indicates p_value < 0.01.

Fig. 10 Overlap between the rich club and the structural core of the GLSN. In **(a)** Jaccard similarity between the gateway-hub-based structural-core ports and the rich-club ports of the GLSN (black line and black y-axis on the left); and ports' rich-clubness ρ_{CM} (red line and red y-axis on the right), with two vertical dash lines indicating the respective peaks of them. Jaccard similarity index is often used for comparing similarity, dissimilarity, and distance between two data sets, defined as the ratio of the number of shared nodes to the number of distinct nodes in the two data sets. It varies between 0 and 1, indicating a lowest level and a highest level of similarity, respectively. For each degree the black line indicates the average Jaccard similarity between the corresponding rich-club ports and the gateway-hub-based structural-core ports detected in 1,000 iterations. The detection of a gateway-hub-based structural core is based on the modular community division by the Louvain algorithm (Blondel et al (2008))⁵², and this algorithm is non-deterministic. Therefore, we performed 1,000 times this algorithm and repeated the procedure of structural-core detection; in 921 runs, we successfully detected a structural core by using the topological indicator termed "gateway-ness". We have shown in the supplementary note 5 that the structural-core detection is strongly robust over multiple runs of the Louvain algorithm. In **(b)** we show a Venn diagram presenting the rich-club set (in pink)

identified at the degree value of 135 (i.e. the peak point of Jaccard similarity), a structural-core set (in blue) represented in the present study (i.e. the one in Fig. 8b), and their overlapped components. Source data are provided as a Source Data file. >>

Another thing that may not be available, but is there any other data related to the volume of trade between ports? Much of the analysis focuses on the structural network and relates findings to port capacity, but what seems to be missing is identifying if these gateway nodes show unique properties in transportation between the core and local connections. For instance, the authors discuss how Reykjavik provides connections for Iceland and Greenland, but it would be helpful to see if shipping patterns correlate well with the apparent connectivity structure.

REPLY #2: Unluckily, there is an issue associated with the general data availability in this field, and this is beyond our abilities. Indeed, to carry out the above further analysis as suggested by the reviewer #3, one needs to collect the data related to the volume of trade between ports. But unfortunately, we want to remind the reviewer #3 that such data on the actual inter-port trade volume at the global level are not openly available. We are speaking of sensitive data that are associated to the business of world shipping companies and countries' economy. For this reason, to the best of our knowledge, not only the data are not openly available, but we doubt that scientists would be authorized to use them in publishable studies, if not after many years from the actual date of release. Our data are quite recent, and we were unable to access this sensitive information. We doubt that this would be possible for other studies on this topic, but we hope that in the future this issue might find a negotiable solution. Given those points stated above, however, we now have added in the end of the "Discussion" section of the revised manuscript (lines 1233-1239, page 36, main text) the following suggestions for future studies.

<< Another direction for future studies is to collect valuable data (or to provide open access to the ones available) on the actual seaborne trade volume among world ports. These are sensitive data and for the moment were not accessible to the authors. Analyses that include actual seaborne trade volume could help assess the extent to which gateway-hub ports show unique properties in freight transportation volume between local connections and core connections. This would enhance our understanding of how liner shipping patterns in practice correlate with the connectivity structure of the global maritime shipping network. >>

Overall, this is a thorough paper, that provides additional tools for examining the topological structure of a network. Nonetheless, I think it will be improved by further differentiating this measure from other hub measure (e.g., rich-club coefficient).

REPLY #3: We are very grateful for the reviewer's valuable comments and kind encouragement. We have carefully improved our analysis of the topological indicator termed gateway-ness, by

further differentiating this measure from the rich-club coefficient (which is another well-developed hub measure). For detailed replies and related analyses, please see our previous reply #1 to reviewer #3. In addition, as motivated by the valuable comment of the reviewer #2, we now further differentiate the structural-core organization (whose detection is based on the gateway-ness) from the small-world-ness (which has been widely reported in various real-world networks). Specifically, we have conducted an experiment to address that the existence of a structural core of the GLSN is not the same as the existence of small-world distance scaling. For detailed analysis and its related results, please see our previous reply #6 to the reviewer #2.

** See Nature Research's author and referees' website at www.nature.com/authors for information about policies, services and author benefits

Reviewers' comments:

Reviewer #1 (Remarks to the Author):

General comments

This is a nice attempt to run plenty of classic and novel measures on what is called the GLSN in this article. I welcome the changes. But those changes are mostly piled-up paragraphs coming to complement the existing text. What this paper needs is not more text (there is already too much), but more thoughts. The length of my detailed remarks speaks by itself. I recommend major revision as the article cannot be accepted due to major flaws. The analysis remains intuitive rather than driven by scientific rigor. Sometimes the text can't be understood by either maritime or network specialists. The whole motivation remains unclear, and the results not well explained.

Detailed comments

Abstract

- why dealing with "new shipping routes" as the paper deals with existing ones? this calls for a predictive approach which is not the case
- liner shipping network occupies only 16% of the world fleet in DWTs or GRTs, it carries the most valuable goods but has specific network behaviour/structure which should be emphasized

Introduction

- most effective mainly because there is no competition apart from airlines(!) to connect continents
- again new routes: why and how? we understand the short mention about China's recent vision but that is only one case, consider deleting or amplifying?
- the LSI is a composite indicator which scientific relevance should be critically assessed just like others of the kind like the world bank LPI etc.
- up to "coastal areas" the text is relatively fuzzy; consider structuring the arguments in a more rigorous way as ports and maritime trade are important at all levels, from inner city to city and region, hinterland, nation, continent, world ... there are plenty of works telling such stories more or less scientifically sound; the French historian Fernand Braudel is for instance often quoted for his famous book on world-economies etc. but also regional scientists like Fujita and Krugman dealt with ports
- I do NOT understand the central research question, from "what unique ..." to "... world countries". The introduction starts with an emphasis on the economic importance of world shipping but we end up with a central question about network structures; it is about countries but the rest of the paper talks about ports, not even cities or countries; the unit of analysis should be clarified right away otherwise readers may be confused between city, port, country, economic impact, network structure, global trade ...
- exploiting advanced tools but proposing new ones: yes, but without a clear research question it comes out of the blue
- efficient enough to transport goods? that would require crossing trade data and shipping data which is not the case in this article
- unequal conditions: same remark, is it your hypothesis that unequal development influences network structure and vice-versa, or not?
- crucial questions are: are those crucial questions derived from the central question? what is the "real-world economic performance"? do you refer to port performance indicators?? readers are lost at page 2 already

- association to socio-economic statuses: no, it is not the true goal of the paper since it is not directly measured and assessed but only presumed and not even verified...
- rapid development of network science: I would prefer to see this earlier as this is apparently the main focus of the paper i.e. to develop a new measure? again what are "real" complex systems? maritime networks rarely explored: see two Routledge books on the matter in 2015 and 2017 plus dozens of articles published since the late 2000s that would suffice to make a research agenda on what has been done already
- AIS data: the data issue is important indeed, but what does this emergence of "new" data (already used for 20 years) imply for your own research? it seems that you are not studying vessel trajectories but scheduled data, say more what it does imply
- changes occurring after 2008: this should not be in the network science review except if this is better explained in terms of network structure?
- what is a neutral network?
- highly efficient way: not in line with Hu and Xu 2009, please explain - Hu and Xu see the liner shipping network as an intermediary structure between airline and railway due to its spatial constraints
- the three hub measures seem to come out from an empirical investigation without any underlying hypothesis; did you search for these properties with a question in mind or did it happen by chance running any possible partitioning algorithm? there are hundreds of other partitioning or clustering algorithms that would have produced different communities and therefore different intra and extra scores ... think about the z-score of Guimera et al 2005, what do you bring new? how does this add to existing measures of port performance, how do we understand differently port functions in the network compared with other simpler measures such as betweenness centrality or other?
- the structural core is topologically central: hopefully it is, but does it have a geographic logic on top of this obvious fact? or does it contain largest ports wherever they locate? you should say more what mean your results to network and maritime specialists
- please define (again) economic performance of ports lines 110-122; provide support for all these theories ... perhaps too fast to say, be more self-critical
- are unique properties of the GLSN: perhaps not, did you test these properties in other networks? I see that authors affiliations are much transdisciplinary so a comparison with other domains would be absolutely necessary to validate such a claim, and why not, test it on a random network to gain in scientific relevance
- = I therefore suggest rewriting the introduction by taking into account those unclear sentences and make clearer the central question / central results based on initial hypotheses etc.

Results

- figure 1 already provides part of your final results that is, largest ports are more central, which is quite obvious given the paper in 2009 by Deng et al 2009 on the GLSN demonstrating a strong fit between strength (TEU) and degree...
- figure 2: why not providing results for both space L and space P? the average clustering coefficient will certainly be lower in the space-L due to the more evident influence of hubs
- realizations: iterations?
- concentration of large ports in north europe "can be explained" by international trade and global industry practice? consider deleting as this statement is useless. north european ports have risen since the middle age through successive steps and a subtle adjustment between local/global conditions, not just by a fuzzy chance top down
- word ports: world ports? do you mean ports of the world or global-level ports?
- economic small-world-ness: again are you basing your results only on space-P?
- communities not only continental but also political (fig3): i don't see the political factor emerging in the results, but rather, a range effect or maritime circulation effect with land mass bottlenecks... and interoceanic canals constraints especially for Panama, consider reformulating or re-interpreting your

own results

- hubs diversity - "we were able to": the reader has no doubt about your ability to apply many measures; however those are not well justified right from the introduction so that we have the impression that you are searching for something but what
- what difference would it make to calculate an even simpler indicator, the percentage of connections within a certain kilometric range (see Ducruet and Notteboom GaWC paper online) and perhaps results would be the same?
- line 512 correlations are only moderately significant suggesting many outliers (fig4); what about zooming on outliers and explain their emergence by a statistical test searching for heteroskedasticity? otherwise you'll always end up by looking at the big ports; in other words I suggest for a study of functional specialization rather than size
- fig 5 should map outliers instead of main ports, or both
- better explain your choices of standard deviation at least 1.5; density at least 0.8... where do these thresholds come from?
- we note that (line 586) why a parenthesis?
- figure 6 remains hard to read and understand
- figure 9 most interesting analysis in this article - and related text
- figure 10: should further explain why in (b), left-side ports are western and right-side ports are asian, this comes out as very interesting but overlooked; does this relate to different historical legacies and development trajectories as in Lee, Song et al. 2008 (Geoforum)?
- I am not sure about the added value of the analyses after figure 10... perhaps too much information kills information?

Reviewer #2 (Remarks to the Author):

The authors have made a large revision, but the main issues remain: Most of the information in the new measures could be obtained with existing methods and the possible gain in the proposed ones are not clear. The shipping network itself is rather well understood by previous work. This is not at all a bad manuscript, but for a relatively good journal as Nature Communications, the scientific advance is too small for publication I think

(The figures are beautiful!)

In order to facilitate the editor's and reviewers' job during revision, each point-by-point reply reports at the end also the exact text that we modified in the revised main manuscript file (whereas, for reason of space, revised text of supplementary notes should be consulted directly in the supplementary information file).

As for request of the journal, the revised parts both in the present main manuscript file and in the present supplementary information file are highlighted by red character colour; the revised parts both in the previous main manuscript file and in the previous supplementary information file, by blue character colour.

Reviewers' comments:

Reviewer #1 (Remarks to the Author):

General comments

This is a nice attempt to run plenty of classic and novel measures on what is called the GLSN in this article. I welcome the changes. But those changes are mostly piled-up paragraphs coming to complement the existing text. What this paper needs is not more text (there is already too much), but more thoughts. The length of my detailed remarks speaks by itself. I recommend major revision as the article cannot be accepted to to major flaws. The analysis remains intuitive rather than driven by scientific rigor. Sometimes the text can't be understood by either maritime or network specialists. The whole motivation remains unclear, and the results not well explained.

Reply #1: We thank the reviewer for welcoming our previous modifications. We have further improved the manuscript in accordance with the reviewer's further comments, and, as shown in the detailed replies below, each of those comments has been explicitly addressed. For what is written in the reviewer's general comments, we offer the following four additional replies. (1) Thanks for acknowledging the novelty of the proposed node topological measure "gateway-ness". As elucidated in our reply #21 below, we emphasize that the adoption of this novel measure and its two classical counterpart measures (i.e., "provincial-ness" and "connector-ness") to analyze the three respective types of hub ports in the GLSN is driven by the central research question of the present study, rather than driven by any

random attempt to run these measures. (2) We have trimmed down about six pages of the previous manuscript. Meanwhile, we have sharpened our thoughts and better organized the manuscript text, from the initial introduction of the central research question (as carefully explained in the reply #8 below) to the final summary of the main results and the discussion about how they contribute to complex network analysis and how they have deepened our understanding of the structural organization complexity of the GLSN (please see the first three paragraphs of the Discussion section of the revised manuscript, page 28 - 30). (3) We argue that the present study's analyses are scientifically sound; we clarify in the reply #25 below the initial hypothesis and the central research question that drive the whole analyses, together with the central result of these analyses. (4) The whole motivation to conduct the present study is the authors' scientific interest to address the following central research question about the structural organization complexity of the GLSN (lines 83 - 85, pages 2 - 3): "what specific topological properties does the structural organization of the GLSN present, and how the structure of this network is associated with its functional outcomes in realizing the cargo transportation of international trade?" As illustrated in our reply #8 below, the revised Introduction section has also provided the related background knowledge to explain why addressing this question is a worthy scientific pursuit. And, as already illustrated in the above point (2), in the revised manuscript the results have been better summarized and explained.

Detailed comments

Abstract

- why dealing with "new shipping routes" as the paper deals with existing ones? this calls for a predictive approach which is not the case

Reply #2: Thanks for the comment. The previous sentence in the abstract was to introduce part of the general background of the global maritime transportation system. Indeed, to better exploit new international shipping routes, we need to understand the current ones. In order to address the reviewer concern, the sentence has been modified as follow (lines 7 - 9, page 1):

"To better exploit new international shipping routes, we need to understand the current ones and their complex systems association with international trade."

- liner shipping network occupies only 16% of the world fleet in DWTs or GRTs, it carries the most valuable goods but has specific network behaviour/structure which should be emphasized

Reply #3: The suggested background information is now added into the manuscript. Please see the text modification in the Introduction section (lines 39 - 43, pages 1 - 2), which is also pasted below. Please notice that the abstract length (i.e., within 150 characters) is too limited to accommodate the added information.

<< Liner shipping does not account for the largest volume of goods (i.e., about 16%); rather, it carries the most valuable goods, taking up about 60% of the value of goods shipped by sea each year. And unlike any other maritime shipping modes, liner shipping service network configuration of a shipping company is specifically designed to pursue the economies of scale. >>

Introduction

- most effective mainly because there is no competition apart from airlines(!) to connect continents

Reply #4: Yes, we agree and never said the opposite. As one can see, what is written in the corresponding sentence (line 19 - 20, page 1) of the manuscript is not against the reviewer's comment. This sentence is also pasted below.

<< Maritime transport, by far the most cost-effective way (in terms of freight cost) to the mass movement of goods and raw materials across the globe, is the backbone of international trade. >>

- again new routes: why and how? we understand the short mention about China's recent vision but that is only one case, consider deleting or amplifying?

Reply #5: Thanks for the suggestion. But we believe that the phrase is fine as it is, as explained in the reply #2 above; and, for the reason of space, we need to be concise. As we mentioned above in reply #2, with this phrase we emphasize the following point (which is now explicitly reported in the introduction of the revised manuscript; lines 34 - 36, page 1):

<< Indeed, to better exploit new international shipping routes, we need to understand the current ones (that form the global maritime transportation) and their complex systems association with international trade, as is the aim of the present study. >>

- the LSCI is a composite indicator which scientific relevance should be critically assessed just like others of the kind like the world bank LPI etc.

Reply #6: Yes, we agree and never said the opposite. The corresponding sentence in the manuscript writes (line 43 – 46, page 2), “In recognition of the vital role of liner shipping connectivity in determining countries’ accesses to world markets, the UNCTAD has since 2004 launched the project of Liner Shipping Connectivity Index aimed at capturing a country’s level of integration into the global liner shipping network (hereinafter referred to as GLSN).” We now add into the revised manuscript the following modification (lines 48 – 52, page 2) in order to clarify the issue. It is also pasted below.

<< Note that the UNCTAD’s LSCI is mentioned merely as part of the general background knowledge of this study, and it is not involved in any scientific calculation/analysis of this study. Such background knowledge only serves to help general readers to better understand why an in-depth investigation of ports’ connectivity in the global liner shipping network—as the present study has done—is of significant importance. >>

- up to "coastal areas" the text is relatively fuzzy; consider structuring the arguments in a more rigorous way as ports and maritime trade are important at all levels, from inner city to city and region, hinterland, nation, continent, world ... there are plenty of works telling such stories more or less scientifically sound; the French historian Fernand Braudel is for instance often quoted for his famous book on world-economies etc. but also regional scientists like Fujita and Krugman dealt with ports

Reply #7: Thanks for the comments. They make us aware that these two sentences providing some background knowledge of port cities are unnecessary to appear in the Introduction section, since the present study deals with ports instead of port cities. For this reason, these two sentences have been just removed from the revised manuscript.

- I do NOT understand the central research question, from "what unique ..."

to "... world countries". The introduction starts with an emphasis on the economic importance of world shipping but we end up with a central question about network structures; it is about countries but the rest of the paper talks about ports, not even cities or countries; the unit of analysis should be clarified right away otherwise readers may be confused between city, port, country, economic impact, network structure, global trade ...

Reply #8: We have taken full consideration of the reviewer's comments, which help us sharpen the central research question; please check it in the manuscript text (lines 82 – 95, pages 2 - 3), which is also pasted below. As the reviewer can notice, in the revised manuscript we have also rewritten many other parts of the Introduction section in order to better provide the related background knowledge for raising the central research question. Now, the Introduction section starts with an emphasis on the importance of maritime transportation—especially the global liner shipping network (GLSN)—to international trade; see the first two paragraphs of the section. The third paragraph of this section addresses how a perspective of network science can help us better understand the structural organization complexity of the GLSN and its relevance to the network function in realizing the cargo transportation of international trade. With the above background knowledge, the fourth paragraph then raises the central research question as pasted below. We believe that the revised research question (together with other parts of the Introduction section) has clarified to readers that this study focuses on investigating the GLSN structure and its relevance to the GLSN's functional outcome. The unit of the analysis of the GLSN structure is, at the node level, individual ports' structural positions in the GLSN; at the system level, topological properties of the structure of the overall GLSN. The unit of analysis of the GLSN's functional outcomes are, at the node level, individual ports' economic performance (i.e., ports' traffic capacities in liner shipping); at the system level, countries' international trade statuses (i.e., the international trade value of countries and the bilateral trade value between countries).

<< These facts give rise to the necessity of addressing a central research question of the present study: what specific topological properties does the structural organization of the GLSN present, and how the structure of this network is associated with its functional outcomes in realizing the cargo transportation of international trade? Among others, such functional outcomes of great practical importance can be, at the node level, the individual ports' economic performance (i.e., ports' traffic capacities in liner shipping); at the system level, countries' international trade statuses (i.e., the international trade value of countries and the bilateral trade value between countries). Derived from this pivotal research question, there are

many crucial inquires of practical relevance. Regardless of ports' disparity in many aspects such as physical conditions, hinterland economies and socio-political environments, do ports present some similarities of connectivity patterns in the structure of the GLSN? How relevant is a port's connectivity pattern to the port's economic performance (i.e., port's traffic capacity)? How efficient is the structural configuration of the GLSN? How relevant are ports' structural positions in the GLSN to their respective countries' international trade statuses? >>

- exploiting advanced tools but proposing new ones: yes, but without a clear research question it comes out of the blue

Reply #9: Please see our above reply #8, where we have explicitly clarified the research question and, to well address that question, the necessity of exploiting advanced tools of complex network analysis and proposing new ones.

- efficient enough to transport goods? that would require crossing trade data and shipping data which is not the case in this article

Reply #10: We understand that in general the concept "efficiency" (which now appears in line 93 in the Introduction of this revised manuscript) can be operationally defined from different perspectives and, depending on the research interest/focus, there could be various ways to investigate the efficiency of the global liner shipping network (GLSN). The present study is interested in analyzing the (global and local) efficiency of the GLSN configuration based on the method of Latora and Marchiori (2002, 2003), which does not require crossing trade data and shipping data. Such efficiency analysis is a part of the economic small-world property analysis of network structure, as is clearly written in both the previous and present manuscripts; for more detailed information on the method and the related analysis, please see in the "Methods" section the subsection "Economic small-world-ness evaluation" (pages 32 - 33) and in the "Results" section the subsection "Economic small-world-ness" (pages 9 - 10).

However, the reviewer's comment reminds us to revise the corresponding sentence in the introduction, to make it more explicit and more consistent with the points stated above. Please see the revised manuscript (line 93, page 3).

<< How efficient is the structural configuration of the GLSN? >>

- unequal conditions: same remark, is it your hypothesis that unequal development influences network structure and vice-versa, or not?

Reply #11: No, this study does not focus on such hypothesis. To avoid any ambiguity, now this phrase is removed from the corresponding sentence, and the corresponding sentence in line 93 has already been revised (as shown in the previous reply #10).

- crucial questions are: are those crucial questions derived from the central question? what is the "real-world economic performance"? do you refer to port performance indicators?? readers are lost at page 2 already

Reply #12: Yes, these crucial inquiries of practical relevance (lines 89 – 95, page 3) are derived from the central research question (in lines 83 – 88, pages 2 - 3). In this study, the real-world economic performance of ports here refers to the port performance indicator named "ports' traffic capacities", which is ground-truth data derived from the database. This is clarified in the subsection titled "Structural embeddedness and economic performance of ports" in both the previous and present manuscripts (line 731 - 734, page 23), and is now also clarified in the corresponding sentence of the Introduction section (lines 92 – 93, page 3).

<< How relevant is a port's connectivity pattern to the port's economic performance (i.e., port's traffic capacity)? >>

- association to socio-economic statuses: no, it is not the true goal of the paper since it is not directly measured and assessed but only presumed and not even verified...

Reply #13: Thanks, the phrase "countries' socio-economic statuses" throughout the revised manuscript is now replaced by the phrase "countries' international trade statuses"; the latter is directly measured and assessed in the present study. For the related detailed results, please see fig. 11 (page 28) and its corresponding subsection titled "Structural core and international trade" (page 27).

- rapid development of network science: I would prefer to see this earlier as this is apparently the main focus of the paper i.e. to develop a new measure?

Reply #14: Now the related sentences regarding “the rapid development of network science” are moved up (lines 59 – 61, page 2), as suggested by the reviewer. But the focus of the paper is not just to develop a new measure. As clarified in the central research question of the present study (lines 82 – 88, pages 2 - 3), this study focuses on investigating the structural organization complexity of the global liner shipping network and on understanding how it is associated to the network’s functional outcomes in realizing the cargo transportation of international trade. To quantitatively characterize the structural properties of the global liner shipping network, various topological measures and many other computational methods are essentially needed. Among them, the topological indicator termed “gateway-ness”—which is defined as the outside-module degree of a node—is developed for the first time in this study; for more technical details, please see the corresponding part of the “Methods” section (lines 1169 - 1179, page 35). Thanks to the “gateway-ness”, this study indeed uncovered a salient topological property of the overall structure of the GLSN (i.e., the structural-core organization of the GLSN), which proved to be of great importance to the structural integration of the GLSN and of highly relevance to countries’ international trade statuses. All of this is well commented in the Discussion section of the revised manuscript, as the reviewer can check in pages 28 - 30.

again what are "real" complex systems?

Reply #15: The terminology “real complex systems” is quite widely used in complex system/network studies, equivalent to the terminology “real-world complex systems”; please see, for example, Goh and Barabási (2008) and Liu, Slotine, and Barabási (2011). To avoid any ambiguity, we however have replaced it by the “real-world complex systems” throughout the manuscript.

maritime networks rarely explored: see two Routledge books on the matter in 2015 and 2017 plus dozens of articles published since the late 2000s that would suffice to make a research agenda on what has been done already

Reply #16: We understand the reviewer’s comment and we recognize that our previous statement was too general, and thus we have modified this statement to make it appropriate and straightforward. Please see the modified text in the revised manuscript (lines 67 – 77, page 2), which is

also reported underneath this reply. Our explanation for the modification is as follows. As the second reviewer wrote in the general remarks to the authors (in the first round of revision), **"this paper analyzes the network of ports connected by freight ships, and it is a relatively rarely studied type of spatial network."** We do acknowledge the existent studies on maritime networks in general (including those the reviewer has mentioned here and those we already cited in the manuscript), as we already replied to the reviewer's same comment in the last round of revision (i.e., reply #28 in the previous reply letter). However, we stress to the reviewer the following aspects that significantly distinguishes our study from the existent ones on maritime shipping network analysis: our study focuses on quantitatively investigate not only (1) the structural properties of the GLSN but also (2) the relevance of such structural properties to international trade, whereas the existent studies have rarely explored the latter one; and indeed, we uncovered an important structural property of the GLSN (i.e., the gateway-hub structural-core organization of the GLSN) which then proved to be highly relevant to countries' trading statuses, and this property had never been reported by previous studies (to the best of the authors' knowledge). Given these aspects, we are confident that the research focus of our study and its related findings are not (at least not adequately) covered by the research agenda on what has been done by previous studies on maritime shipping network analysis. In order to be more specific, we have added the following sentences in the revised manuscript (lines 67 – 77, page 2).

<< The GLSN, though having been investigated by previous studies from a network perspective^{20-23,31,32}, remains relatively rarely studied by means of innovative and advanced network science methods, which aim to underpin its complex system association with international trade. Indeed, the GLSN is to a certain degree understood by previous work in terms of basic structural properties, yet it has not been well understood how the modular structural organization of this network might be associated to some functional outcomes (such as the international trade indicators of countries) of the complex economic system to which it belongs. Scientific advancements in these directions could provide novel methodological approaches for quantifying the structural dynamics (the connectivity changes that occur along time due to modifications in liner shipping service routes) of the GLSN and their impact on international trade. >>

- AIS data: the data issue is important indeed, but what does this emergence of "new" data (already used for 20 years) imply for your own research? it seems that you are not studying vessel trajectories but scheduled data, say more what it does imply

Reply #17: This study is not studying vessel movement trajectories data, but instead, the liner shipping service routes data. For detailed illustration of the liner shipping service routes data and its suitability for the research topic of the present study (i.e., the association between the structure of the global liner shipping network and international trade), please see the first paragraph of the subsection titled "Data for the GLSN construction" that is available in both the last manuscript and the present one (lines 137 - 155, page 4).

Regarding the importance of data issue, what we emphasize is the importance of the improved availability of high-quality liner shipping data because it provides greater opportunities for the scientific community to deepen our understanding of the structural properties of the global liner shipping system and various issues related with this system. Such marked advance in data quality is thanks to the progress of systematical collection of worldwide liner shipping service routes data by liner shipping industry's third-party databases; for more information on the recent progress of systematical collection, please consult leading databases specialized in liner shipping such as the Alphaliner database (<https://www.alphaliner.com/>). Alphaliner is the one that provided the data for the present study.

But the reviewer's comment reminds us to improve the related sentences to make them more straightforward. Please see the modified text in the revised manuscript (lines 77 - 81, page 2), which is also pasted below.

<< The improved availability of high-quality liner shipping data, thanks to the progress of systematical collection of worldwide liner shipping service routes data by third-party databases in liner shipping industry, now provides greater opportunities for the scientific community to more precisely model the GLSN and thus to deepen our understanding of its self-organization complexity. >>

- changes occurring after 2008: this should not be in the network science review except if this is better explained in terms of network structure?

Reply #18: Thanks for the suggestion. Now it is removed. Please see the related modified text in the reply #17.

- what is a neutral network?

Reply #19: As explained in the corresponding sentence (line 100 - 101, page 3), a neutral network is a network which shows neither assortative mixing nor disassortative mixing (Newman, 2002). Specifically, a network is said to show assortative mixing if the nodes in the network that have many connections tend to be connected to other nodes with many connections, and disassortative mixing if the nodes that have many connections tend to be connected to other nodes with a few connections (Newman, 2002). The definition of neutral network and its related measure are clarified in the "Methods" section's subsection titled "Network assortativity", available in both the previous manuscript and the present one (page 31).

- highly efficient way: not in line with Hu and Xu 2009, please explain - Hu and Xu see the liner shipping network as an intermediary structure between airline and railway due to its spatial constraints

Reply #20: First of all, we had carefully searched in shipping network analysis literature for the recommended work of Hu and Xu (2009), but unfortunately, we could not find it. We found only a reference of Hu and Zhu (2009, titled "Empirical analysis of the worldwide maritime transportation network") that seems to have compared the global liner shipping network to air transportation networks and railway networks, in terms of node degree distribution; see in page 3 of their paper the subsection titled "A. degree distribution and degree correlations". We are unsure if this is the reference that the reviewer would like us to consult, and thus we kindly ask the reviewer to check if the reference information is correct.

Our explanation of the term "highly efficient way" is already in the manuscript, and we now summarize it as follows. In our work, the term "highly efficient way" (appearing in the Introduction section; line 103, page 3) is related with the "economic small-world-ness" topological property of the GLSN, which is carefully analyzed in the subsection titled "Economic small-world-ness" (pages 9 - 10). Briefly, we found that the global liner shipping network is an economic small-world network, i.e., a small-world network that is configured in a relative economic way in the sense that the network configuration supports high global and local efficiency at relatively low wiring cost (Latora & Marchiori (2001, 2003)). For detailed

analysis and the related indicators used for economic small-world-ness analysis (i.e., global efficiency, local efficiency, and wiring cost of a network configuration), please see the corresponding parts of the Results section (lines 341 – 362, pages 9 - 10) and the Methods section (lines 1054 – 1088, pages 32 - 33). As one can notice, in the economic small-world-ness analysis, the spatial constraints of a network is indeed considered; in the definitions of those aforementioned indicators, the real geographical length of a connection is considered. To be more prudent (as reminded by the reviewer's comment), we now modify "highly efficient way" into "efficient way".

- the three hub measures seem to come out from an empirical investigation without any underlying hypothesis; did you search for these properties with a question in mind or did it happen by chance running any possible partitioning algorithm? there are hundreds of other partitioning or clustering algorithms that would have produced different communities and therefore different intra and extra scores ... think about the z-score of Guimera et al 2005, what do you bring new? how does this add do existing measures of port performance, how do we understand differently port functions in the network compared with other simpler measures such as betweenness centrality or other?

Reply #21: As is clarified in the beginning of the subsection titled "Hub diversity" (lines 469 – 472, page 13), the three hub measures are based on cartographical analysis in complex network science (Guimera et al 2005), which hypothesizes that the role of a node can be determined, to a great extent, by its connectivity pattern in the network structure. The three hub measures correspond to three different connectivity patterns of nodes. Definitely, the reason that we analyze such topological properties of individual ports is to address the central research question of the present study (which is already explained in detail in our above reply #8). Certainly, analyzing nodes' topological properties is an indispensable part of work to an in-depth understanding of the structural organization complexity of a complex network. And, as the reviewer might know, community partition algorithms are not designed to achieve this task.

As written in the subsection titled "Hub diversity", the three hub measures are the inside-module degree (Z), the outside-module degree (B), and the participation coefficient (P). Among them, the Z (i.e., z-score) and P were initially proposed by Guimera et al (2005), and the B is introduced for the

first time by the present study. What new the topological indicator B brings as compared with the existent indicators Z and P is already clearly shown by the two important findings of the present study. One is that, at the individual port level, B is much more strongly associated with a port's real-world performance (i.e., a port's traffic capacity in liner shipping measured in TEU) than either Z or P is; see the detailed results presented in table 1 (page 25). Another is that, at the system level, the B indicator indeed leads this study to uncover an important topological property of the overall structure of the GLSN (i.e., the structural-core organization of the GLSN), whereas both indicators Z and P cannot; see the detailed results presented in fig. 6 (page 19). Note: The detected structural core of the GLSN indeed proved to be not only of great importance in supporting long-distance maritime transportation of the GLSN (fig. 9, page 22), but also of high relevance to countries' international trade statuses (fig. 11, page 28). And we also emphasize that this important topological property "the structural-core organization of the GLSN" cannot be revealed by using other existent indicators such as betweenness and degree; please see the detailed results presented in supplementary fig. 9. To conclude, those points stated here have clarified how the B indicator, as compared with existent indicators such as Z, P, degree, and betweenness, has made us understand differently the structural organization complexity of the GLSN and its relevance to the network's functional outcomes.

In addition, we are aware that different community partitions can result in different intra-community and extra-community scores of nodes. Therefore, we had already demonstrated that the important finding on the structural-core organization of the GLSN is robust across multiple runs of the Louvain algorithm in community division; please see the corresponding illustrations in the main manuscript text (lines 566 - 573, pages 17 - 18) and the detailed results in supplementary note 5. We also kindly notice the reviewer that, in the previous round of revision, in our reply #8 to the reviewer #2, this community division issue and the choice of the Louvain algorithm had already been explicitly explained. For the reviewer's information, we quote part of that previous reply as follows:

"..... The Louvain algorithm is a greedy algorithm that attempts to maximize the modularity of a partition of the network (Blondel et al, 2008). Although modularity maximization is hard and the Louvain algorithm might miss some small structures in optimizing a partition for the modularity maximization, at large scale this algorithm can be trusted (Fortunato and Hric, 2016). Also, the goal of our work, just like

those of many others, is not to find the best community division (or the most modular structure) of the network. Instead, we, in agreement with Newman (2006), regard the community structure detection simply as a data analysis technique used to shed light on the structure of large-scale network data sets. And this technique indeed helps one conduct more advanced analyses, e.g., network cartography. To address the reviewer's proper concern, we now provide further analyses to show that our findings on the structural-core organization of the GLSN are robust to the non-detrimental property of the Louvain algorithm in community division."

- the structural core is topologically central: hopefully it is, but does it have a geographic logic on top of this obvious fact? or does it contain largest ports wherever they locate? you should say more what mean your results to network and maritime specialists

Reply #22: The term "the structural core is topologically central" appears in the following sentence of the Introduction section (lines 111 - 114, page 3): "The structural core is topologically central and turns out to be significantly important in supporting long-distance maritime transportation in the world, suggesting a crucial role of the structural-core organization in integrating the individually segregated parts of the GLSN."

Our answer to the first question is yes; the structural core is at the center of the network topology. First, the fig. 8 (page 20) provides an independent confirmation that the structural core is topological central. Briefly, this fig. 8 presents the GLSN in a hyperbolic space by using coalescent embedding (Muscoloni et al, 2017), and in this figure one can clearly notice that the structural core emerges at the center of the GLSN. We kindly note that the coalescent embedding is a model free unsupervised machine learning method, which is executed with only the information of the network topology (i.e., which nodes connect to which nodes) and without any knowledge on the node features. That is, the node location in this visualization is inferred directly from the network topology in an unsupervised manner. Therefore, the fig. 8 offers undoubtable evidence that the detected structural core is central in the network topology, provided that an independent method for network visualization also confirms its topological centrality.

Second, in the subsection titled "Topological centrality of the structural core" (page 21) we further validate that the detected structural core of the GLSN,

as a whole, is at the center of the GLSN topology, by measuring the percentage of shortest paths between any two non-core ports in the GLSN that pass through the structural core by node and by link, respectively. We are confident that these two aspects above are sufficiently sound to evidence that the structural core is topologically central.

But we stress to the reviewer that such a finding is not “obvious”, but rather requires solid scientific analyses (such as the present ones we have made) to uncover it. However, we understand that probably such a finding is intuitively anticipated; as we write in the beginning of this subsection (line 644, page 21), “Intuitively, a valid structural core of a complex network should be topologically central in the entire network.” And it was this intuition that drove us to make the present analysis to validate that the detected structural core of the GLSN is indeed topologically central. On top of being topologically central, the importance of the structural core was also demonstrated by further analysis in which there is geographic logic; please see the subsection titled “Significant importance of core connections in supporting long-distance maritime transportation” (page 21). In this subsection, we carefully looked at the shipping distance for cargo transportation among all the non-core ports in the GLSN, proving the fundamental importance of the core connections (i.e., the connections among the structural-core ports) in such cargo transportation; see detailed results presented in fig. 9 of this subsection.

Our answer to the second question is no; we had already carefully demonstrated that the structural core of the GLSN is different from a collection of the largest ports in the network (i.e., a rich club). Please see fig. 10 in page 26 for detailed results. As one can note from this figure, the structural core does not contain just the largest ports wherever they locate, but rather, it includes also a few ports that are relatively small. These ports, though relatively small at the global level, are of fundamental importance to the development of their own regions in the wider context of global economy. The points stated in the present reply are all explicitly written in the manuscript text (lines 792 – 795, page 25).

To conclude, as illustrated in our answers above, we had already explained in the manuscript text what the corresponding results mean. In addition, in the Discussion section we offer some final remarks (lines 936 – 957, pages 29 - 30) to further explain what it means to maritime specialists and network specialists the novel finding of the gateway-hub structural-core organization of the GLSN. We believe, these remarks offer new insights into

maritime specialists' understanding of the complex structural organization of the GLSN and as well as network specialists' understanding of how the integration process of a complex network with modular structure is facilitated by a few nodes with a gateway-hub role (i.e., a novel topological role of node for the first time introduced in the present study). We now paste below the aforementioned final remarks.

<< The gateway-hub structural core of the GLSN is a manifestation of the structural organization complexity of this network, pertaining to the community structure of the GLSN and a few important ports with gateway-hub roles in each community. First of all, such a finding helps to gain new insights into segregation and integration in the GLSN, improving our understanding of how the GLSN functions to physically realize cargo transportation across the globe. Indeed, modular community structure is one of the most ubiquitous properties of complex networks⁶⁹; many networks of interest in the sciences, including social networks, computer networks, and metabolic and regulatory networks, are found to divide naturally into communities or modules. Segregation and integration in networks with such modular structure has been widely discussed, and elucidating its underlying mechanisms is conjectured to be fundamentally important to a thorough understanding of how complex networked systems fulfill their functions^{70,71}. Specifically, segregation can be seen from the existence of individual modular communities that are defined by high density of connectivity among members of the same community and low density of connections between members of different communities. Integrative processes in networks can be viewed from at least two different perspectives, one based on the efficiency of global communication and another on the ability of the network to integrate distributed information. And the integration of a network relies on network hubs, i.e., a few nodes that are highly mutually connected and highly central in the topological structure of the network. Then one crucial starting step to address such integration processes is to identify the network hubs, which can be defined on a number of criteria. As shown here, in the GLSN such hub nodes are found to be a few gateway-hub ports, constituting a gateway-hub structural core that proves to be greatly important in the integration of the GLSN and highly relevant to the network's functional outcomes. >>

- please define (again) economic performance of ports lines 110-122; provide support for all these theories ... perhaps too fast to say, be more self-critical

Reply #23: In this study, the economic performance of ports refers to ports' traffic capacities, which is ground-truth data derived from the database. The information is available in the subsection titled "Structural embeddedness and economic performance of world ports" in both the previous and present manuscripts (line 731 - 733, page 23), and is now specified in all related sentences throughout the manuscript (in addition to the sentences mentioned here by the reviewer).

We now have revised the sentence "our results provide support for all these theories of structural embeddedness,..." in order to introduce the related findings in a more gradual and explicit way. Please see the modified text of the revised manuscript (lines 120 - 124, page 3), which is also pasted below. The related analyses and results are available in the subsection titled "Structural embeddedness and economic performance of ports" (pages 23 -27).

<< We find that the three different connectivity patterns named gateway-ness, provincial-ness, and connector-ness—which represent three different types of structural embeddedness—are all significantly and positively associated with ports' traffic capacities; among them, the gateway-ness is the one most strongly associated. Such findings provide support for the hypothesis of structural embeddedness. >>

- are unique properties of the GLSN: perhaps not, did you test these properties in other networks? I see that authors affiliations are much transdisciplinary so a comparison with other domains would be absolutely necessary to validate such a claim, and why not, test it on a random network to gain in scientific relevance

Reply #24: Regarding the GLSN's topological properties of gateway-ness and the related gateway-hub structural core, we now have replaced the expression "unique topological properties of the GLSN" by the expression "salient topological properties of the GLSN" (line 128, page 3). Although the authors' affiliations are transdisciplinary, it is not within the present study's focus to compare the global liner shipping network structure with various networks from other domains. But the comparison with random networks was already conducted in the last revision of the manuscript, which was explicitly addressed in our reply #5 to the reviewer #2 in the previous reply letter; for the reviewer's information, we now paste that previous reply below. Please find in the present manuscript the results for significance test

shown in Fig. 7 (page 20) and the related manuscript text (lines 563 - 566, page 17).

"REPLY #5 to the reviewer #2 (in the last revision): The reply to this question was already in the article, but from the reviewer comment we understand that it was not properly emphasized in the text, and we very much thank the reviewer for helping us with this comment to improve our manuscript. We now have revised the text and introduced an important figure—Fig. 7 in page 20 of the revised manuscript—that explains the algorithm we designed to detect a structural core and to assess its significance in comparison to a null model. As the reviewer can notice from the figure pasted below (which displays the steps of the algorithm to detect the structural core and its significance), we not only detect a (gateway-hub) structural core but we build a null model against which we assess whether the probability to detect this structural core in the real GLSN is significantly higher than the probability to detect the same structural core in a null configuration of the same GLSN network. The result is that the structural core of the real GLSN is very significant ($p_value < 0.001$), meaning that the probability to generate the same structural core at random is very low. Please check this Fig. 7 and the related text modification in the revised manuscript (lines 577-590, pages 17-18, main text); they are also pasted below.

In addition, please note the following empirical finding on the structural core detection of the GLSN: the structural core of the GLSN was successfully detected by virtue of gateway-ness (i.e., a topological indicator we introduced for the first time in the present study), instead of existing topological indicators. Specifically, this structural core was detected within gateway-hub ports (defined by gateway-ness), but not within provincial hubs or connector hubs (defined by respective topological indicators that already exist); for detailed results please see Fig. 6, which is available both in the previous manuscript and in the revised one (page 19). Furthermore, in our reply # 8 to the reviewer #2, this empirical finding on the structural core detection of the GLSN has been proved robust to the non-detrimental property of the Louvain algorithm in community division (supplementary figure 9 in supplementary note 5); for more details, please see that reply below. All these results mean that one can hardly detect a structural core

of the GLSN if using existing topological indicators such as the ones we had tested in the present study (i.e. provincial-ness, connector-ness, degree, and betweenness). And even in very rare cases a structural core was detected by using those existing topological indicators, it can anyway be detected by using the gateway-ness. See below the revised manuscript; lines 577-590, pages 17-18, main text.

<< We found a structural core consisting of 37 gateway-hub ports (Fig. 6a). But we could not detect any structural core, based on either of the other two types of hub ports (i.e. provincial hubs and connector hubs): with these two types of hub ports, we did filter out two respective sets of hub ports that form subgraphs of density at least 0.8 (Fig. 6b (Left) and 6c (Left)), but neither of the two sets involve all the individual communities of the GLSN (Fig. 6b (Right) and 6c (Right)). Furthermore, the detected gateway-hub structural core of the real GLSN is statistically significant ($p < 0.001$), in the sense that the probability to detect the same structural core in a null configuration of the GLSN network—which is generated by randomly rewiring the links in the real GLSN while keeping the nodes' degrees—is lower than 0.001 (Fig. 7). (We note that the structural-core organization of the GLSN is related with the modular community structure of this network. Therefore, we further tested the robustness of our empirical findings on the structural-core organization of the GLSN to the non-detrimental property of the Louvain algorithm in community division; see supplementary note 5).

Fig. 7 Statistical significance of a structural core in the real GLSN. A structural core that consists of 37 gateway-hub ports with $B \geq 2.43$ and forms a subgraph of density 0.80 is detected in the real GLSN, a(Upper Right). Links among the structural-core ports in the real GLSN are represented in b(Upper Right), with color indicating these ports' respective modules. We further evaluated the statistical significance of this structural core in the real GLSN, by accessing whether the probability to detect the same structural core in a null configuration of the GLSN is significantly low. Specifically, two different statistical tests and respective p-values were computed: a(Lower), the probability that in the null model the set of nodes with $B \geq 2.43$ form a subgraph of density larger than 0.80; b(Lower), the probability that in the null model the same structural-core nodes of the real GLSN form a subgraph of density larger than 0.80. The result, obtained over 1000 repetitions, is that the structural core of the real GLSN is very significant ($p < 0.001$), meaning that the probability to detect the same structural core at random is very low. (See supplementary note 7 for pseudocode of the algorithm for structural core detection and assessment of its significance.) >>"

= I therefore suggest rewriting the introduction by taking into account those unclear sentences and make clearer the central question / central results based on initial hypotheses etc.

Reply #25: As the reviewer can see from our detailed replies above, we do have fully considered the reviewer's comments and have carefully rewritten the related parts of the Introduction section. These parts include, among others, the initial hypothesis (lines 59 – 64, page 2), central research question (lines 82 – 88, pages 2 - 3), and central results (lines 127 – 129, pages 3 - 4).

Results

- figure 1 already provides part of your final results that is, largest ports are more central, which is quite obvious given the paper in 2009 by Deng et al 2009 on the GLSN demonstrating a strong fit between strength (TEU) and degree...

Reply #26: First, the figure 1b (lower right) merely presents inter-port connections using a hyperbolic layout, and we see ports with larger TEU values are also more central in the hyperbolic geometry underlying the GLSN. But this does not belong to the final results of the present study: the gateway-hub structural-core organization of the GLSN and its strong and positive association to the GLSN's functional outcome in realizing the cargo transportation of international trade for countries. These final results are explicitly presented in the Results section's final subsection titled "Structural core and international trade", which are also indicated in the title of this study.

Second, we are aware of the work of Deng et al (2009) and its finding (i.e. a strong association between port degree and container throughput) that shows that largest ports are more topologically central. But their finding does not affect the importance of our study because their finding is unrelated with our study's final results clarified above and even the focus of our study. Our study does not focus on evaluating which ports are more topological central than others, but on investigating ports' connectivity patterns (i.e., how ports are connected with other ports in the GLSN). Note that degree is an indicator representing the number of connections a node has in the network, regardless of how the node is connected with other nodes. Indeed, thanks to the connectivity pattern named "gateway-ness" (which is for the first time introduced by our study), we achieved the aforementioned final results.

- figure 2: why not providing results for both space I and space p? the

average clustering coefficient will certainly be lower in the space-L due to the more evident influence of hubs

Reply #27: We do understand the reviewer's comment that "the average clustering coefficient will certainly be lower in the space-L", and we had already explicitly written it in both the previous manuscript and the present one (lines 333 – 340, page 9). But the space-P representation method is more suitable (for the reasons that we clarify below) for the purpose of the present study than the space-L, and thus the present study presents all results (including fig. 2) under the space-P representation method. We notice the reviewer that the reason for the choice of space-P method over the space-L method was already explained in the subsection "Data for the GLSN construction" of both the previous manuscript and the present one (lines 177 - 189, page 5), by carefully referring to the earlier studies (i.e., Sen et al (2003) and Sienkiewicz Hołyst (2005)) that initially introduced the concept of space L and P for representing the topology of transport networks. And also, the explanation was provided in the reply # 37 to reviewer #1 in the previous reply letter. For the reviewer's information, we now again paste below the manuscript text regarding the explanation (lines 177 - 189, page 5).

<< To further illustrate such a method of GLSN construction, it is worth mentioning the so-called concept of space L and P ^{12,26} that has been used in many transport network studies to properly represent the topology of transport networks. In space L , a link between two nodes exists if they are consecutive stops on a same route; in space P , a link between two nodes means that there is a single route connecting them. Consequently, the node degree in space- L topology is just the number of directions one can take from a given node, and the node degree in space- P topology indicates the total number of nodes reachable to a given node by using a single route²⁶. Both of the two methods of network topology representation have been adopted in maritime transportation network research^{20,23}, and the methodological choices depend on specific research focuses. The present study emphasizes the fact that cargo between any two ports on a same service route can be transported via a single ship, which is indeed important information contained in liner shipping service routes data. Therefore, a GLSN topology was constructed here in space P . >>

- realizations: iterations?

Reply #28: Yes. The two words are used equivalently in computational experiments. But for unity we have replaced the former by the latter throughout the manuscript.

- concentration of large ports in north europe "can be explained" by international trade and global industry practice? consider deleting as this statement is useless. north european ports have risen since the middle age through successive steps and a subtle adjustment between local/global conditions, not just by a fuzzy chance top down

Reply #29: Thanks for the comment. Now it is deleted.

- word ports: world ports? do you mean ports of the world of global-level ports?

Reply #30: "World ports" means "ports in the world". To avoid any possible ambiguity, we have modified the expression accordingly throughout the paper.

- economic small-world-ness: again are you basing your results only on space-P?

Reply #31: Yes, the GLSN of the present study was constructed based on the network topology representation method so-called space-P, and so are all the analyses (including the economic small-world-ness). For the detailed explanation of the methodological choice of network topology representation, please see above our reply #27.

- communities not only continental but also political (fig3): i don't see the political factor emerging in the results, but rather, a range effect or maritime circulation effect with land mass bottlenecks... and interoceanic canals constraints especially for Panama, consider reformulating or re-interpreting your own results

Reply #32: Revised accordingly. Please see the modified text in the revised manuscript (lines 433 - 435, page 12).

<< However, boundaries of port communities are not simply defined by a continental division but are also related with maritime circulation effect such as land mass bottlenecks and interoceanic canal constraints. >>

- hubs diversity - "we were able to": the reader has no doubt about your ability to apply many measures; however those are not well justified right from the introduction so that we have the impression that you are searching for something but what

Reply #33: This subsection titled "Hub diversity" (pages 13 - 17) investigates three kinds of hub roles of individual ports: gateway hubs, provincial hubs, and connector hubs, which are quantified by three topological measures named outside-module degree (B), inside-module degree (Z), and participation coefficient (P), respectively. These three measures aim to analyze the topological properties of individual ports in the GLSN structure by characterizing three different connectivity patterns of them. Indeed, analyzing nodes' topological properties is an indispensable part of work to an in-depth understanding of the structural organization complexity of a complex network. The necessity of conduct such analysis is justified in the Introduction section of the revised manuscript, taking as a starting point a fundamental theoretical hypothesis in network science (lines 61 - 64, page 2): "A fundamental theoretical hypothesis in network science is the concept that structure matters, positing that the functional outcomes of a complex networked system, at both the system level and the individual node level, depend at least in part on the network structure⁶⁻⁸." In consistence with this initial clarification, now the necessity to conduct such analysis has been better explained in the beginning of this subsection; please see the modified manuscript text (lines 469 - 472, page 13), as is also pasted below.

<< With the division of port communities, we then assigned network roles to individual ports based on the pattern of intra-community and inter-community links. We hypothesized that the role of a node can be determined, to a great extent, by its connectivity pattern⁵³, and thus defined three different types of hub roles (*see Methods*): provincial hubs, gateway hubs, and connector hubs. >>

- what difference would it make to calculate an even simpler indicator, the percentage of connections within a certain kilometric range (see Ducret and Notteboom GaWC paper online) and perhaps results would be the same?

Reply #34: As for the reviewer's request, we have now calculated for each port the recommended indicator (i.e. the percentage of its connections

within a certain kilometeric range), adopting multiple threshold values of the kilometeric range used in the calculation. In the following, we explicitly show that (1), this recommended indicator is not the same as the three topological indicators (i.e., gateway-ness (B), provincial-ness (Z), and connector-ness (P)) that are adopted in the present study to characterize the three different types of hub ports in the GLSN structure (i.e., gateway hubs, provincial hubs, and connector hubs); and (2), the results based on this recommended indicator and the results based on the topological indicators are not the same.

The above point (1) can be seen from the figure 1 attached below, which shows that the recommended indicator is merely weakly correlated with—certainly, not the same as—the three studied topological indicators. Particularly, the Pearson correlation coefficient between the recommended indicator and the B (which is a novel topological indicator introduced by the present study to quantify a node’s connectivity pattern in the network structure), at the highest level, is only about -0.57. Note that, if two indicators were (almost) the same, then the correlation between them would be (close to) 1.

The above point (2) can be seen from, for example, the largely different results regarding their association with ports’ traffic capacities. A port’s traffic capacity refers to the total traffic capacity (measured in TEU) deployed on this port by liner shipping companies in the world, which is ground-truth data derived from the database. As shown in the figure 2 attached below, the Pearson correlation coefficient between this recommended indicator and port capacity, at the highest level, is only about -0.39. Note that the Pearson correlation coefficient between the topological indicator B and port capacity is about 0.76 (reported in table 1, page 25). Obviously, the recommended indicator — calculated under whichever threshold value of kilometeric range—performs much worse than the topological indicator B does at capturing ports’ traffic capacities.

Figure 1 Pearson correlation coefficients between the recommended indicator and the three topological indicators (i.e., B , Z , and P , denoting gateway-ness, provincial-ness, and connector-ness, respectively). We calculated for each port the recommended indicator, using multiple threshold values of kilometric range. Note that ports' B , Z , and P values were calculated with 1000 iterations of the Louvain-algorithm-based community division experiment. Hence, for each data point here, we report the mean Pearson correlation coefficient over 1000 iterations of the community division experiment. Standard errors for all data points are smaller than 0.001.

Figure 2 Pearson correlation coefficients between the recommended indicator and port capacity. Note that we calculated for each port the recommended indicator, using multiple threshold values of kilometric range.

- line 512 correlations are only moderately significant suggesting many outliers (fig4); what about zooming on outliers and explain their emergence

by a statistical test searching for heteroskedasticity? otherwise you'll always end up by looking at the big ports; in other words I suggest for a study of functional specialization rather than size

Reply #35: Thanks for kindly suggesting the statistical test searching for heteroskedasticity and for suggesting a study of port's functional specialization, to make the analyses not end up looking at the big ports (which in the context here refer to hub ports, i.e., ports with large B values, Z values, or P values). They are interesting research directions for us to consider in the future work but are not within the scope of the present study. This study, on the contrary, focuses on identifying hub ports through analyzing ports' connectivity patterns, and on how certain hub ports might form a cohesive structural core that facilitates the structural integration of the GLSN as a whole. Indeed, thanks to the connectivity pattern termed "gateway-ness" (which is for the first time introduced by our study), we achieved the most important finding of this study: a salient topological property of the overall structure of the GLSN, i.e., "the gateway-hub structural-core organization of the GLSN".

Given the explanation above, the reviewer's comments however remind us to trim down the unnecessary information on the correlation coefficients between the three connectivity patterns that is presented in line 512 (of the previous manuscript) and in the corresponding Fig. 4(b). And we agree with the reviewer #1's final comment that "too much information kills information", and thus we have trimmed down the unnecessary information here and also in many other places (which are specified in our reply to the reviewer #1's final comment).

- fig 5 should map outliers instead of main ports, or both

Reply #36: We prefer to keep the fig 5 as it is, for the following reason. The aim of this fig 5 is just to show readers the geographical locations of the three types of hub ports identified in the present study, rather than to map any "outliers" related with any statistical analysis. As we have explained in the above reply #35 the modification of fig. 4 and the corresponding reason, conducting any further statistical analysis or "outlier" analysis would be irrelevant to what the present study focuses.

- better explain your choices of standard deviation at least 1.5; density at least 0.8... where do these thresholds come from?

Reply #37: These two thresholds are considered heuristically in investigating the structural-core organization of the GLSN. To make the following explanations more easily to be understood, please first allow us to quote the related modified manuscript text (in the subsection “Defining structural core”; lines 547 – 556, page 17):

“To quantify the possibly existent phenomena of structural-core organization, a structural core of the GLSN is defined as a set of hub ports that meet the following two criteria: (1) this set should consist of the largest number of the most important hub ports that form a subgraph of high density (i.e. the proportion of actual links in the maximum possible number of links); and (2) Here, we considered a density threshold of 0.8 (which is heuristically a sufficiently high density) and three types of hub ports defined heuristically above: provincial hubs (i.e. ports with inside-module degree at least 1.5 standard deviations above the community mean), gateway hubs (i.e. ports with outside-module degree at least 1.5 standard deviations above the community mean), and connector hubs (i.e. ports with participation coefficient of at least 0.7).”

The heuristic selection of the standard deviation threshold of 1.5 to define the provincial hub ports and the gateway-hub ports in the GLSN structure is due to the following consideration. Such analysis of nodes’ roles in the network structure is well known as network cartographical analysis (Guimera and Amaral, 2005). In the empirical application of cartographical analysis to various complex networks, there does not exist a unified standard deviation threshold in defining provincial hub nodes. Even studies on a same type of network could choose different thresholds, so long as they are reasonable and useful for investigating the network structure; for instance, a threshold of 1 standard deviation above the community mean is used in a brain network study (Meunier et al, 2009), and another threshold of 2.5 standard deviations above the community mean is used in another brain network study (Joyce et al, 2010). Here, we use a threshold of at least 1.5 standard deviations above the community mean to define the provincial-hub ports and the gateway-hub ports, and we think it is a reasonable one. It is important for one to understand that, such definitions are not the final result of our study, but help us restrict the focus to these

hub ports in further investigating whether there emerges, from the collective behaviors of a few non-ordinary ports, some important topological property of the overall structure of the GLSN (such as the one reported in the present study, i.e., the structural-core organization of the GLSN).

The heuristic selection of a density threshold of 0.8 to evaluate whether the ports of interest form a subgraph of high density (i.e. the proportion of actual links in the maximum possible number of links) is due to the following consideration. For simple graphs in general, it shall be fair to claim that nodes that form a graph of a density larger than or equal to 0.8 are densely connected with each other.

To sum up, just like those heuristic rules and thresholds in numerous scientific studies that provide a starting point for the propositions (Walker et al, 2006), the two adopted heuristic thresholds in the present study provide us with a starting point to investigate the proposition of the GLSN's structural property termed "the structural-core organization". And it is worth emphasizing that the detected structural-core organization in the GLSN proved to be statistically significant (fig. 7, page 20); significantly important in supporting long-distance maritime transportation (fig. 9, page 22); and, highly relevant to countries' international trade statuses (fig. 11, page 28). Such results justify the two selected thresholds; they clearly show that the finding of the structural-core organization of the GLSN—obtained under the two adopted heuristic thresholds—is indeed valuable in deepening our understanding of the structural organization complexity of the GLSN and its association with the network's functional outcomes in realizing the cargo transportation of international trade for countries.

- we note that (line 586) why a parenthesis?

Reply #38: Now it has been removed. Previously, we used it as a mark to remind readers that the three sentences inside it are some brief illustrations for the supplementary note 5 included in it.

- figure 6 remains hard to read and understand

Reply #39: Now this figure is further improved, together with its caption. It is available in page 19 of the revised manuscript and is also pasted below.

Fig. 6 Structural core detection of the GLSN. A structural core of the GLSN is defined as a set of hub ports that meet two criteria: (1) this set should consist of the largest number of the most important hub ports that form a subgraph of high density (i.e. here at least 0.8); and (2) this set should contain at least one hub port from each module in the network. In implementation, we first calculated the connection densities among ports with largest values of B , Z , and P , as presented in plots **a(Left)**, **b(Left)**, and **c(Left)**, respectively. Red regions in the parameter spaces of these three plots correspond to the three respective sets of hub ports that meet the criterion (1): ports of $B \geq 2.43$, forming a subgraph of density 0.80; ports of $Z \geq 3.25$, forming a subgraph of density 0.82; and ports of $P \geq 0.78$, forming a subgraph of density 0.85. These three sets of hub ports are

further displayed by big dots in geographical plots **a(Right)**, **b(Right)** and **c(Right)**, as well as their respective distributions over different modules (insets). Then, we could evaluate whether any of the three sets meet the criterion (2). It turned out that only the set of ports of $B \geq 2.43$ met this criterion and thus constituted a structural core of the GLSN, while the other two sets did not. Modules are indicated by color. Pseudocode of the algorithm for structural core detection is available in *supplementary note 7*.

- figure 9 most interesting analysis in this article - and related text

Reply #40: Thanks for recognizing the importance of this scientific contribution in our manuscript.

- figure 10: should further explain why in (b), left-side ports are western and right-side ports are asian, this comes out as very interesting but overlooked; does this relate to different historical legacies and development trajectories as in Lee, Song et al. 2008 (Geoforum)?

Reply #41: Thanks for being interested in the results and for deriving further questions from the results. Unfortunately, in the present study we are unable to give any scientific answer to the derived questions. To answer such questions, it is certainly needed to conduct detailed examination of these ports' characteristics regarding historical legacies and development trajectories (in a way such as Lee, Song et al. 2008 had done for the ports of Singapore and Hong Kong). But analyzing such kinds of port characteristics is beyond the present study's research focus. We kindly notice the reviewer that, at the individual port level, the present study focuses on analyzing ports' topological characteristics in the GLSN structure (i.e., mainly three topological characteristics termed "gateway-ness", "provincial-ness", and "connector-ness"); and on assessing to which extent that ports' topological characteristics are related to their economic performance (here referring to ports' traffic capacities in TEU), regardless of individual ports' difference in all other characteristics such as geographical location, physical conditions, hinterland economies and socio-political environments, historical legacies and development trajectories.

However, we agree that the reviewer's further questions point to an interesting direction for future studies, and we now have mentioned it in the Discussion section. Please see the modification of the manuscript text (lines 986 – 990, page 31), which is also pasted below.

<< To get the most out of network analysis, future studies are encouraged to reveal the underlying governing mechanisms that lead the whole system to such structural properties; and also, to understand how ports with specific topological characteristics become as they are, with respect to ports' other characteristics such as physical conditions, historical legacies and development trajectories. >>

- I am not sure about the added value of the analyses after figure 10... perhaps too much information kills information?

Reply #42: We have seriously considered the reviewer's comments and thus modified the analyses after figure 10 (which are the last two sub-sections of the Results section in the previous manuscript) in the following two ways.

Regarding the sub-section titled "Structural core and international trade", we keep it in the revised manuscript because the results of this subsection are indispensable for addressing the present study's central research question. In short, the results therein exactly show how the reported gateway-hub structural-core organization of the GLSN (which is a salient topological property in the structural organization of the GLSN) is highly relevant to countries' international trade statuses (which reflect the GLSN's functional outcomes in realizing the cargo transportation of international trade for countries). The central research question, as written in the Introduction section (lines 83 – 85, pages 2 - 3) and also carefully explained in our reply #8 above, is the following: "what specific topological properties does the structural organization of the GLSN present, and how the structure of this network is associated with its functional outcomes in realizing the cargo transportation of international trade?" Consistent with the reply here, now we have clarified in the beginning of this sub-section the reason for making such analysis; please see the revised manuscript text (lines 827 – 829, page 27), which is also pasted below.

<< To understand how the structural-core organization of the GLSN may affect the network's functional outcomes in realizing the cargo transportation of international trade, we examined the association between the GLSN topological indicators and international trade statuses of countries. >>

Regarding the additional analyses and results of the sub-section titled "Structural core and constraints of the GLSN" appearing in the previous

manuscript, we now have moved them from the main manuscript file to the supplementary information file (i.e., supplementary notes 11 and 12). The supplementary note 11 (pages 21 – 26, supplementary information file) provides further analyses to demonstrate the robustness of the key findings on the structural-core organization of the GLSN, by using multiple datasets. These analyses were intended to address a concern of reviewer #2 in the previous round of revision; for details, please see our reply #4 to the reviewer #2 in the previous reply letter. The supplementary note 12 (pages 26 – 41, supplementary information file) provides further analyses and discussions for the unique constraints of the GLSN and how the formation of the structural-core organization of the GLSN was influenced by such constraints. These analyses were intended to address another concern of the reviewer #2 in the previous round of revision; for details, please see our reply #3 to the reviewer #2 in the previous reply letter. To conclude, we agree with the reviewer #2 that addressing these concerns is worth the effort, and thus we already did it. But we also agree with the reviewer #1 that, to keep the main information of the present study as clear as possible, these additional analyses and results are better not put in the main manuscript; and thus, we now move them to the supplementary notes 11 and 12.

Reviewer #2 (Remarks to the Author):

The authors have made a large revision, but the main issues remain: Most of the information in the new measures could be obtained with existing methods and the possible gain in the proposed ones are not clear. The shipping network itself is rather well understood by previous work. This is not at all a bad manuscript, but for a relatively good journal as Nature Communications, the scientific advance is too small for publication I think

Reply #1: We thank the reviewer for acknowledging the quality of our manuscript. As the reviewer can see from our above replies to the reviewer #1, we have further improved the manuscript with careful modifications throughout the Introduction, Results, and Discussion sections. Based on the revised manuscript, we now reply to the reviewer #2' concerns with the following three points. Briefly, the point (1) clarifies that the information in the new measures cannot be obtained with any existing methods, the point (2) clarifies that the gains of the new measures lie in their contributions to not only the global liner shipping network but also network analysis in general, and the point (3) clarifies that the global liner shipping network

has not been well understood by previous work. We hope, with the further improved manuscript and the following three clarification points, to convince the reviewer #2 that the scientific advance of the present study is sufficient for publication in Nature Communications.

(1) The information in the new measures cannot be obtained with any existing methods. The proposed node topological indicator termed gateway-ness (B) is a novel measure for network cartographical analysis, as compared with its two classical counterpart indicators—provincial-ness (Z) and connector-ness (P) that are proposed by Guimera and Amaral (2005). Definitions of these three indicators are provided in the Methods section's sub-section titled "Defining various hub ports in the GLSN". What new the topological indicator B brings, as compared with the existent indicators Z and P, is clearly shown by the two important findings of the present study. One is that, at the individual port level, B is much more strongly associated with a port's economic performance (i.e., a port's traffic capacity in liner shipping measured in TEU) than either Z or P is; see the detailed results presented in table 1 (page 25). Another is that, at the system level, the B indicator indeed leads this study to uncover a salient topological property of the overall structure of the GLSN (i.e., the structural-core organization of the GLSN), whereas both indicators Z and P cannot; see the detailed results presented in fig. 6 (page 19). Please note: The detected structural core of the GLSN proved to be not only of great importance in supporting long-distance maritime transportation of the GLSN (fig. 9, page 22), but also of high relevance to countries' international trade statuses (fig. 11, page 28). And also, we emphasize that this important topological property "the structural-core organization of the GLSN" cannot be revealed by using other existent indicators such as betweenness and degree; please see the detailed results presented in supplementary fig. 9. Those points stated here have clarified how the B indicator, as compared with existent indicators such as Z, P, degree, and betweenness, has made us understand differently the structural organization complexity of the GLSN and its relevance to the network's functional outcomes (in realizing the cargo transportation of international trade for countries).

(2) The gains of the proposed novel indicator B and the unveiling of the related topological property "gateway-hub structural-core organization of the GLSN" are now further clarified in the revised manuscript. In short, the B indicator is a valuable new tool for network cartographical analysis, contributing to complex network science in general rather than to maritime transportation network research only; please see our careful discussion

about this contribution in the second paragraph of the Discussion section in the revised manuscript (lines 919 – 935, page 29), which we also paste below. And the unveiling of the related topological property “gateway-hub structural-core organization of the GLSN” helps to gain new insights into segregation and integration in the GLSN, contributing to our understanding of how the GLSN functions to physically realize cargo transportation across the globe; please see our careful discussion about this contribution in the third paragraph of the Discussion section in the revised manuscript (lines 936 – 962, pages 29 - 30), which we also paste below.

Modifications in the revised manuscript (lines 919 – 935, page 29):

<< From the point of view of network analysis, our study highlights the importance of topological role investigation of real-world networked systems to a better understanding of the interface between network structure and functional dynamics^{5,19,53,65,66}, and provides valuable empirical evidences for a crucial theoretical conjunction of network cartography that different networks seem to be formed by nodes with network-specific roles. The pioneer scheme of network role assignment introduced by Guimerà and Amaral^{53,65} classifies nodes for modular networks into topological roles based on two statistics (i.e. inside-module degree and participation coefficient), and has witnessed a great success in its application to various networks (e.g. protein-protein interaction networks⁶⁷ and brain networks⁶⁸). Unlike these networks, however, the GLSN is primarily affected by neither the ports with high inside-module degrees nor the ports with high participation coefficients. Rather, it is highly subject to ports with high outside-module degrees, which we term gateway hubs. Our work suggests that network-specific roles could vary across different types of networks, thereby underscoring the importance of appropriately using and defining such roles when a specific network is under investigation. As we have illustrated, the gateway-hub role with its formal definition provides one example that is worth further attention, which might also be applicable to other kinds of spatial networks. Indeed, thanks to the introduction of the gateway-ness, a gateway-hub structural core of the GLSN was uncovered. >>

Modifications in the revised manuscript (lines 936 – 962, pages 29 - 30):

<< The gateway-hub structural core of the GLSN is a manifestation of the structural organization complexity of this network, pertaining to the community structure of the GLSN and a few important ports with gateway-hub roles in each community. First of all, such a finding helps to gain new

insights into segregation and integration in the GLSN, improving our understanding of how the GLSN functions to physically realize cargo transportation across the globe. Indeed, modular community structure is one of the most ubiquitous properties of complex networks⁶⁹; many networks of interest in the sciences, including social networks, computer networks, and metabolic and regulatory networks, are found to divide naturally into communities or modules. Segregation and integration in networks with such modular structure has been widely discussed, and elucidating its underlying mechanisms is conjectured to be fundamentally important to a thorough understanding of how complex networked systems fulfill their functions^{70,71}. Specifically, segregation can be seen from the existence of individual modular communities that are defined by high density of connectivity among members of the same community and low density of connections between members of different communities. Integrative processes in networks can be viewed from at least two different perspectives, one based on the efficiency of global communication and another on the ability of the network to integrate distributed information. And the integration of a network relies on network hubs, i.e., a few nodes that are highly mutually connected and highly central in the topological structure of the network. Then one crucial starting step to address such integration processes is to identify the network hubs, which can be defined on a number of criteria. As shown here, in the GLSN such hub nodes are found to be a few gateway-hub ports, constituting a gateway-hub structural core that proves to be greatly important in the integration of the GLSN and highly relevant to the network's functional outcomes. Second, as ports are not isolated entities, issues pertaining to port development can be better understood with reference to ports' positions in the GLSN structure. The analysis of ports' gateway-hub roles in our study offers new insights into the nature of relationships between ports within specific regions and in different regions, which may have potential benefits to policy considerations for port development at both the individual port level and the national level. >>

(3) The GLSN is to a certain degree understood by previous work in terms of basic structural properties, yet it has not been well understood how the modular structural organization of this network might be associated to some functional outcomes (such as the international trade indicators of countries) of the complex economic system to which it belongs. Illustrations are as follows. First, existent studies on maritime shipping networks are relatively rare, not to mention studies on the specific type of global liner shipping

network (GLSN). This fact was also acknowledged by the second reviewer, as the reviewer wrote in the general remarks to the authors (in the previous round of revision), "this paper analyzes the network of ports connected by freight ships, and it is a relatively rarely studied type of spatial network." Second, and more importantly, those a few existent studies on the global liner shipping network focused on only the structural properties of the network itself, leaving unexplored the crucial question that how the structural organization of this network is relevant to the network function in realizing the cargo transportation of international trade for countries. This question is exactly the central research question of the present study (lines 82 – 88, pages 2 - 3). We have explained why addressing this question is crucial to understanding how the GLSN functions to physically support the international trade, in the third paragraph of the Introduction section (lines 53 – 81, page 2). Third, we emphasize that the present study properly addressed the proposed crucial research question, using the most suitable type of shipping data (i.e., liner shipping service routes data). Briefly, previous studies had used two types of shipping data to construct liner shipping networks: container vessels' movement trajectory data and liner shipping service routes data. Due to the unique property of liner shipping—vessels go back and forth on pre-fixed service routes, liner shipping service routes data are more suitable than container vessels' movement trajectory data for investigating the association between the structure of the GLSN and international trade; detailed explanations are available in the first paragraph of the subsection titled "Data for the GLSN construction" (lines 137 – 155, page 4).

(The figures are beautiful!)

Reply #2: Thanks for the compliment.

Reviewers' comments:

Reviewer #1 (Remarks to the Author):

General comments

This article is improving in the way that it adds more explanation on what it is performing. However, what remains lacking is a clear research question, still. The search for a core structure is not backed by an industry-specific question nor by a methodological question. Or, such questions are sufficiently developed to catch the attention of the reader, who goes through the text rediscovering existing knowledge. Top ports are top ports, however it is measured. The core-periphery structure is not well related with the international trade literature although it potentially would benefit well to it and to the transport literature - the same would apply to the literature on network partitioning, but again, the authors go too quickly in their own direction without sufficiently explaining their choices. The article should better fit as a slight addition (or confirmation) to existing knowledge in a transport journal.

Detailed comments

Introduction

- should refer to important works on international trade and maritime transport such as Clark, Dollar & Micco 2004, Limao and Venables 2001
- what means "total goods loaded on ships"?
- "strong correlation": between which variables and by which fit?
- the current ones: note that contemporary shipping networks have been analyzed through numerous works already, even from the 19th century (see Ducruet, Tavasszy, Hu and Zhu 2009, Deng et al. 2009, etc.)
- economies of scale: should be explained briefly for non-specialists
- from "Note that ..." to "importance." is not useful, it looks like a reply to reviewers
- the mention of LSCI should be brief and only for the sake of supporting your arguments
- arguments for a new analysis of the GLSN are somewhat weak, since industry data is less precise than vessel movement data (it should be said to what extent is the data used in your paper is original and how);
- and about previous studies, the fact that the current paper uses novel approaches is good, but again, the purpose is a description of the network structure, not of the impact on international trade, which would need much more data and other methods.
- the introduction is too long overall
- the central research question can't be understood; it again makes the link with international trade, economic importance, countries, that do not appear in the subsequent analysis only looking at nodes and links

Results

- note that vessel movement data may also be more precise than schedule data as it shows the real circulation pattern, with also feeder and shortsea, sometimes sea-river
- previous studies also used AIS data, which should be emphasized, see Kaluza et al. 2010 for instance and many others
- figure 1 does not show the difference between space L and space P; a further discussion about the implication of choosing one of them is needed
- ports with larger TEU are more central in the hyperbolic layout: what does this mean for research?
- influence of shipping on trade: this is NOT the purpose of this study, but it is regularly claimed to be so
- assortativity may be more meaningful if including feeder and shortsea flows but it is not the case in

the proposed paper, consider clarifying

- regional port systems: see the work of Ducruet and Notteboom 2012 highlighting nodal regions where neighboring ports are centralized upon one main hub
- why not providing a comparative analysis in the figures for 2015 and 2017 instead of only in the text of figure 2?
- it would be interesting to compare the small world ness between space L and space P, as space L better reflects the hub structure of the GLSN compared with space P (which better reflects trade structure), as proposed by the authors lines 338-340
- same for efficiency, space P is often higher than space L; could this be emphasized?
- not well explained how the distance of links was integrated in the calculation of efficiency
- economic organization: should discuss the non-planar dimension of the GLSN where nodes may have many links i.e. be less constrained than planar nets
- same for LPC-corr analysis, planar vs. non-planar discussion is missing
- "by socio-economic factors": which ones? lines 413-415

Multiscale modularity

- the "move" to the search for community structure is somewhat mechanical and without clear explanation
- see Ducruet and Zaidi 2012 for the search of such structures using another algorithm in the GLSN, namely topological decomposition (and a comparison between space L and space P)
- the choice of the methodology is not sufficiently explained compared to the myriad of other possible ones
- land mass bottlenecks: it should be more discussed in light of the results obtained
- the economic, logistical and geopolitical division of the world is not well explained using figure 3, where the pattern looks very interesting; what does it tell us about world economy and world shipping?
- hubs diversity: again, the search for port functions comes a bit out of the blue
- why 1.5 above the SD? why higher than 0.7? needs an explanation of threshold choice
- figure 5: it is not clear whether ports cumulate two or more functions; whether specific ports are already identified as "hubs" in the transport literature (e.g. Singapore), etc. a thorough interpretation is needed
- without an interpretation we move on to the next section, which is inappropriate

Gateway-hub structural core

- seems to be formed: this goes with a lack of a proper research question
- why 0.8??
- figures 6 and 8 are not well explained; the identified main hubs could simply large ports, regardless of their function; needs further analysis
- lines 644 and after: are structural-core ports the ones with large betweenness (and/or a low clustering coefficient)? all the different analyses seem to conclude this simple fact
- defining feeder connections between these values mean that this category includes shortsea and other non feeder ones; the local scale (as said lines 702-704) is more reflective of "hub-feeder" activity...

Structural embeddedness and economic performance of ports

- economic performance: not sure about its definition and its place in this paper
- theorising: the first paragraph seems to refer to methods in sociology like structural equivalence or other
- lines 760-763: conclusions about port policy should also refer to shipping lines' port choice and port competition
- we found that ... (line 777): such results look like new explorations out of the blue, not based on a

clear question and not sufficiently interpreted

- figure 10 shows the major ports in the world; again, why not calculating the simple correlation with throughput (TEUs) to check whether the whole analysis is finally providing this simple evidence?
- correlation with trade data on countries and pairs (and fig 11): should refer to Guerrero et al 2015 (maritime networks book) who use a gravity model and restrain the analysis to trade of containerized products (needs a methodological introduction)

Discussion

- relation between network structure and functional dynamics: not clear
- the discovered structure may simply be a group of the largest ports
- lack of longitudinal data: see studies of maritime networks by Ducruet over 120 years 1890-2010 or 30 years 1977-2008

Reviewer #1 (Remarks to the Author):

General comments

This article is improving in the way that it adds more explanation on what it is performing. However, what remains lacking is a clear research question, still. The search for a core structure is not backed by an industry-specific question nor by a methodological question. Or, such questions are sufficiently developed to catch the attention of the reader, who goes through the text rediscovering existing knowledge. Top ports are top ports, however it is measured. The core-periphery structure is not well related with the international trade literature although it potentially would benefit well to it and to the transport literature - the same would apply to the literature on network partitioning, but again, the authors go too quickly in their own direction without sufficiently explaining their choices. The article should better fit as a slight addition (or confirmation) to existing knowledge in a transport journal.

REPLY #1: We thank the reviewer for acknowledging the improvement of the last revised version of our manuscript. We now have improved the manuscript with careful modifications throughout the Introduction, Results, and Discussion sections. In particular, we have further clarified the present study's research question and have better justified its contribution. However, we interpret the reviewer's sentence "the article should better fit as a slight addition (or confirmation) to existing knowledge in a transport journal" as a 'spur' to emphasize our inventions and findings, which are indubitably the proposal of gateway-hub structural-core methodology and the discovery of its strong and positive association with international trade. Not any previous study of maritime or other transportation networks has ever arrived to this discovery. In the following, we provide a point-by-point rebuttal of the reviewer's criticisms.

"However, what remains lacking is a clear research question, still. The search for a core structure is not backed by an industry-specific question nor by a methodological question."

We now have further clarified our research question, for which we have elucidated why the search for a structural-core organization of the GLSN (here called by the reviewer as "a core structure") is highly relevant to a specific question regarding the liner shipping industry and to a methodological question regarding the complex network science. Please see the further clarified research question and its related explanation in the Introduction section of the present revised manuscript (lines 76 – 96, page 2), which are also pasted below.

<< These facts give rise to the necessity of addressing a central research question of the present study: what specific topological properties does the structural organization of the GLSN present,

and how the structure of this network is associated with its functional outcomes in realizing the cargo transportation of international trade? Among topological properties, the exploration of the structural-core organization is crucial because it refers to the emergence of certain hub ports that (as a cohesive core) play an important role in the structural integration of the entire network. This corresponds to a specific question in liner shipping industry: which ports are the most important hubs in the GLSN structure, because they form a core such that cargo transportation between any ports in the network can be achieved efficiently by means of them? Such question is practically relevant because one of the most important issues in individual companies' liner shipping service network design is to strategically pre-choose hub ports¹⁷⁻¹⁹. Methodologically, we pursue this particular research interest through investigating the modular community structure of the GLSN, since modular community structure is one of the most ubiquitous properties of complex networks²⁰ that can influence their function and structural core organization. Indeed, we bring forward an analysis to elucidate how the structural integration of a modularly segregated network is achieved via network hubs^{21,22} that resembles a core. Then, we explore the association between a new network structural measure (that we propose and term gateway-ness) and the GLSN's two functional outcomes of practical importance: individual ports' economic performance (i.e., ports' traffic capacities in liner shipping), which is at the node level; countries' international trade statuses (i.e., the international trade value of countries and the bilateral trade value between countries), which is at the system level. >>

The reason why it is important to address such research focus (i.e., the association between structural organization complexity of the GLSN and its functional outcomes in realizing the international trade for countries) is already provided in the Introduction section. It is now revised as follows (lines 57 – 66, pages 1 – 2):

<< The global liner shipping network (GLSN) is a self-organized complex transportation network, as a result of world's individual liner shipping companies' service networks that widely pursue the economies of scale (i.e., the adoption of large container vessels to decrease the shipping cost at sea per cargo unit). The function of the GLSN in supporting international trade is to transport containerized cargoes between countries, which it does by shipping cargoes from port to port across the GLSN until they reach their intended destinations. Certainly, the structure of the GLSN will affect how it accomplishes this function. A fundamental theoretical hypothesis in network science is the concept that structure matters, positing that the functional outcomes of a complex networked system, at both the system level and the individual node level, depend at least in part on the network structure⁶⁻¹⁰. >>

“Or, such questions are sufficiently developed to catch the attention of the reader, who goes through the text rediscovering existing knowledge. Top ports are top ports, however it is measured.”

First, the proposed node topological indicator termed gateway-ness (B) is a novel measure for network cartographical analysis, as compared with its two classical counterpart indicators—provincial-ness (Z) and connector-ness (P) that are proposed by Guimera and Amaral (2005). Definitions of these three indicators are provided in the Methods section’s sub-section titled “Defining various hub ports in the GLSN”.

Second, the new knowledge that the topological indicator B brings, as compared with the existent indicators Z and P, is clearly indicated in the two important findings of the present study. One is that, at the individual port level, B is much more strongly associated with port’s economic performance (i.e., a port’s traffic capacity in liner shipping measured in TEU) than either Z or P is; see the detailed results presented in table 1 (page 16). Another is that, at the system level, the B indicator indeed leads this study to uncover a salient topological property of the overall structure of the GLSN (i.e., the structural-core organization of the GLSN), whereas both indicators Z and P cannot; see the detailed results presented in fig. 6 (page 11). Please note: The detected structural core of the GLSN proved to be not only of great importance in supporting long-distance maritime transportation of the GLSN (fig. 8, page 14), but also of high relevance to countries’ international trade statuses (fig. 9, page 18). And also, we emphasize that this important topological property “the structural-core organization of the GLSN” cannot be revealed by using other existent indicators such as betweenness and degree; please see the detailed results presented in supplementary fig. 8 and its related explanation in the supplementary note 5.

Third, the structural-core ports (here called by the reviewer as top ports) revealed in the present study are not the same as the top ports defined by any existing measures. In our reply #38 to the corresponding detailed comment of reviewer #1, we explicitly show the difference between the structural-core ports and the ones with largest betweenness, and the difference between the structural-core ports and the ones with lowest clustering coefficient. Based on such facts, we emphasize to the reviewer that no existing studies had ever reported the structural-core ports (as revealed in the present study, thanks to the new topological measure termed “gateway-ness”) or any concept of structural core (as proposed in the present study), not to mention the main finding of our paper: the gateway-ness and the related gateway-hub structural core are salient topological properties of the GLSN, which prove highly relevant to the functional outcomes of this network (i.e., traffic capacity of ports and the international trade statuses of countries).

“The core-periphery structure is not well related with the international trade literature although it potentially would benefit well to it and to the transport literature - the same would apply to the literature on network partitioning, but again, the authors go too quickly in their own direction without sufficiently explaining their choices.”

First, the finding on the structural-core organization of the GLSN is highly related with, and contributes to, international trade literature, because understanding countries’ international trade statuses (i.e., the international trade value of countries and the bilateral trade value between

countries) are among those most common and exciting research pursuits in a large body of international trade literature (see our references such as Limao and Venables (2001), Clark et al. (2004), Guerrero et al. (2015), and Bernhofen et al. (2016)). In the present study, we have shown how countries' international trade statuses (i.e., the international trade value of countries and the bilateral trade value between countries) are strongly and positively associated with the structural-core organization of the GLSN; see results in fig. 9 (page 18) and the corresponding analysis in the subsection titled "Structural core and international trade" (pages 16 – 18). Now, the contribution of this finding on the international trade literature is clarified in the Discussion section of the present revised manuscript (lines 565 – 572, page 18) and is also pasted below.

<< In particular, the finding on the strong and positive association between the structural-core organization of the GLSN and countries' international trade statuses, for the first time, quantifies the relatedness between the network structure of a global maritime transportation system and international trade. Such quantification contributes to wider literature on maritime economics and international trade that pursues understanding how and the extent to which the factors regarding global maritime transportation systems can explain, and may influence, trade between countries and/or individual countries' trade with the rest of the world^{3-5,53}. >>

Second, the finding on the (gateway-hub) structural-core organization of the GLSN is highly related with, and contributes to, transport literature. Detailed explanations are now added into the Discussion section of the present revised manuscript (lines 573 – 589, page 19), which are also pasted below.

<< From the perspective of maritime transportation practices, the finding that a few gateway-hub ports form a cohesive structural core of the entire GLSN and support long-distance maritime transportation across the network improves our understanding of the structural organization complexity of the GLSN. The GLSN emerges as the union of individual companies' liner shipping service networks. In a competitive market environment, a good knowledge of the gateway-hub structure-core of the GLSN helps liner shipping companies better design their own hub-and-spoke service networks (the most prevalent configuration for designing liner shipping service networks), especially regarding the choice of hub ports' locations^{18,19}. The hub-and-spoke network configuration is also widely adopted in the design of many other transportation, distribution, and infrastructure systems (e.g., air transportation, railway transportation, and telecommunication systems); indeed, one of the most crucial and general issues in designing such network configuration is to strategically select hub nodes and well understand how the selected hub nodes can serve as transshipment and switching points to improve the overall efficiency and economy of flow transportation in the entire network¹⁷. The gateway-hub structural-core organization discovered in the GLSN therefore provides new insights to transport literature that aims at better understanding and designing hub-and-spoke networks for various transportation systems. >>

Third, the finding on the (gateway-hub) structural-core organization of the GLSN is highly related with, and contributes to, the literature on network partitioning. Note that the detection of the gateway-hub structural-core is thanks to the “gateway-ness”, a node topological indicator for the first time proposed and formalized in a mathematical formula in the present study. Briefly, one contribution is that, the “gateway-ness” itself adds a new analytic tool to the specific field of network cartography (i.e., an established network science approach, which aims to better understand the structural organization complexities of modular networks through investigating nodes’ network roles), directly contributing to literature on complex modular networks and thus on network partition. We kindly notice the reviewer that modular division is one of the most widely adopted methods in complex network partition, as the modular community structure is one of the most ubiquitous properties of complex networks (Girvan and Newman, 2002; Newman, 2006). Another contribution to literature on complex modular networks (and thus on network partition) is that, the topological property termed the gateway-hub structural-core organization brings new insights into the complexity of the modular structural organization of real-world networks, especially regarding the issue that how the individually segregated modules in complex modular networks are integrated as a whole via a few hub nodes. For the two contributions stated above, detailed discussions are respectively provided in the third and fourth paragraphs of the Discussion section (lines 590 – 625, pages 19 – 20), which are now also pasted below.

<< From the point of view of network analysis, our study offers new tools to investigate the interface between network structure and functional outcomes^{7,43,48,54,55} in real-world networked systems. We provide novel evidences in support of a pivotal theoretical notion of network cartography: networks are formed by nodes with functional network-specific roles. The pioneer scheme of network node-role assignment introduced by Guimerà and Amaral^{43,54} classifies nodes for modular networks into topological roles based on two statistics (i.e. inside-module degree and participation coefficient), and has witnessed a great success in its application to analysis of various networks (e.g. protein-protein interaction networks⁵⁶ and brain networks⁵⁷). Unlike these networks, however, here we discover that the GLSN is primarily affected by neither the ports with high inside-module degrees nor the ports with high participation coefficients. Rather, the GLSN functionality significantly depends on ports with high outside-module degrees, which we term gateway hubs. Our work suggests that network-specific node-roles could vary across different types of networks; indeed, the GLSN analysis benefits more from gateway hubs than previously defined types of hubs. As we have illustrated, the gateway-hub role with its formal definition provides one example that is worth further attention, which might also be applicable to the node-role analysis of other kinds of modular networks.

The topological property termed gateway-hub structural core brings new insights into the complexity of the modular structural organization of real-world networks, in terms of how the individually segregated modules in such complex modular networks are integrated as a whole via a few hub nodes. Indeed, modular community structure is one of the most ubiquitous properties

of complex networks³⁸; many networks of interest in the sciences, including social networks, computer networks, and metabolic and regulatory networks, are found to divide naturally into communities or modules. Segregation and integration in networks with such modular structure has been widely discussed, and is considered fundamentally important for understanding how complex networked systems fulfill their functions^{21,22}. Specifically, segregation can be seen from the existence of individual modular communities that are defined by high density of connectivity among members of the same community and low density of connections between members of different communities. Integrative processes in networks can be viewed from at least two different perspectives, one based on the efficiency of global communication and another on the ability of the network to integrate distributed information. And the integration of a network relies in large part on network hubs, i.e., a few nodes that are highly mutually connected and highly central in the topological structure of the network. Then one crucial starting step to address such integration processes is to identify special classes of network hubs, which can be defined on a number of criteria. As shown here, in the GLSN such hub nodes form a gateway-hub structural core of ports that proves to be greatly important in the integration of the GLSN and highly associated to the network's functional outcomes. >>

Detailed comments

Introduction

- should refer to important works on international trade and maritime transport such as Clark, Dollar & Micco 2004, Limao and Venables 2001

REPLY #2: Revised accordingly. Now the two recommended studies are referred to, in the present revised manuscript (lines 51 – 53, page 1). The revision is also pasted below.

<< The importance of maritime transport in supporting international trade makes it indispensable to the sustainable economic development of our world³⁻⁵. >>

- what means "total goods loaded on ships"?

REPLY #3: The term "total goods loaded on ships" (lines 48 – 49, page 1) means the total volume of all types of goods loaded on ships, as defined by the data source (i.e., UNCTAD, <https://unctadstat.unctad.org/wds/TableViewer/tableView.aspx?ReportId=32363>). This data source is available in the UNCTAD data centre website link (which we show in our manuscript as the reference 2: <https://unctadstat.unctad.org/wds/ReportFolders/reportFolders.aspx>). Now we have added the explanation into the revised manuscript text (lines 48 – 49, page 1), which is also pasted below.

<<···, and total goods loaded on ships (i.e., total volume of all types of goods loaded on ships): ···>>

- "strong correlation": between which variables and by which fit?

REPLY #4: As written in the preceding sentence (lines 46 – 49, page 1), the two correlation coefficients reported in the manuscript text (line 50, page 1) are the Pearson correlation coefficient between world's total Gross Domestic Product and total goods loaded on ships and the Pearson correlation coefficient between the global merchandise trade (i.e. merchandise imports and exports) and total goods loaded on ships, during the period from 1970 to 2016. These variables are all publicly available in the UNCTAD database (as indicated in reference 2). Pearson correlation coefficient is calculated by linear regression. Now we have specified in the manuscript text (lines 49 – 50, page 1) that the two correlation coefficients are Pearson correlation coefficients; the revision is also pasted below.

<<...During the period from 1970 to 2016 their respective Pearson correlation coefficients ...>>

- the current ones: note that contemporary shipping networks have been analyzed through numerous works already, even from the 19th century (see Ducruet, Tavasszy, Hu and Zhu 2009, Deng et al. 2009, etc.)

REPLY #5: The corresponding sentence containing the term "the current ones" was revised (lines 69 – 72, page 2). We never disagree with the fact that maritime shipping networks have been analysed by previous work, such as those mentioned by the reviewer and those we already cited in the manuscript. But we revised the sentence to make it even clearer:

<< Indeed, to better exploit new international shipping routes, we need to improve understanding of the current ones (i.e., shipping routes that form the global maritime transportation) and their complex systems association with international trade, as is the aim of the present study. >>

- economies of scale: should be explained briefly for non-specialists

REPLY #6: Revised accordingly. Now we have briefly explained the meaning of economies of scale in liner shipping. Please see the revised manuscript text (lines 58 – 60, page 1) pasted below.

<< pursue the economies of scale (i.e., the adoption of large container vessels to decrease the shipping cost at sea per cargo unit). >>

- from "Note that ..." to "importance." is not useful, it looks like a reply to reviewers

REPLY #7: We have removed this sentence.

- the mention of LSCI should be brief and only for the sake of supporting your arguments

REPLY #8: We agree. As indicated in our reply #7 above, we have removed the mention of LSCI.

- arguments for a new analysis of the GLSN are somewhat weak, since industry data is less precise than vessel movement data (it should be said to what extent is the data used in your paper is

original and how);

and about previous studies, the fact that the current paper uses novel approaches is good, but again, the purpose is a description of the network structure, not of the impact on international trade, which would need much more data and other methods.

REPLY #9: First of all, we find it difficult to agree with the reviewer's comment that "arguments for a new analysis of the GLSN are somewhat weak, since industry data is less precise than vessel movement data". Liner shipping service routes data (i.e., the type of shipping data adopted in the present study, here called by the reviewer as industry data) are more precise and more suitable than container vessels' movement trajectory data, for the particular topic that is for the first time investigated in the present study: the association between the structure of the global liner shipping network and international trade. Detailed reasons that justify this statement, along with the issue that to which extent and how the liner shipping service routes data used in our paper is original (regarding the particular topic for the first time investigated in the present paper), had already all been explicitly explained in the manuscript (now available in the second paragraph of the subsection titled "Data on liner shipping service routes" in the Methods section; (lines 671 – 693, page 21), which we now also paste below.

Secondly, we thank the reviewer for acknowledging the novelty of the approaches proposed in our study. As clarified in the Introduction (lines 69 – 71, page 2), the aim of the present study is "to improve understanding of the current ones (i.e., shipping routes that form the global maritime transportation) and their complex systems association with international trade". Clearly, such an aim includes not only (1) analysing the structure of the GLSN (that is formed by liner shipping service routes in the world) but also (2) analysing how the GLSN structure is associated with international trade. However, we would like to emphasize to the reviewer that "the association between the GLSN structure and international trade" (as indicated in our point (2) above) is indeed different from "the impact of GLSN structure on international trade" (as claimed by the reviewer). And we have never claimed in our paper that the purpose of our study were to reveal the "the impact of GLSN structure on international trade". Our results regarding the association between the GLSN structure and international trade are reported in the subsection titled "Structural core and international trade" (pages 16 – 18). But we agree with the reviewer's point that understanding the impact of the GLSN structure on the international trade would need much more data and other methods, which we now have pointed out in the Discussion section of the present revised manuscript (lines 646 – 650, page 20). The revised text is also pasted below.

Manuscript text (lines 671 – 693, page 21):

<< Previous studies had used two types of shipping data to construct liner shipping networks: container vessels' movement trajectory data^{11,12} and liner shipping service routes data^{13,14}. The two types of data are by nature very different: the former are container vessels' voyage data logs, which can be known only after voyages are completed; the latter are service schedule data, which

are regular and are known much earlier than making voyages. Both types of data have merits and are useful for analyzing different research topics. However, regarding the particular topic that is (for the first time) investigated in the present study (i.e., the association between the structure of the global liner shipping network and international trade), liner shipping service routes data are more suitable than container vessels' movement trajectory data, due to the following facts of liner shipping. There in fact exists a unique property of liner shipping, which does not exist in any other maritime shipping mode: shipping companies always pre-fix liner shipping service routes, including the end-point ports and multiple other ports between them and the vessels deployed; vessels go back and forth on such pre-fixed service routes. The information of this unique property is precisely contained in liner shipping service routes data but is lost in container vessels' movement trajectory data; regarding the trajectory data, one cannot know the two end-point ports of a service route and thus loses the information on the unique property of liner shipping. Indeed, the unique property of pre-fixed routes makes liner shipping service essentially different from other modes of maritime shipping services, in the sense that it does not just simply serve the demand of international trade but also could potentially influence the demand of international trade between countries. As the economic theory of induced traffic demand⁵⁸ posits, increased transportation capacity can stimulate extra traffic demand. Note that, in industry practice, shipping companies always pre-release their liner shipping service routes, which in many cases could be even one year prior to making voyages. >>

Manuscript text (lines 646 – 650, page 20):

<< Furthermore, additional research will be required to understand the impact of the GLSN structure on international trade; in particular, establishing the causal influence mechanisms underlying the observed correspondence between the GLSN's structural-core organization and international trade may require additional longitudinal network and economic data and as well as methods on causal inference. >>

- the introduction is too long overall

REPLY #10: Now it has been shortened, as can be seen from the present revised manuscript.

- the central research question can't be understood; it again makes the link with international trade, economic importance, countries, that do not appear in the subsequent analysis only looking at nodes and links

REPLY #11: The central research question has been further explained in the present revised manuscript, as already carefully addressed in the reply #1 above. In the present reply, we specifically address the reviewer's comment that "it again makes the link with international trade, economic importance, countries, that do not appear in the subsequent analysis only looking at

nodes and links”. To assist our reply, please allow us to quote part of the manuscript text clarifying the central research question (lines 76 – 96, page 2):

“These facts give rise to the necessity of addressing a central research question of the present study: what specific topological properties does the structural organization of the GLSN present, and how the structure of this network is associated with its functional outcomes in realizing the cargo transportation of international trade?..... Then, we explore the association between a new network structural measure (that we propose and term gateway-ness) and the GLSN’s two functional outcomes of practical importance: individual ports’ economic performance (i.e., ports’ traffic capacities in liner shipping), which is at the node level; countries’ international trade statuses (i.e., the international trade value of countries and the bilateral trade value between countries), which is at the system level.”

First of all, as explicitly written above, the central research question makes the link of “the GLSN structure” with “this network’s functional outcomes in realizing the cargo transportation of international trade”. And the phrase “this network’s functional outcomes in realizing the cargo transportation of international trade” explicitly refers to two specific outcomes that are respectively at the node level and at the system level: the individual ports’ economic performance (i.e., ports’ traffic capacities in liner shipping), countries’ international trade statuses (i.e., the international trade value of countries and the bilateral trade value between countries). Therefore, the links made in our research question are the following two: (1) the link between the GLSN structure and ports’ economic performance (which in the present paper refers to ports’ traffic capacities in liner shipping); and (2) the link between the GLSN structure and countries’ international trade statuses (which in the present paper refer to the international trade value of countries and the bilateral trade value between countries).

Second, the two links mentioned in the research question all appear in the subsequent analyses. Regarding the analysis (and the related results) for the link between the GLSN structure and ports’ economic performance, please see the subsection titled “Structural embeddedness and economic performance of ports” (pages 14 – 16). Particularly, table 1 there clearly shows the extent to which individual ports’ connectivity patterns (which are all defined by the GLSN structure only) are linked with their economic performance. Regarding the analysis (and the related results) for the link between the GLSN structure and countries’ international trade statuses, please see the subsection titled “Structural core and international trade” (pages 16 – 18). Particularly, figure 9 there clearly shows the extent to which the GLSN topological indicators (which are all defined by the GLSN structure only) are linked with international trade indicators of countries.

Results

- note that vessel movement data may also be more precise than schedule data as it shows the real circulation pattern, with also feeder and shortsea, sometimes sea-river

REPLY #12: We never wrote anything that disagree with this comment of the reviewer. We have carefully explained in the manuscript text that both the two types of shipping data (i.e., vessel movement trajectory data and liner shipping service routes data) have merits and are useful for analysing different research topics, and that liner shipping service routes data are more suitable for the present study. Please see the detailed explanation in the current manuscript text (the second paragraph of the subsection titled “Data on liner shipping service routes” in the Methods section; lines 671 – 693, page 21). And we kindly notice the reviewer that this detailed explanation was also available in the previous revised manuscripts, together with the corresponding detailed replies to reviewers’ comments (our reply # 1 to reviewer #2 in the reply letter of first-round revision and reply # 17 to reviewer #1 in the reply letter of second-round revision).

- previous studies also used AIS data, which should be emphasized, see Kaluza et al. 2010 for instance and many others

REPLY #13: In the corresponding manuscript text (line 672, page 21) that mentions the point that previous studies used AIS data (i.e., vessels’ movement trajectory data), the studies of Ducruet and Notteboom (2012) and Kaluza et al. 2020 are already emphasized and cited (as references 11 and 12, respectively, in the present revised study).

- figure 1 does not show the difference between space L and space P; a further discussion about the implication of choosing one of them is needed

REPLY #14: Figure 1 aims to illustrate how we have constructed the global liner shipping network in the present study by using the network representation method of space P only, and therefore in this figure we do not need to show the difference between the space P and space L (which is irrelevant to the present study). A further discussion about the respective implication of the two network representation methods of space L and space P, the difference between them, and the present study’s choice of space P over space L is already available in the manuscript (lines 696 – 709, pages 21 – 22). And we kindly notice the reviewer that this discussion was also available in the previous revised manuscripts and in the corresponding previous replies to reviewers’ comments (i.e., reply # 37 to reviewer #1 in the reply letter of first-round revision, and reply #27 to reviewer #1 in the reply letter of second-round revision).

- ports with larger TEU are more central in the hyperbolic layout: what does this mean for research?

REPLY #15: The sentence “ports with larger TEU are more central in the hyperbolic layout” appears in the caption of the fig. 1. So, this comment of the reviewer refers to the hyperbolic layout we use to present the edges and nodes of the GLSN under study (fig. 1b, page 3). We now reply to the reviewer directly with the following explanatory text added into the caption of fig. 1 (lines 133 –

139, page 4):

<< The coalescent embedding hyperbolic layout locates at centre the nodes that are fundamental for the efficient navigability of a complex network²⁶. As such, the observed phenomenon that ports with larger TEU are more central in the hyperbolic layout means that ports with larger TEU are fundamental for the efficient navigability of the GLSN in transporting cargos traded worldwide. This suggests that ports' traffic capacity measured in TEU is indeed a meaningful indicator to be associated with international trade (as we will show in the rest of the study). >>

In addition, the reviewer's comment also reminds us to add the following explanatory text into the caption of fig. 7 (lines 354 – 360, page 12) where we show that the detected gateway-hub structural-core is at the centre of the hyperbolic layout:

<< The detected gateway-hub structural-core is at the center of the hyperbolic layout. Nodes at the center of this layout are crucial to support the efficient navigability of a complex network²⁶. From a previous analysis we know that these nodes at the center have also larger TEU (as presented in fig. 1b), this indicates that the gateway-hub structural core is indeed mainly composed by ports with larger TEU that are fundamental for efficient navigability of the network. Therefore, structural-core ports are important candidates to be associated with international trade (as we will analyse in the next section). >>

- influence of shipping on trade: this is NOT the purpose of this study, but it is regularly claimed to be so

REPLY #16: The term “influence of shipping on trade” was summarized by the reviewer from the following sentence in the manuscript (now available in the supplementary note 2; lines 592 - 596, page 39, supplementary information file): “This study follows the meaningful attempts of previous studies^{12,13} to quantitatively explore the influence of containerized shipping on international trade flows; for instance, reference¹² adopted a gravity model to analyse the influence of shipping distance and frequency of shipping services on containerized trade flows between countries. This study for the first time investigates how the topological structure of a modern GLSN is associated to international trade.”

As clearly written in this sentence, what we claim is not that the present study for the first time investigates the influence of shipping on trade, but instead that “the present study for the first time investigates how the topological structure of a modern GLSN is associated to international trade”. Indeed, in Fig. 9 and its corresponding subsection titled “Structural core and international trade” (i.e., the final subsection of the Results section; pages 16 – 18), we have shown the strong association between the GLSN topological indicators (which are defined for individual countries and for individual pairs of countries, based on the structural-core organization of the GLSN) and international trade indicators (which refer to the international trade values of individual countries and the bilateral trade values of individual pairs of countries).

- assortativity may be more meaningful if including feeder and shortsea flows but it is not the case in the proposed paper, consider clarifying

REPLY #17: The adopted data set on world's liner shipping service routes includes all the service routes of at least the top 100 liner shipping companies in the world, thus including the feeder and shortsea service routes of these companies. The data set does not further indicate which route is a feeder route or a shortsea route, but this piece of unknown information does not negatively influence the validity of the assortativity result reported in the manuscript (lines 151 – 154, page 4). However, we think the reviewer's comment is constructive and therefore have added the following sentence into the related discussion of the assortativity result (now available in the supplementary note 2; lines 639 – 641, page 40, supplementary information file), which is also pasted below.

<< Assortativity analysis of the GLSN may be even more valuable if it can be combined with the information on feeder and shortsea service routes in the GLSN. >>

- regional port systems: see the work of Ducruet and Notteboom 2012 highlighting nodal regions where neighboring ports are centralized upon one main hub

REPLY #18: We are aware of the work of Ducruet and Notteboom 2012 and have already emphasized and cited this work (as reference 11 in multiple places in the main manuscript and as reference 23 in the supplementary information). Now this work is also mentioned in the corresponding place that the reviewer pointed out here (now available in the supplementary note 2; lines 634 – 635, page 40, supplementary information file). Please see the revised manuscript text pasted below.

<< In regional port systems hub ports tend to be connected with many small ports in their regions²³, ...>>

- why not providing a comparative analysis in the figures for 2015 and 2017 instead of only in the text of figure 2?

REPLY #19: We kindly notice the reviewer that we had already analysed the results for the year 2017, compared it with the results for the year 2015, and confirmed the robustness of the results. Detailed illustrations are clearly written in all the previous revised manuscripts and also in the present revised one (lines 200 – 203, page 5), which we paste below.

<< To confirm these basic topological properties of the GLSN, we repeated the analysis by using the liner shipping service routes data of 2017. The results for the GLSN of 2017 are consistent with the present results for the GLSN of 2015 (Supplementary Fig. 1). >>

- it would be interesting to compare the small world ness between space L and space P, as space L better reflects the hub structure of the GLSN compared with space P (which better reflects trade structure), as proposed by the authors lines 338-340

REPLY #20: We do not think that it would be interesting to compare the small-world-ness analysis (or any other analyses of the present paper) between the two network representation methods so-called the space L and space P, for the following reason: the space P is suitable for the specific focus of the present study, while the space L is not. A further discussion about the respective implication of the two network representation methods of space L and space P, the difference between them, and the present study's choice of space P over space L is already available in the manuscript text (lines 696 – 709, pages 21 – 22).

Second, in lines 338 – 340 of the last revised manuscript, we wrote the following sentence: *“Previous studies^{20,23} had compared some basic topological properties of global maritime shipping networks observed under the two representations of network topology.”* We notice the reviewer that in this sentence (and any other places in our manuscript), we never proposed that “it would be interesting to compare the small world ness between space L and space P”, nor did we proposed that “space L better reflects the hub structure of the GLSN compared with space P (which better reflects trade structure)”. The reason that we added into our revised manuscript this sentence and also another sentence preceding this one (now these two sentences are available in the supplementary note 2; lines 707 – 714, pages 41 – 42, supplementary information file) was just to give readers the simple information as follows: previous studies had compared some basic topological properties of global maritime shipping networks observed under the two representations of network topology, for addressing their corresponding research topics. And this revision was encouraged by a previous comment of the reviewer #1, as can be seen from our reply #47 to reviewer #1 in our reply letter of the first round of revision.

- same for efficiency, space P is often higher than space L; could this be emphasized?

REPLY #21: As we have explained in our replies #14 and #20 above, we do not think that it would be interesting to compare any analyses of the present paper (including the efficiency analysis) between the two network representation methods so-called the space L and space P. For detailed explanations, please refer to those two previous replies. In addition, our study is already vast and could not contain further analysis.

- not well explained how the distance of links was integrated in the calculation of efficiency

REPLY #22: How the distance of links was integrated in the calculation of efficiency is precisely explained in the definition of efficiency. We kindly notice the reviewer that, we already clarified the definition of efficiency in the corresponding Results section where the efficiency was introduced for the first time (lines 167 – 173, page 4): “For spatially embedded networks, efficiency in the flow transfer between a pair of ports is defined as the reciprocal of the shortest distance between them (i.e. the smallest sum of the physical distance throughout all the possible paths between them). Global efficiency of the GLSN is calculated as the average efficiency of all pairs of

nodes in the network. Local efficiency of a port is calculated as the global efficiency of the subnetwork consisting of all the neighbors of this port, and the local efficiency of the GLSN is the mean over all ports' local efficiencies." Moreover, the mathematical formulas of the global efficiency and local efficiency were already provided in the corresponding Methods section (lines 758 – 783, pages 23 – 24). With those formulas, one shall be able to precisely understand how the distance of links was integrated in the calculation.

- economic organization: should discuss the non-planar dimension of the GLSN where nodes may have many links i.e. be less constrained than planar nets

REPLY #23: We do not think more discussions on the “non-planar dimension of the GLSN” would significantly improve our manuscript, because the “non-planar dimension of the GLSN” issue is not closely related with our research focus. The focus of the present study is on characterizing the structural properties of the GLSN and their association with this network's functional outcomes, not on what factors/issues (such as the non-planar dimension issue) that lead the GLSN to such structural properties. And we admit that the present study cannot be all-inclusive, just like most other scientific studies. In addition, we kindly notice the reviewer that the present manuscript is under a maximum length limit, as instructed by the editor. To conclude, we would like not to add into the present manuscript any further discussion or information that is not closely related with our research focus, to avoid making the manuscript unnecessarily lengthy.

- same for LPC-corr analysis, planar vs. non-planar discussion is missing

REPLY #24: The same logic as our reply #23. We do not think more discussions on the “planar vs. non-planar” would significantly improve our manuscript, because the “planar vs. non-planar” issue is not closely related with our research focus. The focus of the present study is on characterizing the structural properties of the GLSN and their association with this network's functional outcomes, not on what factors/issues (such as the planar vs. non-planar issue) that lead the GLSN to such structural properties. Once again, we admit that the present study cannot be all-inclusive, just like most other scientific studies.

- "by socio-economic factors": which ones? lines 413-415

REPLY #25: This term “by socio-economic factors” is mentioned in a brief discussion about the high LCP-corr of the GLSN (now available in the supplementary note 2; lines 699 – 703, page 41, supplementary information file): “The *LCP-corr* of the GLSN is very high, like airport networks²⁸. Airport networks, different from merely geographically constrained networks such as road-maps, have high *LCP-corr* because significant socio-economic and political factors also contribute to shape their topology²⁸. Therefore, this result might imply that the GLSN topology is influenced significantly also by socio-economic factors, rather than by geographical constraints alone.” Within this context, the socio-economic factors refer to the social and economic factors that might have some influence on the global liner shipping in general. Here, we prefer not to refer to any specific

factors, because that is not the focus of the present study and is not analysed here. The focus of our study is already clarified in our reply #23 above.

Multiscale modularity

- the "move" to the search for community structure is somewhat mechanical and without clear explanation

REPLY #26: We now have added another brief sentence into the existing explanation. We think the revised explanation (lines 207 – 211, page 6) should be sufficient for readers to understand the move from the previous analysis of basic topological properties of the GLSN to the present analysis of community structure. The revised explanation is also pasted below.

<< Modular community structure is one of the most ubiquitous properties of complex networks³⁸. Many real-life complex systems have the property of multiscale modularity (or hierarchical modularity), and there are several general advantages to modular and hierarchically modular network organization, including greater robustness, adaptivity and evolvability of network function³⁹. >>

- see Ducruet and Zaidi 2012 for the search of such structures using another algorithm in the GLSN, namely topological decomposition (and a comparison between space L and space P)

REPLY #27: We are aware of the work of Ducruet and Zaidi 2012. In that work, the authors applied to global maritime transportation networks an algorithm that implements successive removal iterations of larger (or smaller) degree ports, and they name it topological decomposition analysis. Apparently, due to the large difference in methodology between the topological decomposition and the modular community partition, the structure revealed by topological decomposition analysis of Ducruet and Zaidi 2012 is just very different from the modular community structure revealed in the present paper. Such difference does not affect any analysis in the present study.

- the choice of the methodology is not sufficiently explained compared to the myriad of other possible ones

REPLY #28: Given the present context of modular community analysis, we think that here the reviewer was probably referring to our choice of the fast unfolding algorithm to implement the modular community partition of the GLSN. We had already explained in the manuscript text why we made this choice (lines 213 – 217, page 6), which we now also paste below.

<< The division of port communities is based on a criterion of modularity maximization²⁰. And we adopted an algorithm of fast unfolding communities in large networks⁴⁰, to search for an optimal

partition that maximizes the modularity (Q). This algorithm might miss some small structures, but at large scale can be trusted⁴¹. >>

And we kindly notice the reviewer that, in our reply letter of first-round revision, in our reply #8 to the reviewer #2, this community division issue and the choice of the Louvain algorithm (compared to other possible algorithms) had already been explicitly explained. Furthermore, in our reply letter of second-round revision, in our reply #21 to the reviewer #1, we had also quoted part of that previous reply. For the reviewer's information, we now once again quote part of that previous reply as follows:

“Reply #8 (to reviewer #2, first-round revision): The Louvain algorithm is a greedy algorithm that attempts to maximize the modularity of a partition of the network (Blondel et al, 2008). Although modularity maximization is hard and the Louvain algorithm might miss some small structures in optimizing a partition for the modularity maximization, at large scale this algorithm can be trusted (Fortunato and Hric, 2016). Also, the goal of our work, just like those of many others, is not to find the best community division (or the most modular structure) of the network. Instead, we, in agreement with Newman (2006), regard the community structure detection simply as a data analysis technique used to shed light on the structure of large-scale network data sets. And this technique indeed helps one conduct more advanced analyses, e.g., network cartography.”

- land mass bottlenecks: it should be more discussed in light of the results obtained

REPLY #29: We do not think more discussions on the land mass bottlenecks would significantly improve our manuscript, because the “land mass bottlenecks” issue is not closely related with our research focus. As already stated in our replies #23 and #24 above, the focus of the present study is on characterizing the structural properties of GLSN (and their association with this network's functional outcomes), not on what factors/issues (such as the issue of land mass bottlenecks) that lead the GLSN to such structural properties. Once again, we admit that the present study cannot be all-inclusive, just like most other scientific studies. And we kindly notice the reviewer that the present manuscript is under a maximum length limit, as instructed by the editor. To conclude, we would like not to add into the present manuscript any further discussion or information that is not closely related with our research focus, to avoid making the manuscript unnecessarily lengthy.

- the economic, logistical and geopolitical division of the world is not well explained using figure 3, where the pattern looks very interesting; what does it tell us about world economy and world shipping?

REPLY #30: Thanks for the reviewer's interest in the community structure results presented in fig. 3. First, we had already carefully explained the community structure results with reference to the economic, logistical and geographical division of the world, shedding light on better understanding the world economy and world shipping; please see details in the corresponding manuscript text (lines 217 – 228, page 6), which is now also pasted below. In our opinion, the present explanation is sufficient for readers to well understand the related results.

Second, we kindly notice the reviewer that the economic, logistical and geopolitical division of the world cannot be completely well explained/understood using our fig. 3 only, because that division is shaped by various factors that are wider than the global liner shipping network (not to mention the modular community structure of this network presented in fig.3). Therefore, we are cautious to avoid overinterpreting our results, with respect to the economic, logistical and geopolitical division of the world. And it is not within the scope of the present study to well address the economic, logistical and geopolitical division of the world.

<< The seven upper-module port communities in the GLSN are observed to be spatially compact and to correspond to geographically neighboring regions (Fig. 3b), demonstrating the relevance of the method in this case. However, boundaries of port communities are not simply defined by a continental division but are also related with maritime circulation effect such as land mass bottlenecks and interoceanic canal constraints.

The modular structure of the GLSN seems to reflect the contemporarily parallel trends of regionalization and globalization of international trade and economy^{11,42}; the OD matrix in Fig. 3b clearly shows the intra-regional concentration of global seaborne trade flows, meanwhile trade between different regional markets can be seen from the existence of a few inter-module links. In the GLSN long-range links are relatively few and mainly appear as inter-module ones (Supplementary Fig. 2), as are the case of many spatial networks (see Supplementary Note 3 for a brief discussion). >>

- hubs diversity: again, the search for port functions comes a bit out of the blue

REPLY #31: We kindly notice the reviewer that, in our reply letter of second-round revision, we had meticulously addressed this same comment of the reviewer in our reply #21 and reply # 33 to reviewer #1. For the reviewer's information, we now quote below the reviewers' previous comments (in part) and our previous replies (in part).

Text quoted from our reply #21 to reviewer #1, in our reply letter of second-round revision:

"- the three hub measures seem to come out from an empirical investigation without any underlying hypothesis; did you search for these properties with a question in mind or did it happen by chance running any possible partitioning algorithm?

REPLY #21: *As is clarified in the beginning of the subsection titled “Hub diversity” (lines 469 – 472, page 13), the three hub measures are based on cartographical analysis in complex network science (Guimera et al 2005), which hypothesizes that the role of a node can be determined, to a great extent, by its connectivity pattern in the network structure. The three hub measures correspond to three different connectivity patterns of nodes. Definitely, the reason that we analyze such topological properties of individual ports is to address the central research question of the present study (which is already explained in detail in our above reply #8). Certainly, analyzing nodes’ topological properties is an indispensable part of work to an in-depth understanding of the structural organization complexity of a complex network. And, as the reviewer might know, community partition algorithms are not designed to achieve this task.”*

Text quoted from our reply #33 to reviewer #1, in our reply letter of second-round revision:

“- hubs diversity - “we were able to”: the reader has no doubt about your ability to apply many measures; however those are not well justified right from the introduction so that we have the impression that you are searching for something but what

Reply #33: *This subsection titled “Hub diversity” (pages 13 - 17) investigates three kinds of hub roles of individual ports: gateway hubs, provincial hubs, and connector hubs, which are quantified by three topological measures named outside-module degree (B), inside-module degree (Z), and participation coefficient (P), respectively. These three measures aim to analyze the topological properties of individual ports in the GLSN structure by characterizing three different connectivity patterns of them. Indeed, analyzing nodes’ topological properties is an indispensable part of work to an in-depth understanding of the structural organization complexity of a complex network. The necessity of conduct such analysis is justified in the Introduction section of the revised manuscript, taking as a starting point a fundamental theoretical hypothesis in network science (lines 61 – 64, page 2): “A fundamental theoretical hypothesis in network science is the concept that structure matters, positing that the functional outcomes of a complex networked system, at both the system level and the individual node level, depend at least in part on the network structure⁶⁻⁸.”*

- why 1.5 above the SD? why higher than 0.7? needs an explanation of threshold choice

REPLY #32: We kindly notice the reviewer that, in our reply letter of second-round revision, in the reply #37 to reviewer #1, we had already meticulously explained the heuristic choice of these thresholds. For the reviewer’s information, we now quote below the reviewers’ previous comment and our previous reply (which also applies to the threshold choice of higher than 0.7 for defining the connector-hub ports).

Text quoted from our reply #37 to reviewer #1, in our reply letter of second-round revision:

“- better explain your choices of standard deviation at least 1.5; density at least 0.8... where do these thresholds come from?”

Reply #37: These two thresholds are considered heuristically in investigating the structural-core organization of the GLSN. To make the following explanations more easily to be understood, please first allow us to quote the related modified manuscript text (in the subsection “Defining structural core”; lines 547 – 556, page 17):

“To quantify the possibly existent phenomena of structural-core organization, a structural core of the GLSN is defined as a set of hub ports that meet the following two criteria: (1) this set should consist of the largest number of the most important hub ports that form a subgraph of high density (i.e. the proportion of actual links in the maximum possible number of links); and (2) Here, we considered a density threshold of 0.8 (which is heuristically a sufficiently high density) and three types of hub ports defined heuristically above: provincial hubs (i.e. ports with inside-module degree at least 1.5 standard deviations above the community mean), gateway hubs (i.e. ports with outside-module degree at least 1.5 standard deviations above the community mean), and connector hubs (i.e. ports with participation coefficient of at least 0.7).”

The heuristic selection of the standard deviation threshold of 1.5 to define the provincial hub ports and the gateway-hub ports in the GLSN structure is due to the following consideration. Such analysis of nodes’ roles in the network structure is well known as network cartographical analysis (Guimera and Amaral, 2005). In the empirical application of cartographical analysis to various complex networks, there does not exist a unified standard deviation threshold in defining provincial hub nodes. Even studies on a same type of network could choose different thresholds, so long as they are reasonable and useful for investigating the network structure; for instance, a threshold of 1 standard deviation above the community mean is used in a brain network study (Meunier et al, 2009), and another threshold of 2.5 standard deviations above the community mean is used in another brain network study (Joyce et al, 2010). Here, we use a threshold of at least 1.5 standard deviations above the community mean to define the provincial-hub ports and the gateway-hub ports, and we think it is a reasonable one. It is important for one to understand that, such definitions are not the final result of our study, but help us restrict the focus to these hub ports in further investigating whether there emerges, from the collective behaviors of a few non-ordinary ports, some important topological property of the overall structure of the GLSN (such as the one reported in the present study, i.e., the structural-core organization of the GLSN).

The heuristic selection of a density threshold of 0.8 to evaluate whether the ports of interest form a subgraph of high density (i.e. the proportion of actual links in the maximum possible number of links) is due to the following consideration. For simple graphs in general, it shall be fair to claim that nodes that form a graph of a density larger than or equal to 0.8 are densely connected with each other.

To sum up, just like those heuristic rules and thresholds in numerous scientific studies that provide a starting point for the propositions (Walker et al, 2006), the two adopted heuristic thresholds in the present study provide us with a starting point to investigate the proposition of the GLSN's structural property termed "the structural-core organization". And it is worth emphasizing that the detected structural-core organization in the GLSN proved to be statistically significant (fig. 7, page 20); significantly important in supporting long-distance maritime transportation (fig. 9, page 22); and, highly relevant to countries' international trade statuses (fig. 11, page 28). Such results justify the two selected thresholds; they clearly show that the finding of the structural-core organization of the GLSN—obtained under the two adopted heuristic thresholds—is indeed valuable in deepening our understanding of the structural organization complexity of the GLSN and its association with the network's functional outcomes in realizing the cargo transportation of international trade for countries."

- figure 5: it is not clear whether ports cumulate two or more functions; whether specific ports are already identified as "hubs" in the transport literature (e.g. Singapore), etc. a thorough interpretation is needed

REPLY #33: First of all, it is already clearly presented in fig. 5 whether ports cumulate two or more functions (of hub roles): as written in the caption of fig. 5 (lines 288 – 293, pages 9 – 10), "In each plot, triangles denote ports which have only that particular type of hub role under investigation; circles, ports which have one additional type of hub role; crosses, ports which have all the three types of hub roles. Outside-module degrees (B), inside-module degrees (Z), and participation coefficients (P) of the hub ports are reported in a file named "Supplementary File to Fig. 5". Source data are provided as a Source Data file."

Second, regarding those hub ports defined in fig.5, the issue that "whether specific ports are already identified as "hubs" in the transport literature (e.g., Singapore)" is not closely related with the central results of the present study (and their related analyses, either), for the reason clarified in the following points (1), (2) and (3). And we would like not to add into our manuscript any further interpretation or information on issues that are not closely related with our central results, to avoid making the manuscript unnecessarily lengthy (as also reminded by the reviewer #1 for multiple times and now required by the journal). Point (1) is that, defining those hub ports by itself is not a final result of our study; instead, it simply helps us restrict our focus to those hub ports in the further analysis, which is of more importance and tries to understand whether there emerges, from the collective behaviors of a few non-ordinary ports, some salient topological property of the overall structure of the GLSN (such as the one reported in the present study, i.e., the structural-core organization of the GLSN). Point (2) is that, our final results are the gateway-hub structural-core organization of the GLSN and its strong and positive association to the GLSN's functional outcomes in realizing the cargo transportation of international trade for countries, as explicitly

presented in the Results section's final subsection titled "Structural core and international trade" and also indicated in the title of this study. Note that: these final results are made available by the topological measure termed "gateway-ness", which is for the first time proposed in the present study and defines the gateway-hub ports presented in fig.5. Point (3) is that, we are confident that our final results—either in part or as a whole—had never been reported in any existing studies (including those in transport literature).

- without an interpretation we move on to the next section, which is inappropriate

REPLY #34: We kindly notice the reviewer that, just before moving on to the next section, we did have provided the corresponding interpretation (of the hub ports defined in the present study); please see the manuscript text (lines 257 – 272, pages 7 – 8), which is also pasted below. And we think the current interpretation is sufficient. Scientifically, it is important for one to understand that, us authors are—and should be—cautious to avoid putting too much further interpretations on the individual hub ports defined in the present study, due to the following reason: such definitions of hub ports are by themselves not the central/final result of our study, but help us restrict the focus to these hub ports in further investigating whether there emerges, from the collective behaviors of a few non-ordinary ports, some important topological property of the overall structure of the GLSN (such as the one reported in the present study, i.e., the structural-core organization of the GLSN).

<< Figure 4 shows the Z , B , and P values for each port, calculated in contexts of both modular communities and submodular communities. But for clarity, in the following analysis we focus on interpreting the results for the modular communities (hereinafter shortened to communities); in a similar way, it should be easy to understand results for the submodular communities. The fractions of provincial hubs, gateway hubs and connector hubs are 8.3%, 6.6% and 6.6% respectively, with 87.1% of the ports being non-hubs. As indicated in Fig. 5, 95.3% of those gateway hubs are also with at least another type of hub role: 29.7% of them are provincial hubs, 32.8% connector hubs, and the rest 32.8% both provincial hubs and connector hubs. Particularly, those ports that simultaneously serve as provincial hubs, gateway hubs and connector hubs concentrate greatly in the world's major trading regions of East Asia, Northwest Europe, North America and Europe Mediterranean. Indeed, port development is essentially related with the development of regional economy and international trade⁴⁴. By contrast, 49.4% of the provincial hubs and 32.8% of the connector hubs turned out to be without any other type of hub roles (Fig. 5). Hence, it seems to be a plausible conjecture that ports with gateway-hub roles are of significant importance in the structural integration of the GLSN, as we will analyze in the next section. >>

Gateway-hub structural core

- seems to be formed: this goes with a lack of a proper research question

REPLY #35: The phrase “seems to be formed” appears in the following sentence (now available in the supplementary note 4; lines 751 – 752, page 43, supplementary information file): “The GLSN seems to be formed by ports with network-specific roles—provincial hubs, gateway hubs and connector hubs.” This sentence is a brief summary of the analysis of three types of hub ports (which is presented in the preceding subsection titled “hubs diversity”; pages 7 – 10). The analysis of such three types of hub ports, which is based on three types of connectivity patterns (i.e., provincial-ness, gateway-ness, and connector-ness), is exactly driven by the following question (which is written in the subsection titled “hubs diversity” in the present revised manuscript; lines 253 – 256, page 7): “These three indicators help explore a crucial question: regardless of ports’ disparity in many aspects such as physical conditions, hinterland economies and socio-political environments, do ports present some similarities of connectivity patterns in the structure of the GLSN?” This is a proper research question justified by a fundamental hypothesis in network science, which we illustrate as follows. This research question aims to analyze the topological properties of individual ports in the GLSN structure through characterizing their connectivity patterns. Certainly, analyzing nodes’ topological properties in a network structure is an indispensable and basic part of work to an in-depth understanding of the structural organization complexity of a complex network. And the necessity of an in-depth understanding of the structural organization complexity of a complex network, at both the system level and the node level, is justified by the following fundamental theoretical hypothesis in network science (which is written in the Introduction section; lines 63 – 66, pages 1 – 2): “A fundamental theoretical hypothesis in network science is the concept that structure matters, positing that the functional outcomes of a complex networked system, at both the system level and the individual node level, depend at least in part on the network structure^{6–10}. ”

- why 0.8??

REPLY #36: We kindly notice the reviewer that, in our reply letter of second-round revision, we had already meticulously explained this threshold choice of 0.8 in our reply #37 to reviewer #1; the reviewer #1 had previously asked us to “better explain your choices of standard deviation at least 1.5; density at least 0.8... where do these thresholds come from?” Now in the present reply letter, for the reviewer’s information, that previous comment of the reviewer #1 and our previous reply are already quoted in the reply #32 above. For details, please refer to our reply #32 above.

- figures 6 and 8 are not well explained; the identified main hubs could simply large ports, regardless of their function; needs further analysis

REPLY #37: Both figures 6 and 8 present the identified 37 structural-core ports of the GLSN; these ports are not simply the large ports, as clearly shown in the supplementary fig. 7 of the present revised manuscript (page 6, supplementary information file). In addition, we kindly notice the reviewer that, in our reply letter of second-round revision, in our reply #22 to reviewer #1, we had already explicitly showed that the identified structural-core ports are not simply the large ports. And the related analysis in the manuscript was also provided in that previous reply. For the reviewer's information, that previous comment of the reviewer #1 and (part of) our previous reply are now pasted below.

Text quoted from our reply #22 to reviewer #1, in the reply letter of second-round revision:

"- the structural core is topologically central: hopefully it is, but does it have a geographic logic on top of this obvious fact? or does it contain largest ports wherever they locate? you should say more what mean your results to network and maritime specialists

Reply #22: *.....Our answer to the second question is no; we had already carefully demonstrated that the structural core of the GLSN is different from a collection of the largest ports in the network (i.e., a rich club). Please see fig. 10 in page 26 for detailed results. As one can note from this figure, the structural core does not contain just the largest ports wherever they locate, but rather, it includes also a few ports that are relatively small. These ports, though relatively small at the global level, are of fundamental importance to the development of their own regions in the wider context of global economy. The points stated in the present reply are all explicitly written in the manuscript text (lines 792 – 795, page 25)....."*

- lines 644 and after: are structural-core ports the ones with large betweenness (and/or a low clustering coefficient)? all the different analyses seem to conclude this simple fact

REPLY #38: The detected 37 structural-core ports are not the 37 ports with largest betweenness (please see fig 1 attached below), and are not the 37 ports with lowest clustering coefficient either (please see fig 2 attached below). Such results clearly show that, the structural-core ports detected in the present study (thanks to the "gateway-ness", a node topological measure for the first time proposed here) cannot be detected by using existing measures such as betweenness and clustering coefficient. Therefore, what claimed to be "simple fact" in the reviewer's present comment that "all the different analyses seem to conclude this simple fact" is untrue. And we clarify that, in our manuscript, there does not exist any analysis that seems to conclude this untrue "simple fact".

Fig. 1 A Venn diagram showing the overlapped and the mutually exclusive sets of ports between the 37 structural-core ports (in blue) and the 37 ports with largest betweenness (in yellow)

Fig. 2 A Venn diagram showing the overlapped and the mutually exclusive sets of ports between the 37 structural-core ports (in blue) and the 37 ports with lowest clustering coefficient (in orange)

- defining feeder connections between these values mean that this category includes shortsea and other non feeder ones; the local scale (as said lines 702-704) is more reflective of "hub-feeder" activity...

REPLY #39: The definition of feeder connections in the present study is not contrary to the reviewer's comment here. Specifically, as presented in fig. 8 (page 14), the feeder connections of the GLSN in the present study are defined as the connections between the structural-core ports and the non-structural-core ports. The adopted dataset on liner shipping service routes in the

world, based on which the present GLSN was constructed, do not exclude any particular type of liner shipping service routes (such as shortsea and non-feeder ones). The content in previous manuscript lines 702-704 (which is now in lines 423 – 427 and is pasted below) appears in the caption of fig. 8. The aim of adding such content is simply to help readers better understand the results presented in fig. 8, as previously suggested by the reviewer #1 (see our reply #55 to reviewer #1 in the reply letter of first-round revision). Nevertheless, we think the reviewer #1 appreciates the related analysis and results of this fig. 8 (which was numbered as fig. 9 in the previous revised manuscripts), as the reviewer #1 wrote in the second round of referee report “figure 9 most interesting analysis in this article - and related text”.

<< To better understand those three types of connections related with the structural-core organization of the GLSN, one is encouraged to refer to the hub-and-spoke service network configuration (Hu and Zhu, 2009)¹⁴ that is widely adopted by world liner shipping carriers in practice. The hub-and-spoke configuration is illustrated in the Supplementary Fig. 5. >>

Structural embeddedness and economic performance of ports

- economic performance: not sure about its definition and its place in this paper

REPLY #40: In this paper, the economic performance of ports refers to the port performance indicator named “ports’ traffic capacities”, which is ground-truth data derived from the database. This is clarified in both the previous and present manuscripts, now in the Introduction section (line 94, page 2) and in the subsection titled “Structural embeddedness and economic performance of ports” (lines 444 – 446, page 15). We kindly notice the reviewer that this same question had already been replied several times in the previous two rounds of revision; please see our previous replies #10 and #34 to the reviewer #1 in the reply letter of first-round revision and our previous replies #12 and #23 to the reviewer #1 in the reply letter of second-round revision.

- theorising: the first paragraph seems to refer to methods in sociology like structural equivalence or other

REPLY #41: The first paragraph of this section titled “Structural embeddedness and economic performance of ports” refers to some well-established network theories that connect nodes’ network properties and their performance/outcome, aiming at explaining to readers why in this section we are interested to analyze the association between ports’ properties in the GLSN and their economic performance. And we wrote further explanatory text at the very beginning of this section’s second paragraph (lines 441 – 444, page 15), which we now paste below.

<< As such, we are interested to test the relatedness between ports’ economic performance and various patterns of structural embeddedness. That is, in spite of individual differences in other factors (e.g. geographical and political factors), we would expect that structurally similar ports would have similar economic performance. >>

- lines 760-763: conclusions about port policy should also refer to shipping lines' port choice and port competition

REPLY #42: Thanks for the comment. Now in the present revised manuscript, shipping lines' port choice and port competition are referred to in the corresponding text (now lines 466 – 472, page 15). Please check the revision as follows.

<< Such results suggest that, for the development of an individual port, adding connections with ports outside its own modular community would help it attract more traffic capacity from liner shipping companies than adding connections within its own modular community, regardless of how the added outside-module connections are distributed among different modular communities. It is worth mentioning that how many and which kind of connections can be added to any given port largely depends on market factors such as shipping companies' port choice and port competition. >>

- we found that ... (line 777): such results look like new explorations out of the blue, not based on a clear question and not sufficiently interpreted

REPLY #43: In that line 777 (of the last revised manuscript), we write that “we found that the gateway-ness and the rich-club coefficient of ports was correlated:”. This analysis was made to further quantify the difference between ports' gateway-ness and rich-club coefficient, as suggested by the reviewer #3 in the first round of revision. We think that suggestion of the reviewer #3 is constructive and valuable, and therefore we had deeply addressed it and interpreted the related results (which are available in all the previous revised manuscripts). For the whole explanation and interpretation of the content in that line 777, please see the present manuscript text (lines 473 – 483, pages 15 – 16), which is also pasted below. And we also kindly encourage the reviewer #1 to refer to our previous reply #1 to the reviewer #3, in our reply letter of first-round revision.

<< We further characterized the gateway-ness by comparing it to the rich-club coefficient, which is a degree-based topological indicator of the rich-club effect in a network (i.e., a tendency for high-degree nodes to be more densely connected among themselves than nodes of a lower degree). The rich-club coefficient quantifies a special pattern of local embeddedness. Specifically, we adopted the unnormalized version (φ) originally proposed in reference⁵⁰ and two normalized versions (ρ_C and ρ_{CM} , proposed respectively by Colizza et al⁵¹ and by Cannistraci and Muscoloni⁵²) to calculate ports' rich-club coefficients in the GLSN (Supplementary Fig. 6). It turned out that the Pearson correlation coefficient between the rich-club coefficient—should it be normalized or not—and port capacity was significantly lower than that between the gateway-ness and port capacity (Table 1). And the structural core is not the same as any possible rich club of the GLSN (Supplementary Fig. 7). >>

- figure 10 shows the major ports in the world; again, why not calculating the simple correlation with throughput (TEUs) to check whether the whole analysis is finally providing this simple evidence?

REPLY #44: Fig. 10 (now numbered as supplementary fig. 7 in the present revised manuscript) shows the overlap between the rich club and the structural core of the GLSN, and the related explanation and interpretation of this figure are provided (now in the caption of this supplementary fig. 7). Again, we kindly notice the reviewer #1 that this entire analysis was suggested by reviewer #3. These points have already been carefully replied in our reply #43 above; for more details, please refer to the reply #43.

Regarding the data on individual ports' throughput (TEUs) that are now mentioned here by the reviewer #1, unfortunately, we do not have any access to such data. To the best of the authors' knowledge, such data for individual ports in the world are not publicly or freely available. And we had already clarified this data issue in our previous reply #34 to reviewer #1, in our reply letter of first-round revision.

- correlation with trade data on countries and pairs (and fig 11): should refer to Guerrero et al 2015 (maritime networks book) who use a gravity model and restrain the analysis to trade of containerized products (needs a methodological introduction)

REPLY #45: The recommended book chapter by Guerrero, Grasland, and Ducruet (2015) had already been cited and introduced in our manuscript, as illustrated in our previous reply #38 to reviewer #1 in our reply letter of first-round revision. We now have further added a methodological introduction of this recommended book chapter; please see the modification in the present revised manuscript (supplementary note 2; lines 592 - 595, page 39, supplementary information file), which is also pasted below.

<< This study follows the meaningful attempts of previous studies^{12,13} to quantitatively explore the influence of containerized shipping on international trade flows; for instance, reference¹² adopted a gravity model to analyse the influence of shipping distance and frequency of shipping services on containerized trade flows between countries. >>

Discussion

- relation between network structure and functional dynamics: not clear

REPLY #46: Thanks for the comment. This original phrase "the interface between network structure and functional dynamics" appeared in (and only in) the first sentence of the second paragraph of the Discussion section, in the previous manuscripts. To be more prudent, we now have revised this original phrase into "the interface between network structure and functional outcomes"; please see the revised manuscript text (lines 590 – 592, page 19), which is also pasted below. The reason

for this revision is that, in the present study we did not analyze the evolution/dynamics of the GLSN, due to the unavailability of high-quality longitudinal data on global liner shipping service routes (which was extensively explained in our previous reply #30 to reviewer #1, in our reply letter of first-round revision).

<< From the point of view of network analysis, our study offers new tools to investigate the interface between network structure and functional outcomes^{7,43,48,54,55} in real-world networked systems. >

- the discovered structure may simply be a group of the largest ports

REPLY #47: As we had already explicitly addressed in the above reply #37 to the reviewer #1, the discovered structural core is not simply a group of the largest ports. Please find there our detailed reply. Once again, we also kindly notice the reviewer that, in our reply letter of second-round revision, in our reply #22 to reviewer #1, this same comment was already carefully addressed.

- lack of longitudinal data: see studies of maritime networks by Ducruet over 120 years 1890-2010 or 30 years 1977-2008

REPLY #48: This point about “lack of longitudinal data” appears in the first sentence of the last paragraph in the Discussion section. We quote that sentence below (lines 637 – 639, page 20), which is now revised as follow:

“One limitation of existing studies of GLSN that are based on liner shipping service routes data at the global level, including the present study, is the lack of longitudinal analyses of the GLSN structure due to the current limited open access availability of such longitudinal data.”

Clearly, what is claimed in the above sentence is that, of existing studies of GLSN that are based on liner shipping service routes data at the global level, one limitation is caused by the current limited open access availability of such longitudinal data. Within such context, it is explicit that we are referring to the limited availability of longitudinal data on liner shipping service routes at the global level, which is by nature very different from the longitudinal data on vessels’ movement trajectories (as used in studies previously published by Ducruet). And we sincerely remind the reviewer of the following two related points: (1) the difference between these two types of shipping data had been carefully discussed in both the previous revised manuscripts and the present one (lines 671 – 693, page 21), and had also been addressed in our previous replies #6 to reviewer #1 and reply #1 to reviewer #2 in the reply letter of first-round revision; (2) the limited availability of longitudinal data on liner shipping service routes at the global level was extensively addressed in our previous reply #30 to reviewer #1 in the reply letter of first-round revision.

REVIEWERS' COMMENTS:

Reviewer #1 (Remarks to the Author):

General comment

Well improve article, although there remains difficulties to follow the overall logic. The findings are not sufficiently confronted with reality. I still doubt about the innovation brought by the study to transport and network studies.

Detailed comments

- article overly too long
 - needs a thorough checking of the English language quality (grammar) by a professional native editor
 - page 3 lines 67-68: "by means of innovative and advanced" reads rather awkward, consider changing how to "criticize" existing research for its lack of innovation...
 - central question lines 76-79 too long and hard to follow
 - lines 82-84 industry question also hard to follow
 - line 144: why exponential function? usually by a power law to check for scale-free distr.
 - neutral assortativity: no, it is negative, even not significant so it should be emphasized
 - most ubiquitous properties: unclear why it is ubiquitous, consider revising
 - lines 253-256: this question should be placed in introduction not here, as one subquestion of the central question
 - lines 267-268: port development is also explained by transshipment in many cases
 - lines 247-253: except from statistical measures, we need a confrontation between results (3 types of hubs) and "economic reality", a discussion on how do the results match the functions of these ports in the real world
- Please take some concrete examples and give more arguments otherwise the discussion is too abstract
- structural core: as no reference is given, consider referring to the notion of rich-club effect
 - lines 319-321: discuss why Houston and Savannah are included compared with New York or Los Angeles for instance
 - the case of Reykjavik: check whether this corresponds to "anomalous centrality" as shown by Guimera et al. 2005 for the airline network
 - lines 457-460: relationship between degree/throughput and BC/throughput was already tested by Deng et al. 2009 (not innovative here)
 - "adding connections outside its modular community": far from reality, consider bridging your results with realistic challenges for ports and shipping lines

REVIEWERS' COMMENTS:

Reviewer #1 (Remarks to the Author):

General comment

Well improve article, although there remains difficulties to follow the overall logic. The findings are not sufficiently confronted with reality. I still doubt about the innovation brought by the study to transport and network studies.

Reply #1: We thank the reviewer for acknowledging the improvement of our revised manuscript, and we are confident that the overall logic is sufficiently clear. To illustrate why our findings are sufficiently confronted with the reality in the liner shipping sector, we now have provided more discussion on the policy relevance of the proposed measures of gateway-ness and gateway-hub structural core to a realistic challenge facing port authorities and national governments. Please check out the added text in the third paragraph of the Discussion section, which is also pasted below. And we believe that our study brings valuable innovation to transport and network studies, as we have carefully discussed throughout the Discussion section.

<< Well-positioned ports enabled by frequent and regular liner shipping services are key to countries' access to global markets. A realistic challenge that faces port authorities and national governments is to effectively monitor how well their ports are positioned in the evolving GLSN, relative to any other ports of interest. We suggest that ports with largest gateway-ness values form a structural core that is positioned at the center of the GLSN (Fig. 7), and thus the proposed measures of gateway-ness and gateway-hub structural core can have policy relevance to quantitative monitoring ports' GLSN positions. We emphasize how the insight brought by the proposed measures distinguishes from that offered by the traditional measure of container throughput, in monitoring the relative competitive positions of two competing ports. Take, for example, the case of ports of Hong Kong and Guangzhou (which are two geographically adjacent world-class ports in fierce competition): Guangzhou port has surpassed Hong Kong port in terms of container throughput; Hong Kong, however,

remains more competitively positioned within the GLSN, in the sense that it is still a part of the structural core of the GLSN whereas Guangzhou has not yet become (Fig. 7). >>

Detailed comments

- article overly too long

Reply #2: Now the article length has been further shortened.

- needs a thorough checking of the English language quality (grammar) by a professional native editor

Reply #3: We have further checked and improved the English throughout the manuscript.

- page 3 lines 67-68: "by means of innovative and advanced" reads rather awkward, consider changing how to "criticize" existing research for its lack of innovation...

Reply #4: We have removed this phrase and revised the corresponding sentence as follows (see, in the present revised manuscript, the second-to-last sentence in the second paragraph of the Introduction section):

<< The GLSN, though having been investigated by previous studies from a classical network perspective¹⁰⁻¹⁵, introduction of innovative network science methodologies that characterize the structural organization complexity of GLSN and its complex system association with international trade remain a scientific ambition to pursue. >>

- central question lines 76-79 too long and hard to follow

Reply #5: We have shortened this sentence as follows (see, in the present revised manuscript, the first sentence in the third paragraph of the Introduction section):

<< These facts motivate us to address a central research question of the present study: what specific topological properties and organization principles characterize the GLSN structure, and how are they associated with the GLSN's functional outcomes? >>

- lines 82-84 industry question also hard to follow

Reply #6: We have improved this sentence as follows (see, in the present revised manuscript, the third sentence in the third paragraph of the Introduction section):

<< This corresponds to a specific question in liner shipping industry: which ports are the most important hubs in the GLSN structure, in the sense that they form a core that facilitates the efficiency of cargo transportation in the network? >>

- line 144: why exponential function? usually by a power law to check for scale-free distr.

Reply #7: We kindly notice the reviewer that this same question had already been carefully addressed in our previous reply letter of the first-round revision (reply #40 to the reviewer #1). We now once again reply as follows.

First, we had already checked if the ports' degree distribution can be well fitted by a power-law distribution; the results show that it cannot. Please see in the first sentence of the caption of fig. 2, which we now also paste below.

<< In panel **a**, complementary cumulative distribution function of degree is reported in log-log scale (dots), fitted by an exponential function (dash line) instead of power-law; tests on the power-law distribution of the data failed based on the method of Clauset et al⁵⁹ and the method of Voitalov et al⁶⁰ as well. >>

Second, we had already explained and discussed the exponential distribution of ports' degree. Please see, in the supplementary information file, the first paragraph of the subsection named "Degree centrality distribution and assortativity" of the supplementary note 2. It is now also pasted below.

<< Indeed, unlike many real-world networks where the vertex connectivities follow a scale-free power-law distribution¹⁷, spatial networks where nodes are embedded into real space usually face certain spatial constraints restricting the appearance of so large degree nodes that would facilitate scale-free behaviors of node degree distribution¹⁸. While not scale-free, node degrees for single ports can vary over a great significant range of two-digit magnitude. Specifically, ports of Rotterdam and Antwerp have the highest degree of 324, while a lot of small ports are of the lowest degree of 1. Indeed, world liner shipping activities concentrate heavily on a few big ports; for instance, the top 30 ports account for over half of the total global container throughput in the year 2015. Most of those big ports are intermediate hubs, whose emergence was mainly fostered by world liner shipping companies in the pursuit of economies of scale (Rodrigue and Notteboom, 2010)¹⁹. Since the mid-1990s, the world has seen a fast increase of vessel size, frequency of liner shipping companies' services, and the concentration of their services at a few large intermediate hubs (Zohil and Prijon, 1999)²⁰. >>

- neutral assortativity: no, it is negative, even not significant so it should be emphasized

Reply #8: As the reviewer also acknowledges, here the assortativity value (= -0.024) apparently cannot be claimed significantly negative. But the reviewer's comment reminds us to be more prudent; we have modified the original expression "the GLSN presents a neutral assortativity (= -0.024)" into "the GLSN is close to a neutral assortativity (= -0.024)".

- most ubiquitous properties: unclear why it is ubiquitous, consider revising

Reply #9: Thanks for the comment. To be more prudent, we have changed the word “ubiquitous” to the word “prominent”.

- lines 253-256: this question should be placed in introduction not here, as one subquestion of the central question

Reply #10: Thanks for the comment. We agree that this question sentence, in writing style, is not well in harmony with the local text, although the information it contains is necessary to be provided there. Hence, we have modified this question sentence into the one pasted below; please check this modification in the last sentence of the third paragraph of the subsection titled “Multiscale modularity and hubs diversity”. We think this modification is more straightforward and suitable than the one suggested by the reviewer.

<< These three indicators help explore whether ports present some connectivity patterns similarities in the structure of the GLSN. >>

- lines 267-268: port development is also explained by transshipment in many cases

Reply #11: We agree with the reviewer’s comment, which does not contradict the point conveyed in our sentence in lines 267-268 (of the last revised manuscript version). To be more prudent, we however have modified the sentence as follows (the second-to-last sentence in the fourth paragraph of the subsection titled “Multiscale modularity and hubs diversity”, in the present revised manuscript):

<< Indeed, port development is mainly related with the development of regional economy and international trade³⁸. >>

- lines 247-253: except from statistical measures, we need a confrontation between results (3 types of hubs) and "economic reality", a discussion on how do the results match the functions of these ports in the real world

Please take some concrete examples and give more arguments otherwise the discussion is too abstract

Reply #12: In lines 247-253 (of the last revised manuscript version), we define three types of hub ports based on three respective topological indicators: provincial-ness, connector-ness, and gateway-ness (which is for the first time introduced by the present study). We stress to the reviewer that, our results highlight only the proposed measures of gateway-ness and the gateway-hub structural core, towards a better understanding of the structural organization complexity of the GLSN. In line with this statement, we now have shown a meaningful confrontation between the insight provided by the proposed measures and the “economic reality” reflected by a traditional measure, by taking a concrete example of two competing ports. Please check out the added text in the end of the third paragraph of the

Discussion section.

<< We emphasize how the insight brought by the proposed measures distinguishes from that offered by the traditional measure of container throughput, in monitoring the relative competitive positions of two competing ports. Take, for example, the case of ports of Hong Kong and Guangzhou (which are two geographically adjacent world-class ports in fierce competition): Guangzhou port has surpassed Hong Kong port in terms of container throughput; Hong Kong, however, remains more competitively positioned within the GLSN, in the sense that it is still a part of the structural core of the GLSN whereas Guangzhou has not yet become (Fig. 7). >>

Unfortunately, due to the unavailability of the data on actual seaborne trade volume among ports, we are unable to precisely evaluate ports' transportation functions in the real world. Therefore, we are unable to further analyze or discuss how the results for ports' gateway-ness match their functions in the real world.

- structural core: as no reference is given, consider referring to the notion of rich-club effect

Reply #13: We prefer to keep it as it is, for the following reasons. First, the notion of gateway-hub structural core is indeed not cited from any existing studies, but instead is for the first time proposed in the present study. Second, it would be improper to refer to the notion of rich club when introducing the notion of structural core, as the two are by definition very different from each other. And we had already shown that the structural core is not the same as any possible rich club of the GLSN; for details, please refer to supplementary fig. 7 in the supplementary information file.

- lines 319-321: discuss why Houston and Savannah are included compared with New York or Los Angeles for instance

Reply #14: Here the reviewer refers to the result that ports of Houston, Savannah, and New York are included in the gateway-hub structural core, whereas the port of Los Angeles is not. We do not think a further discussion on the issue "why Houston and Savannah are included compared with New York or Los Angeles for instance" would significantly improve our manuscript, for the following reason. As clearly illustrated in fig.6a, the gateway-hub structural core is a concept built on the modular structure of the GLSN, which is essentially defined by the topology of the entire network (i.e., the connections among all ports). That is, whether or not any given port is included into this structure core is determined by all the connections among all the ports in the network. Therefore, within the framework of the structural core analysis proposed here, it can hardly bring us any meaningful insights to discuss or explain why any given ports are included into this structure core by comparing with only a few other ports.

- the case of Reykjavik: check whether this corresponds to "anomalous centrality" as shown by Guimera et al. 2005 for the airline network

Reply #15: Guimera et al. 2005 studied the anomalous centrality of a global air network. We checked their paper but did not find any specific analysis or information about Reykjavik. Therefore, we are unable to know whether the airport Reykjavik presents anomalous centrality, not to mention whether the case of Reykjavik in the GLSN (i.e., the anomalous centrality of the seaport Reykjavik) corresponds to the anomalous centrality of the airport Reykjavik. But we are confident that the unavailability of such information in their paper does not at all negatively impact our present study.

- lines 457-460: relationship between degree/throughput and BC/throughput was already tested by Deng et al. 2009 (not innovative here)

Reply #16: What we write in the manuscript does not at all contradict the reviewer's comment here. In lines 457-460 (of the last revised manuscript version), we write the following (which is now further improved in writing): "The finding that degree performs better than (or at least as good as) betweenness is interesting, indicating that local embeddedness may be an important factor for port economic performance. Different from nodes in many other real-life complex networks where more connections do not necessarily mean better outcome⁴⁰, in the case of container port development within the context of the GLSN, it seems that the more connections a port has the more traffic capacity it will get." As the reviewer can see, here (and throughout the manuscript) we never claim this finding were something innovative in our study.

- "adding connections outside its modular community": far from reality, consider bridging your results with realistic challenges for ports and shipping lines

Reply #17: Thanks for the comment. This phrase appeared in line 467 of the last revised manuscript version, which we now have modified as follows: "Such results suggest that, for the development of an individual port, having connections with ports outside its own modular community". In short, the overall results in the present study highlight the importance of the proposed measures "gateway-ness" and "gateway-hub structural core" to a better understanding of port development issues within the context of the GLSN. To bridge our results with realistic challenges in the liner shipping sector, we now have discussed the policy relevance of our proposed measures to a realistic challenge facing port authorities and national governments; for details, please check our reply #1 to Reviewer #1. Besides, please note that, we had already bridged our results regarding the measure of gateway-hub structural core with a realistic challenge facing liner shipping companies: the problem of hub port selection. Please see the second and third sentences in the second paragraph of the Discussion section, which we now paste below.

<< A good knowledge of the gateway-hub structure-core of the GLSN may help liner shipping companies better select the hub ports' locations^{17,18}, in the sense that it helps understand

how the selected hub nodes can serve as transshipment and switching points to improve the overall efficiency and economy of flow transportation in the entire network¹⁶. Indeed, the hub-and-spoke network configuration has been widely adopted in the design of liner shipping service networks and also many other transportation systems (e.g., air and railway transportations), and one of the most crucial issues in designing such network configuration is to strategically select hub nodes. >>